# Dynamic characterization and interpretation for protein-RNA interactions across diverse cellular conditions using HDRNet

Haoran Zhu[1], Yuning Yang[2], Yunhe Wang[3], Fuzhou Wang[4], Yujian Huang[5], Yi Chang[1], Ka-chun Wong [4] ✉ & Xiangtao Li [1] ✉

RNA-binding proteins play crucial roles in the regulation of gene expression, and understanding the interactions between RNAs and RBPs in distinct cellular conditions forms the basis for comprehending the underlying RNA function. However, current computational methods pose challenges to the cross-prediction of RNA-protein binding events across diverse cell lines and tissue contexts. Here, we develop HDRNet, an end-to-end deep learning-based framework to precisely predict dynamic RBP binding events under diverse cellular conditions. Our results demonstrate that HDRNet can accurately and efficiently identify binding sites, particularly for dynamic prediction, outperforming other state-of-the-art models on 261 linear RNA datasets from both eCLIP and CLIP-seq, supplemented with additional tissue data. Moreover, we conduct motif and interpretation analyses to provide fresh insights into the pathological mechanisms underlying RNA-RBP interactions from various perspectives. Our functional genomic analysis further explores the gene-human disease associations, uncovering previously uncharacterized observations for a broad range of genetic disorders.

RNA-binding proteins (RBPs) are an essential group of proteins that interact with RNA by recognizing specific RNA-binding domains, and are involved in post-transcriptional regulation of RNA splicing, translocation, sequence editing, intracellular localization, and translational control[1-3]. Accurately identifying RBP binding states in specific cellular conditions is a significant challenge that required for unraveling the underlying regulatory mechanisms and understanding their biological function. Traditional experiment-based biological methods such as systematic evolution of ligands by exponential selection (SELEX)[4], RNAcompete[5], and RNA Bind-n-Seq[6] have been developed to characterize the sequence preferences of RBPs in vitro, while RNA immunoprecipitation (RIP)[7] and other immunoprecipitation-based technologies[8-10] were proposed to identify RBP binding sites in vivo. Unfortunately, these laboratory-based experiments are time-consuming,

labor-intensive, and susceptible to measurement errors. Therefore, developing high-throughput, accurate, and robust approaches to investigate RBP binding modes is of great importance[11].

Thanks to the development of cross-linked immunoprecipitation sequencing technology[8,12], many RBP-RNA binding targets have been uncovered[13,14], enabling us to develop effective data-driven computational methods[15]. These methods can be broadly classified into two categories[16], either predicting RNA-binding sites on the protein surface[17] or modeling the preferred RNA sequences of RNA-binding proteins[18]. From the protein perspective, several computational tools have been developed to predict the RBP-RNA binding sites at the protein level. For instance, SCRIBER, uses predictions of binding residues for several partner types to effectively reduce cross-prediction of the output protein-binding residues, combining novel and previously

[1]School of Artificial Intelligence, Jilin University, 130012 Changchun, China. [2]Donnelly Centre for Cellular and Biomolecular Research, University of Toronto, Toronto, ON, Canada. [3]School of Artificial Intelligence, Hebei University of Technology, Tianjin, China. [4]Department of Computer Science, City University of Hong Kong, Hong Kong, Hong Kong SAR. [5]College of Computer Science and Cyber Security, Chengdu University of Technology, 610059 Chengdu, China. ✉e-mail: kc.w@cityu.edu.hk; lixt314@jlu.edu.cn

used input types[19], while aPRBind, developed by Liu et al., combines protein sequence and structural features for RNA-binding residue prediction[20]. DRNApred[21], a fast sequence-based method that accurately predicts and discriminates DNA- and RNA-binding residues was proposed by Yan et al., with regression that penalizes cross-predictions, and a two-layered architecture. However, these statistical or machine learning-based algorithms often encounter performance limitations as the size of the dataset continues to grow. This is primarily due to their inability to effectively capture complex patterns and relationships in large-scale data, which can lead to unsatisfactory predictive performance. Recently, to address such limitations, deep learning methods have been developed; for instance, DeepSite employs a 3D deep convolutional neural network to predict the binding site using protein structure[22]. Xia et al. proposed GraphBind, an end-to-end graph neural network that uses hierarchical graph neural networks to identify nucleic-acid-binding residues on proteins[23]. Zhang et al.[24] presented DeepDISOBind, a deep multi-task architecture that accurately predicts DNA-, RNA- and protein-binding regions from protein sequences. Most recently, Lam et al. introduced NucleicNet, a deep learning model that predicts the binding preference of RNA backbone constituents and different bases from local physicochemical characteristics of the protein structure surface[25]. These methods have collectively enhanced our understanding of the binding properties of RBPs at the protein surface level.

In parallel, the establishment of links between RBPs and their targets from the perspective of RNA sequences enables understanding of the regulation mechanism. A variety of efforts have been developed to address it in identical cellular conditions, referred to as static protein-RNA interactions; for instance, Deepbind[26] was developed to understand RBP binding preferences in RNA sequence data using a deep neural network (DNN). Ilan et al. developed DLPRB[27], a new DNN approach based on convolutional neural networks (CNN) and recurrent neural networks (RNN) for learning intrinsic RBP binding preferences and predicting novel interactions. Pan et al. developed iDeep[28], a hybrid convolutional neural network and deep belief network-based model to predict the RBP interaction sites and motifs on RNAs. Daniel et al. proposed GraphProt[29] that integrates sequence and computationally predicted RNA secondary structure information into graph-kernel features. Laverty et al. introduced PRIESSTESS[30], a universal RNA motif-finding/scanning strategy capable of identifying enriched RNA sequence and/or structure motifs that are subsequently reduced to a set of core motifs by logistic regression with LASSO regularization. However, these methods only account for a specific cellular condition and are hence limited in their ability to predict RNA-protein interactions in other cell lines while considering the dynamic contexts.

Indeed, the binding behavior of RBP-RNA interactions has been demonstrated to be dynamic in different cell types, as it is influenced by different cellular or tissue environments[31]. In other words, the binding sites of the RBP that are exclusively present in specific cells or tissues can be designated as dynamic protein-RNA interactions. Recently, a new deep learning-based method called PrismNet[32], was developed to accurately predict dynamic RBP binding in various cellular conditions by integrating in vivo experimental RNA structure information[33]. Nonetheless, unfortunately, we conceive that there are still room for improvements; for instance, the architectural design of the deep neural network within PrismNet can still be investigated in the context of neural network architecture search, indicating the potential for further improvements in RBP-RNA binding prediction. In addition, the one-hot encoding representation in PrismNet exhibits limitations that may hinder its ability to capture RNA features globally. Furthermore, the heterogeneity of transcripts results in widely sequential relationships across different contexts, such as various cell lines, tissues, and normal or disease conditions. The one-hot encoding may overlook this contextual information, treating each position independently and potentially missing crucial sequence patterns. Therefore, it is crucial to address these limitations by exploring alternative coding methods to leverage the contextual and sequential nature of RNA sequences.

The contextual relationship has been a focus of increasing attention, as word embedding techniques[34–36] have proven to be effective frameworks for automatically encoding RNA sequences due to their syntactic, lexical, and semantic similarities to human language[37]. On this basis, much effort has been devoted to the application of advanced NLP techniques in RNA-related problems[38–40]. However, most existing research on the application of NLP techniques in RNA-related problems has relied on static embedding models such as Word2Vector[41], GloVe[35], Doc2Vector[36], and FastText[42]. Similar to one-hot representation, static embedding methods maintain the same nucleotide encoding across all contexts, which leads to a failure to capture the context-based dynamic semantic information of RNA sequences and therefore cannot represent RNA sequence features globally. Moreover, static embedding does not capture the underlying information of nucleotide sequences in different cellular contexts, thus limiting the ability of dynamic prediction in other cellular conditions. To address this knowledge gap, we propose to adopt and customize the Bidirectional Encoder Representations from Transformers (BERT) model[43] to learn the context-dependent information of RNA sequences, which can generate robust expressions containing the global contextual information by pre-training with a large-scale unlabeled text corpus in a self-supervised fashion. In contrast to static embeddings where nucleotide embeddings remain the same regardless of the context, the primary advantage of dynamic embeddings lies in their ability to generate context-specific features in different nucleotide contexts. Moreover, considering the heterogeneity of transcripts under different cellular, tissue, or physiological conditions (e.g., normal or diseased), dynamic embeddings can overcome the limitation of a single representation for diverse contextual sequences, enriching the global features of the sequences and surpassing the performance bottleneck in dynamically predicting RBP binding sites. Therefore, by leveraging the advantages of the transformer model, we adopt the BERT model to encode RNA sequences in dynamic embedding vectors that then contain rich, global contextual semantic information for identifying dynamic RNA-binding events between different cell lines, tissues, or physiological conditions (normal or diseased).

Here, we propose HDRNet (High-throughput Dynamic Cellular RNA-binding Event Identification using Deep Neural Network), a new end-to-end deep learning model for identifying RNA-binding interactions from eCLIP-seq data in various cellular conditions. To capture the hierarchical relationships between nucleotide sequences, we adopt multi-source biological information, including in vivo RNA secondary structure information and bio-language features, to characterize both the sequence and structural features of RNA. Then, we combine biological information from different sources using the unified alignment model to uncover possible relationships between nucleotide sequences and latent structural information. Additionally, hierarchical multiscale residual networks (HMRN) are leveraged to comprehend the contextual dependencies between the nucleotides, and deep protein-RNA binding predictor (DPRBP) is developed to extract the contextual significance of the nucleotide sequences by stacking several pyramid convolutional blocks, and incorporating batch normalization and residual shortcut connections into the network to boost robustness and reduce overfitting. We evaluate HDRNet on 261 linear RNA datasets and compare it with other baseline methods. We demonstrated its validity and scalability in both static and dynamic cellular conditions. In addition, we performed motif and interpretation analyses to gain new insights into the pathological mechanisms of RNA-RBP interactions. Our functional genomic analysis revealed the association between genes and human diseases, leading to previously unknown observations on a wide range of inherited diseases.

## Results

### Overview of HDRNet

The HDRNet framework has the capability to perform accurate prediction of RBP binding events by leveraging robust features from multi-source biological information to aid in the identification of high-attention binding peaks and subsequent analysis of RBP binding data, as depicted in Fig. 1. The HDRNet pipeline consists of four key components aimed at achieving the reliable prediction of RNA-binding protein (RBP) interactions. (1) The dynamic global contextual information and in vivo RNA secondary structure information are extracted to characterize both the sequence and structural properties of RNA; (2) A unified alignment of multi-source feature representation is developed to embed the dynamic contextual information and icSHAPE contour vectors with dimensional homogeneity to generate the potential feature representations; (3) We establish the hierarchical multi-scale residual network (HMRN) to extract the sequence and structural information and then calculate the binding score of the RNA-RBP interaction using the deep protein-RNA binding predictor (DPRBP), which picks the most prominent nucleotide characteristics progressively; (4) The latent embedded representation learned by the HDRNet model allows capturing the high-focus binding peaks and binding patterns of RNA sequences to investigate the association between dynamic binding sites and human diseases. In addition, it is also possible to jointly interpret gene-level knowledge in a transcriptomic context, providing insights into disease regulatory mechanisms.

The HDRNet framework starts by generating multiple sources of biological information to represent the RNA sequence or structure, where the dynamic global contextual embedding representation comes from tagging the input RNA sequence as genetic codon tokens using the k-mer method and then encoding each token as a dynamic embedding vector using the pre-trained deep bidirectional

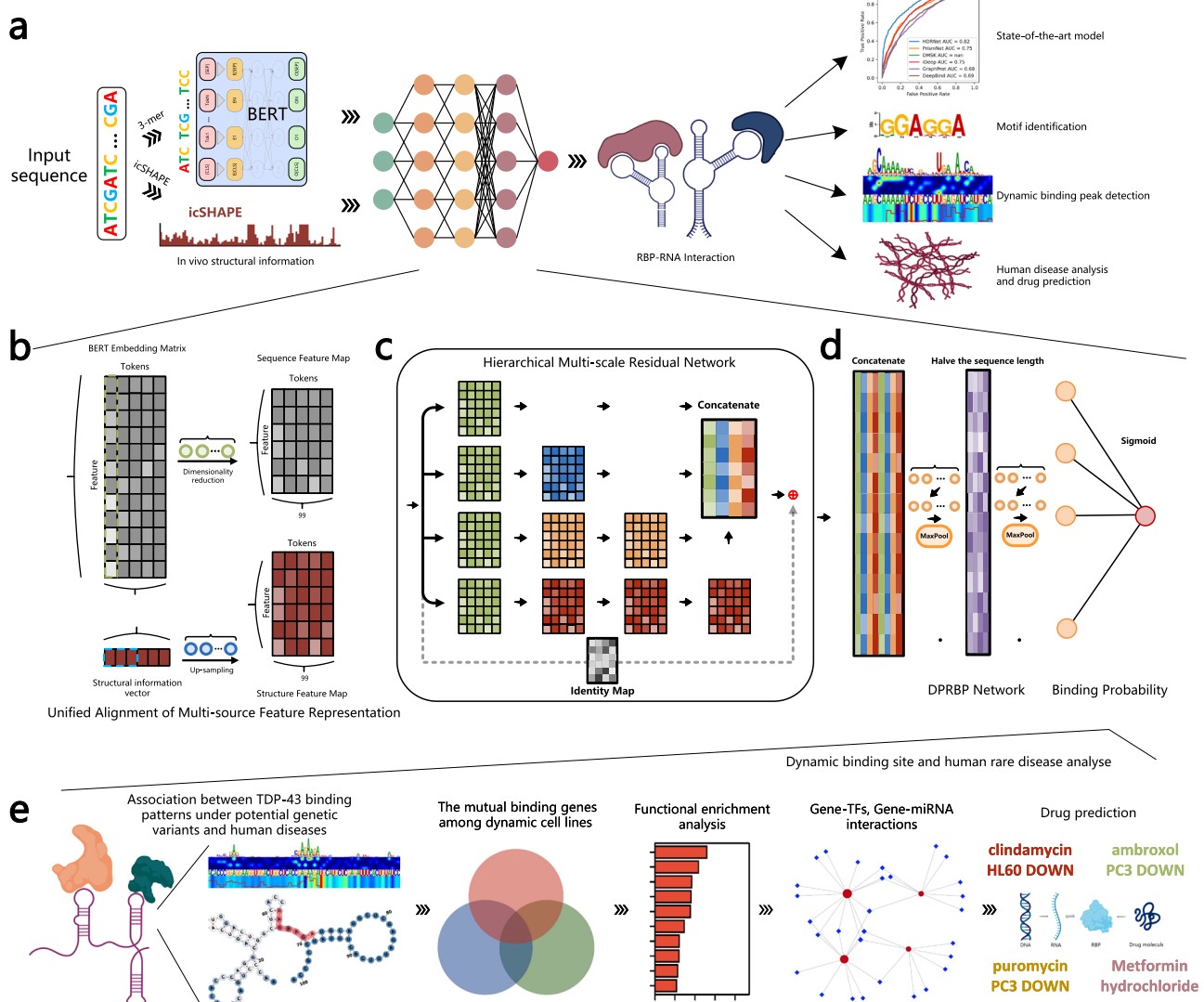

**Fig. 1 | The network architecture of the HDRNet algorithm. a** The overall framework of HDRNet, an end-to-end architecture for RNA-RBP binding identification. After being tokenized, the RNA sequence is fed into the multi-head self-attention layers to generate dynamic contextual embedding. The icSHAPE pipeline is employed to generate RNA secondary structure profiles. **b** The unified alignment of multi-source feature representation in HDRNet. We employ two diverse CNN blocks to enrich feature representation and guarantee an equal output size. **c** The hierarchical multi-scale residual network (HMRN) stage of HDRNet. The sequence and structure feature map is fed into two different multi-scale ResNet so that local contextual dependencies can be extracted. **d** The deep protein-RNA binding predictor (DPRBP) stage. We developed a multi-layer feature selector module to capture significant context-sensitive information on the sequence before giving a final prediction. **e** A variety of functional genomic analyses are carried out to shed new light on the interpretation of RBP datasets and possible treatment of neurological diseases.

transformer model (BERT). Dynamic embedding implies that the same token has different encoding in different contexts and therefore contains more nucleotide contextual information as well as long-distance dependencies. Meanwhile, the icSHAPE-pipeline is employed to analyze and generate the in-vivo RNA secondary structure profile, which is a numerical vector of the same length as the RNA sequence, providing the model with valuable information on the structural binding preferences of RBP.

Afterward, HDRNet utilizes a unified alignment of multi-source feature representation, which consists of two convolutional neural network (CNN) modules, to preliminary extract the underlying features of both the sequence and structure, and to unify their feature dimensionality, respectively. HDRNet then adopts two HMRNs to analyze the sequence and structural features independently. Each extractor comprises of CNN modules with distinct kernel size to comprehend the contextual dependencies between the nucleotides and their structures at varying distance, for accurate interaction prediction. Thereafter, HDRNet amalgamates all the extracted features and implements the DPRBP with a multi-layer feature selector module, which learns the underlying representation and selects the crucial nucleotide tokens in each layer of the module. Furthermore, HDRNet employs a sigmoid activation function on the network's output to forecast the binding of the RNA sequence to the protein.

Finally, we can identify and align the binding motifs of RNA binding proteins by exploring the interpretability of HDRNet. Further, we delved into the underlying connections between the dynamic binding sites and human diseases by identifying the high-attention binding regions and their potential variants. To support this finding, our study also integrated transcriptomics to unveil disease regulatory mechanisms, including gene ontology enrichment, KEGG pathway analysis, protein-protein interaction network analysis, transcription factor-gene interaction analysis, miRNA-gene interaction analysis, and drug prediction analysis. Our analyses provide new insights into the interpretation of RBP datasets and the treatment of human diseases.

## HDRNet provides better performance than baseline methods in static cellular protein-RNA interaction identification

We evaluated the performance of our proposed HDRNet by comparing it with five other state-of-the-art computational methods, namely PrismNet[32], PRIESSTESS[30], DMSK[39], iDeep[28], DeepBind[26], and Graphprot[29] on 261 static RBP binding site datasets. The area under the receiver operating characteristic curve (AUC) was adopted as the performance metric for all computational methods. For each RBP dataset, we partitioned the binding sites into training and test sets. Then, we used the test set to benchmark all prediction methods[32]. Note that we used the data from a uniform pipeline[32], which ensures that the data is accurate and consistent and that the comparison is fair.

Circos plot in Fig. 2a shows the overall experimental results of HDRNet compared to the other methods across the 261 RBP datasets. As depicted in this figure, we notice that HDRNet consistently outperformed other approaches. In particular, our model substantially enhanced the performance in identifying FMR1 and FXR2 binding sites in HEK293 cells; for instance, FMR1: PrismNet = 0.67 vs. HDRNet = 0.80; FXR2: PrismNet = 0.71 vs. HDRNet = 0.87. Moreover, we conducted additional analyses on the associated characteristics of FMR1 and FXR2, identifying their specific features (Supplementary Note 6, Supplementary Fig. 8). Meanwhile, we notice that HDRNet outperformed PRIESSTESS on almost all datasets. This discrepancy may arise from the fact that PRIESSTESS is highly dependent on the motif-extracting process in the initial stage, and may not be able to identify salient patterns across the datasets with insufficient binding patterns, resulting in a limited training feature set. Moreover, the size of the dataset also plays a crucial role in influencing the efficiency of motif recognition. When the dataset is relatively small, it can potentially result in inaccuracies or the failure to recognize motifs, consequently

diminishing the predictive performance. In addition, the logistic regression model based on LASSO regularization within PRIESSTESS could yield suboptimal performance when confronted with nonlinear decision boundaries, while deep learning methodologies tend to exhibit more favorable outcomes. Although PRIESSTESS demonstrated suboptimal performance, its unique Motif extraction process could potentially enhance HDRNet, as illustrated in Supplementary Note 3 and Supplementary Figs. 4 and 5. In the violin plot of Fig. 2b, we observe that our proposed HDRNet performed more consistently and had better prediction performance on the majority of the datasets than the competing approaches. The reason for the improved performance may come not only from the self-supervised capability of the transformer that captures the global contextual and semantic information of the RNA sequences but also from the ability of the proposed network architecture to learn and transform long-range dependencies. We also provide the receiver operating characteristic (ROC) curve of the first four datasets that are plotted in Fig. 2c. The ROC curve analyses demonstrated that HDRNet had a higher true positive rate (TPR) compared to the other methods, indicating that HDRNet has a higher sensitivity for identifying RBP binding sites. Moreover, we also identified existing sub-groups of binding events that are better characterized by HDRNet, as discussed in Supplementary Note 5 and Supplementary Fig. 7. In addition, we used the t-SNE clustering method to analyze the validity of the output feature of HDRNet. As shown in Fig. 2d, we clearly observe that HDRNet provided the best clustering results compared with the other baseline methods, where the positive (RNA fragment that is an RBP binding site) and negative (RNA fragment that is not an RBP binding site) samples are separated, demonstrating the superior feature extraction capability of HDRNet. In summary, these results indicate the high effectiveness and feasibility of the proposed HDRNet.

## New insights by characterizing RNA binding events between different cell lines in a dynamic manner

The binding of RBPs is influenced by different cellular environments, and therefore, this binding is expected to be dynamic in diverse cell lines. BERT extracts the dynamic semantic information from RNA sequences globally, and is capable of revealing different RNA-protein interactions in particular cellular conditions. We evaluated the performance of HDRNet in predicting dynamic RNA-protein interactions on 62 RBP datasets obtained from K562 and HepG2 cell lines.

Specifically, we trained HDRNet on the RBP datasets of the K562 cell line. We then used the corresponding RNA sequence information generated by BERT and in vivo secondary structure information to predict the RBP binding sites in the HepG2 cell line, which was performed as an independent test set. Subsequently, we trained HDRNet on the HepG2 cell line data, and used data on K562 cell line as an independent test set. We then compared the dynamic prediction performance of HDRNet with other state-of-the-art deep learning methods, including PrismNet, PRIESSTESS, iDeep, DMSK, GraphProt, and Deepbind. Figure 3a provides the heatmap of the dynamic prediction results, from which we can see that the proposed HDRNet outperformed the other methods for both K562 and HepG2 predictions while Deepbind performed the worst across the majority of datasets, with either the lowest AUC value or being incapable of making dynamic predictions. When predicting the RBP sites in K562 cells after training on HepG2 cells, the average AUC result of HDRNet was 0.79, which is 4% higher than PrismNet. In particular, the AUC results of RBM15 and XRN2 RBPs are 12% and 9% higher than PrismNet, respectively. Similarly, predicting HepG2 cells binding sites after training on K562 cells, the average AUC result of HDRNet was also 4% better than PrismNet. The results for RBM15, SF3B4 and SLTM also showed improvement over PrismNet, with the performance gains of 9%, 11% and 9%, respectively. Meanwhile, we also notice that HDRNet substantially outperformed PRIESSTESS for the dynamic tasks with

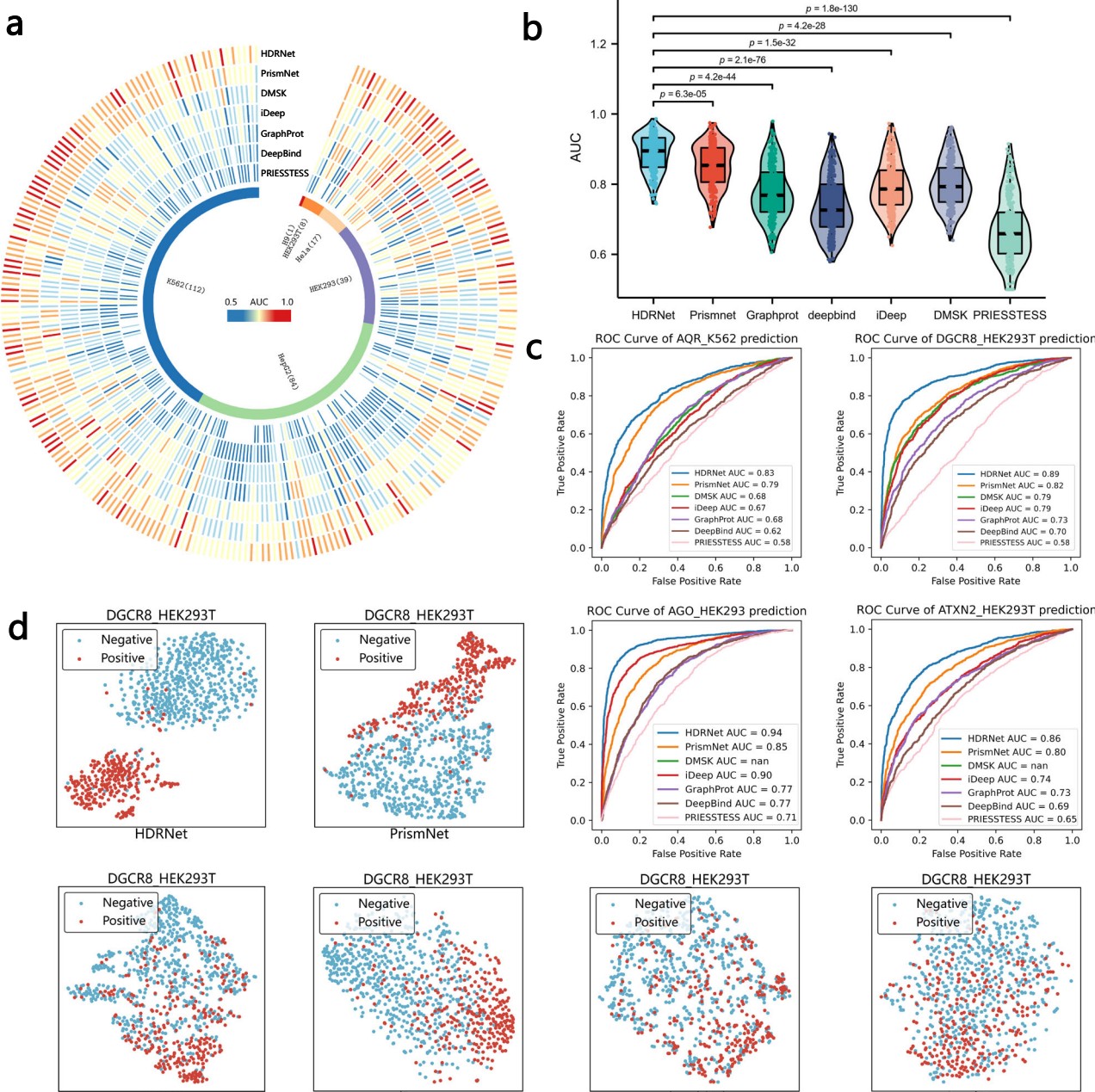

**Fig. 2 | HDRNet predicts RBP binding events more accurately than other state-of-the-art methods. a** The Circos heatmap of the respective AUC scores of HDRNet vs. other methods, including PrismNet, PRIESSTESS, DMSK, iDeep, GraphProt, and DeepBind on all 261 RBP datasets. **b** The violin plot of the overall results of HDRNet and the other methods (*n* = 261 in each group; center line, median; box limits, upper and lower quartiles; whiskers, 1.5× interquartile range; Dunn's test). **c** The ROC curve on the first 4 RBP datasets using HDRNet and the other methods. **d** t-SNE clustering results of the output features of HDRNet and the other baseline methods. Source data are provided as a Source Data file.

higher AUC values while PRIESSTESS outperformed Deepbind, GraphProt, iDeep and DMSK in terms of dynamic prediction. Figure 3b shows the predicted vs observed binding sites of HNRNPA1 on the *MT-ND5* transcript. According to eCLIP, the *MT-ND5* transcript contains 11 HNRNPA1 binding sites in HepG2 cells, and 6 binding sites in K562 cells. We found that HDRNet correctly predicted all 11 binding sites within the *MT-ND5* transcript in HepG2 cells with no false positives, by using the model trained on K562 cells. In contrast, DeepBind and GraphProt, correctly predicted only 2 of the 11 sites, and iDeep and DMSK correctly predicted 4 of the 11 sites, and PrismNet and PRIESSTESS correctly predicted 7 of the 11 binding sites. Figure 3c depicts the ROC curves of the first two datasets. Similar to the before AUC results,

HDRNet had superior dynamic prediction than the other methods supported by better TPR values. Furthermore, we also used t-SNE to represent the significance of HDRNet features in dynamic prediction tasks. As shown in Fig. 3d, we see that the clustering results of HDRNet were still the best under the dynamic prediction task, illustrating the superior feature learning ability and robustness of HDRNet in cross-cell prediction. In addition, we compared HDRNet with seven machine-learning algorithms. As shown in Supplementary Note 1 and Supplementary Fig. 1, HDRNet exhibited better performance compared to these algorithms, in both static prediction and dynamic prediction tasks. Moreover, HDRNet had better performance for RBPs with high and low expression levels and for target RNA events with high and low

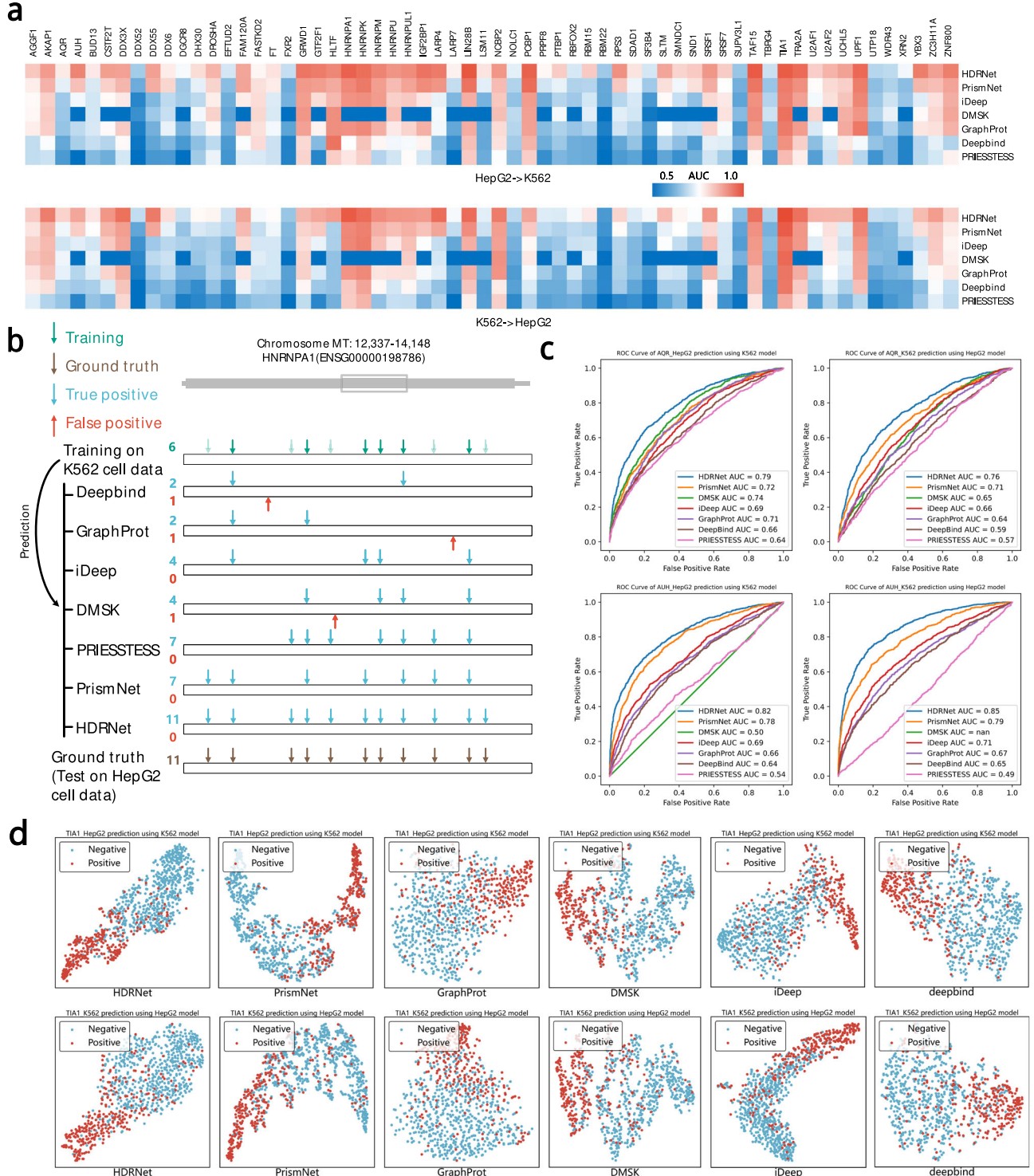

**Fig. 3 | HDRNet successfully performs dynamic RBP binding predictions in both K562 cells and HepG2 cells. a** The heatmap of the respective dynamic prediction AUC scores of HDRNet and other baseline methods, including PrismNet, PRIES-STESS, iDeep, DMSK, GraphProt, and deepbind. **b** Predicted vs observed binding sites of HNRNPA1 on the *MT-ND5* transcript. Green/Brown, observed binding sites in K562/HepG2 cells by eCLIP, used as the training/ground truth reference data; Blue and red indicate, respectively, true positive and false positive predictions in HepG2 cells, based on the models trained using K562 data. **c** The dynamic prediction ROC curve of the first two RBP datasets using HDRNet and the other methods. **d** t-SNE clustering results of the output features of HDRNet and the other baseline methods in a dynamic manner. Source data are provided as a Source Data file.

expression levels in different cellular contexts, as illustrated in Supplementary Note 7 and Supplementary Figs. 9 and 10. In summary, the results indicate that the proposed dynamic contextual information representation scheme contains comprehensive features that are valuable for the identification of dynamic RBP binding sites.

**HDRNet predicts dynamic RNA-protein interactions across tissues in normal and disease conditions and captures significant binding regions**

We went on to explore the dynamic prediction capabilities of HDRNet between different in vivo tissue contexts, especially in normal and

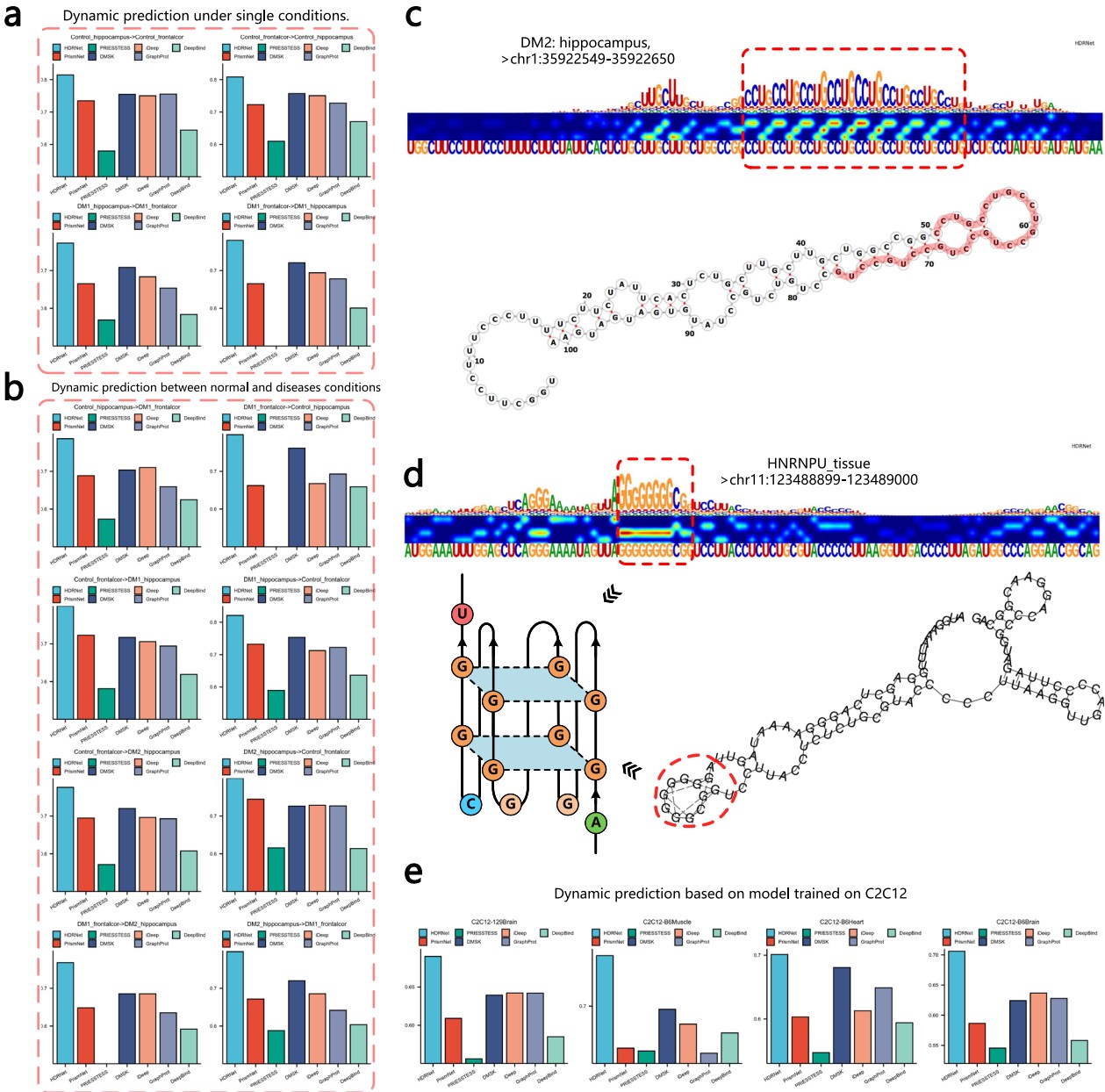

**Fig. 4 | HDRNet Predicts dynamic binding sites across tissues. a** Performance comparison of MBNL2 dynamic prediction in different tissues in human brain. **b** MBNL2 dynamic binding prediction performance comparison across normal and disease conditions. **c** The high attention binding region of MBNL2 in DM2 hippocampus dataset captured by HDRNet. HDRNet successfully identifies the disease-related RNA repeats. **d** HDRNet identifies the salient DGCR8 binding region of G-rich segment with the G-quadruplex structure. **e** HDRNet outperforms other baselines in the dynamic prediction tasks on mice tissues, using model trained on C2C12 dataset. Source data are provided as a Source Data file.

disease conditions. Specifically, we collected the MBNL2 (Muscleblind Like Splicing Regulator 2) binding data[44] (GEO accession: GSE68890) from POSTAR[45], which studied the regulation of MBNL proteins in human brain tissues. In particular, a total of 5 datasets were obtained from autospy tissues (hippocampus and frontal cortex) of patients with myotonic dystrophy type 1 (DM1, 2 datasets), myotonic dystrophy type 2 (DM2, 1 dataset of hippocampus), and control patients (2 datasets). HDRNet and the baseline models, including PrismNet, DMSK, iDeep, GraphProt, DeepBind, and PRIESSTESS were then trained on these datasets separately, and the trained models were tested on different tissues with different control context. As illustrated in Fig. 4a, HDRNet had the best performance in dynamic prediction across tissues comparing with baseline models. Moreover, we notice that PRIESSTESS failed to optimize the binding sites in the frontal

cortex of DM1 dataset, and thus cannot perform dynamic prediction. In addition to dynamic prediction in the same control condition, we also observe that HDRNet demonstrated the capability of dynamic prediction between control context. For instance, as depicted in Fig. 4b, when predicting the DM1-frontal cortex RBP binding sites using the control-hippocampus model, HDRNet achieved the highest AUC of 0.8, whereas the other baseline methods only reached up to 0.7. This indicates that HDRNet could provide potential insights into disease-related biological analyses. Indeed, as revealed in refs. 44,46, MBNL2 directly interacts with DM1 expanded CUG repeats and DM2 CCUG expansion RNAs in the brain, which functionally depletes the MBNL proteins. As illustrated in Fig. 4c and Supplementary Fig. 11a, by employing the SHAP tool[47] to extract high-attention dynamic semantic information, we discovered that HDRNet successfully captured these

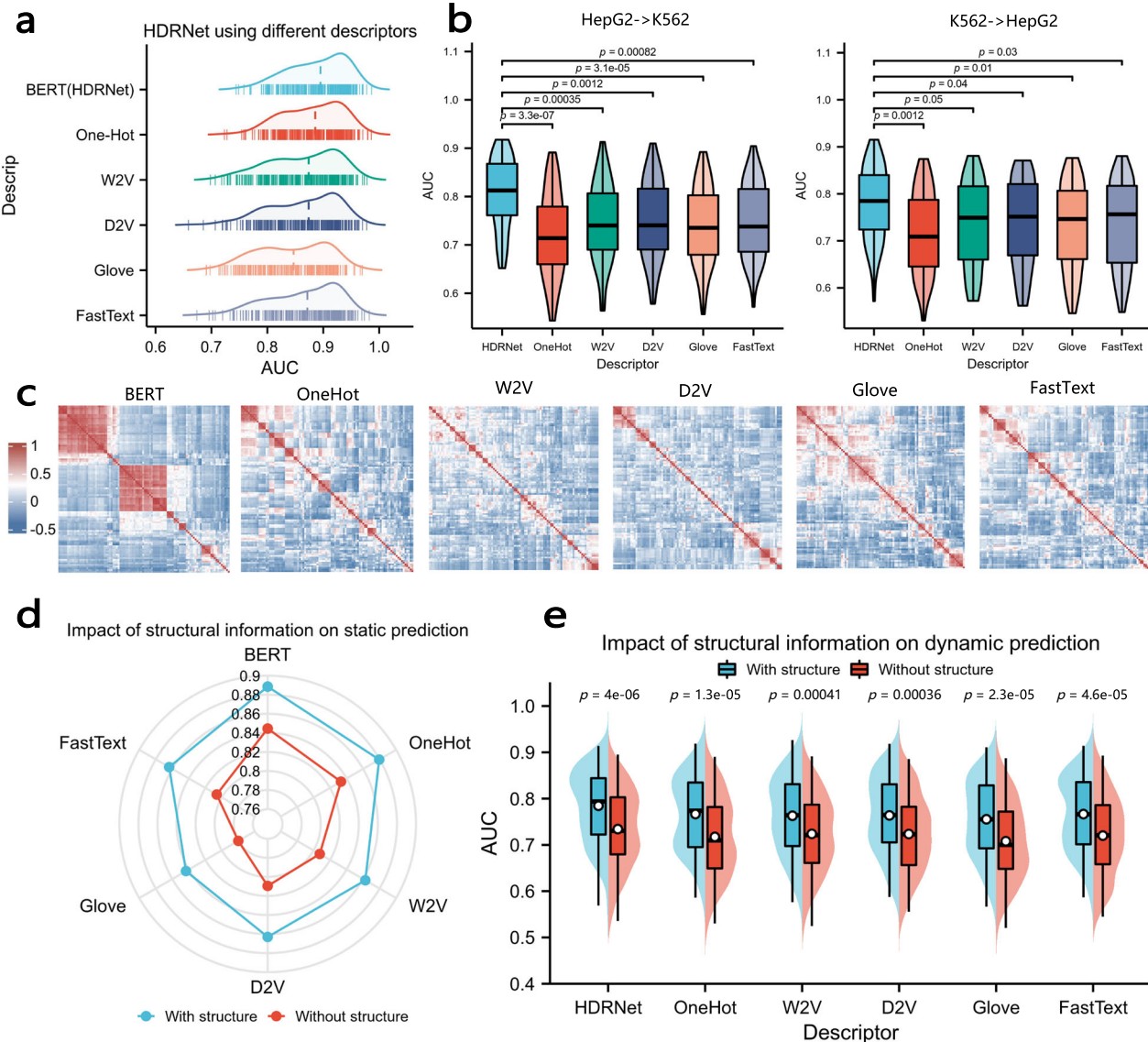

**Fig. 5 | Validation Study of HDRNet on contextual and structure information.**
**a** Overall static prediction performance of HDRNet using different feature
descriptors. **b** Dynamic prediction performance comparison of HDRNet using dif-
ferent feature descriptors ($n = 62$ in each group; center line, median; box limits,
upper and lower quartiles; whiskers, $1.5 \times$ interquartile range; Games-Howell test).
**c** The correlation heat map of features generated by HDRNet using different NLP
descriptors. **d** The average AUC scores for static prediction tasks using different

NLP descriptors for HDRNet with and without in-vivo structure. **e** The performance
comparison of HDRNet using in vivo structures vs. without structural information,
with different NLP descriptors for dynamic prediction tasks ($n=62$ in each group;
center line, median; center dot, mean; box limits, upper and lower quartiles;
whiskers, $1.5 \times$ interquartile range; Wilcoxon rank sum test). Source data are pro-
vided as a Source Data file.

disease-related high-attention regions. For example, in both DM1
datasets, HDRNet highlights continuous segments of CUG expansions,
while in the DM2 dataset, HDRNet also detected significant regions of
continuous CCUG expansions. These findings further confirm the
superior biological interpretability of HDRNet and its potential to
provide theoretical support for pathological research.

Besides tissue-specific dynamic prediction, we asked whether the
HDRNet model trained using cell line data is sufficient for predicting
dynamic interactions in tissues. To clarify this, we retrieved two
additional eCLIP RBP datasets from ENCODE, DGCR8, and HNRNPU,
which were both derived from adrenal gland tissue. We then employed
the models trained on K562 and HepG2 cell line data to validate these
tissue RBP binding data. As expected, Supplementary Fig. 11b
demonstrates that HDRNet performed best when predicting the
dynamic binding in tissue data using the cell line-trained model in both
of the newly retrieved datasets. Interestingly, we observed an

improvement in the performance of PrismNet on the new eCLIP data
compared to the previous MBNL2 data, albeit slightly inferior to
HDRNet. We speculate that this improvement is due to the fact that the
PrismNet model was proposed based on eCLIP data. However, its use
of static encodings limits its performance on data from other plat-
forms, which highlights the advantage of HDRNet. Moreover, HDRNet
notably discerned the prominent binding domains of DGCR8 and
HNRNPU, including the CGG-rich segment associated with DGCR8 and
the G-quadruplex structure in the context of HNRNPU, as illustrated in
Fig. 4d and Supplementary Fig. 11c. In addition, to further validate the
robustness of HDRNet for dynamic prediction of RBP binding sites on
different data platforms, we collected MBNL1 binding data from[46] that
studied the direct regulatory targets of MBNL1 in brain, heart, muscle,
and myoblasts from mice. We obtained a total of five datasets (GEO
accession: GSE39911), including two from the brain, one from muscle,
one from heart, and one from myoblasts (C2C12 cells). As illustrated in

Fig. 4e and Supplementary Fig. 12a, HDRNet provided significant performance improvements compared to other baseline methods in each dynamic prediction task. Furthermore, HDRNet also highlighted the specific disease-related binding preferences similar to MBNL2, as depicted in Supplementary Fig. 12b. Overall, the experimental results and analyses presented above and in Supplementary Note 8 demonstrate that only HDRNet is capable of handling dynamic prediction tasks across different platforms and even different species, highlighting the remarkable effectiveness and robustness of HDRNet. Moreover, through the investigation of RBP data from diseases, we have observed that HDRNet is able to extract the salient binding regions associated with diseases, thereby providing new insights for subsequent pathological studies.

## Validation of HDRNet from the contextual information and structural perspectives

We further discuss the superiority of HDRNet from the perspective of NLP methods and RNA secondary structure, respectively. We first conducted experiments comparing BERT with other existing static embedding methods, including Word2Vector[41], Glove[35], Doc2Vector[36], FastText[48], and the One-Hot coding scheme. Briefly speaking, the difference between dynamic and static methods is that dynamic encoding generates different embeddings in different contexts, while the embedding of each token is fixed in the static encoding methods. Figure 5a depicts the consolidated results of static prediction on all 261 datasets using different feature descriptors. We notice that integrating BERT brought the best performance compared with the other static embedding methods, with an average AUC of 0.84 for HDRNet using BERT on the 261 RBP datasets, much higher than One-Hot (0.83), Word2Vec (0.80), Doc2Vector (0.80), Glove (0.76) and FastText (0.80). The main reason is likely due to the pre-training process and the multi-head self-attention mechanism; words consisting of multiple nucleotides contain more information about the word's position and the connection between each nucleotide and its context. We then evaluated the performance of the dynamic embedding method and static embedding methods on the dynamic prediction task using HDRNet. First, to predict the RBP binding sites in K562 cells using HepG2 models, we observe in Fig. 5b that HDRNet using BERT as the feature descriptor performed the best of all the NLP methods, with an average AUC of 0.81, better than the other encoding schemes, One-Hot (0.71), W2V (0.75), D2V (0.74), Glove (0.75), and FastText (0.75). Indeed, we found that the static NLP methods do not provide better RNA-RBP binding recognition performance, also indicating that static coding schemes do not represent the contextual information of nucleotides. Second, we evaluated the performance of predicting binding sites in HepG2 cells using the K562-trained model. As indicated in Fig. 5b, BERT still achieved the highest performance (0.78) despite a decreased difference in results between BERT and static NLP methods. In addition, Fig. 5c visualizes the correlation heatmap of features generated by HDRNet using different NLP descriptors, where we observed that the features generated by HDRNet using BERT showed a stronger correlation than the other static descriptors, indicating that HDRNet effectively learns the contextual dependence of the RNA sequence via the dynamic contextual information generated by BERT. These evaluation results demonstrate the superior adaptability of the dynamic global contextual information generated by BERT, which contains rich context dependencies of nucleotide sequences in different cellular conditions, providing better adaptability in dynamic prediction tasks.

Then, to further investigate the advantage of adding in vivo secondary structure information in vivo, we conducted several experiments to dissect how RNA sequence and structural information contribute to the accurate prediction by HDRNet. Firstly, we evaluated the performance of HDRNet with or without the secondary structure information in static prediction tasks, respectively. As expected,

Fig. 5d shows that the HDRNet model adding secondary structure features outperformed the model without structural information for almost all RBP binding prediction, indicating that RBPs incorporate structural preferences for binding recognition, and that secondary structure features enrich the static representation from a biological perspective leading to improved prediction results. In addition, we evaluated the impact of secondary structure information on the performance of dynamic binding site predictions. As depicted in Fig. 5e, we observed a significant reduction in accurate dynamic prediction when only sequence features were used as input data in HDRNet. Moreover, we extended our evaluations to include other in vivo or computationally predicted secondary structure features, and evaluated the performance of HDRNet using various combinations of these different structural feature descriptor. As illustrated in Supplementary Note 2 and Supplementary Figs. 2 and 3, the original HDRNet consistently demonstrated optimal performance, indicating that the structure is relevant to RBP-RNA binding and the robustness of HDRNet.

## Evaluation of hyperparameter selection and ablation study

We explored the impact of several key parameters on HDRNet functioning. We first investigated the performance of different k-mer sizes in HDRNet, including 3bp to 6bp. As visualized in Fig. 6a, we note that the 3-mer model achieved the best AUC value of 0.875, surpassing the 4-mer, 5-mer, and 6-mer models, achieving 0.868, 0.862, and 0.869, respectively. We found surprisingly only a slight decrease in performance as the value of K increased, which is in contrast to the results of previous studies[49]. We speculate that the reason for this is the powerful feature learning capability of HDRNet, able to extract important information from dynamic embeddings at different levels. After the Hierarchical Multi-scale Residual Network and before DPRBP, we also tried a different way to combine features by adding sequence features and structural features together instead of stringing them together. The experimental results are summarized in Fig. 6b, we observe that the concatenated features outperformed the summed features, and we speculate that this is because summing up two features destroys the feature construction learned by the hierarchical network, which leads to a decrease in performance.

Then, to assess the contribution of each component in our proposed deep network architecture, we ablated each component of HDRNet as follows: (1) We first removed the Hierarchical Multi-scale Residual Network and DPRBP and replaced them with a fully connected network, called $HDRNet_{DUSO}$; (2) We employed only the DPRBP stage of HDRNet for prediction, called $HDRNet_{DPRBP}$; (3) We tested HDRNet without the Hierarchical Multi-scale Residual Network, called $HDRNet_{N_{MVRS}}$; (4) We replaced the max-pooling layer of DPRBP in HDRNet with an average-pooling layer, called $HDRNet_{AVGpool}$. As depicted in Fig. 6c, the performance of HDRNet outperformed all the ablated frameworks, with an optimal AUC value of 0.88. Indeed, with a fully connected network, $HDRNet_{DUSO}$ is unable to provide accurate predictions, indicating that DPRBP has significant feature extraction and prediction ability. Furthermore, using only DPRBP, the performance of $HDRNet_{DPRBP}$ decreased by 4%, demonstrating that multiple deep learning mechanisms are effective in learning and integrating the underlying features of sequence and structure information. Moreover, the proposed HDRNet is higher than $HDRNet_{N_{MVRS}}$ without the Hierarchical Multi-scale Residual Network (0.86). In depth, to intuitively visualize the feature maps of HDRNet and $HDRNet_{N_{MVRS}}$, the correlation heatmaps are depicted in Fig. 6d. We can clearly observe that there was a significant positive and negative correlation in the Hierarchical Multi-Scale Resnet output, allowing us to accurately identify RNA-RBP binding. Additionally, we highlighted the advantages of the hierarchical structure, as discussed in Supplementary Note 4 and depicted in Supplementary Fig. 6, demonstrating the necessity of the Hierarchical Multi-scale Residual Network.

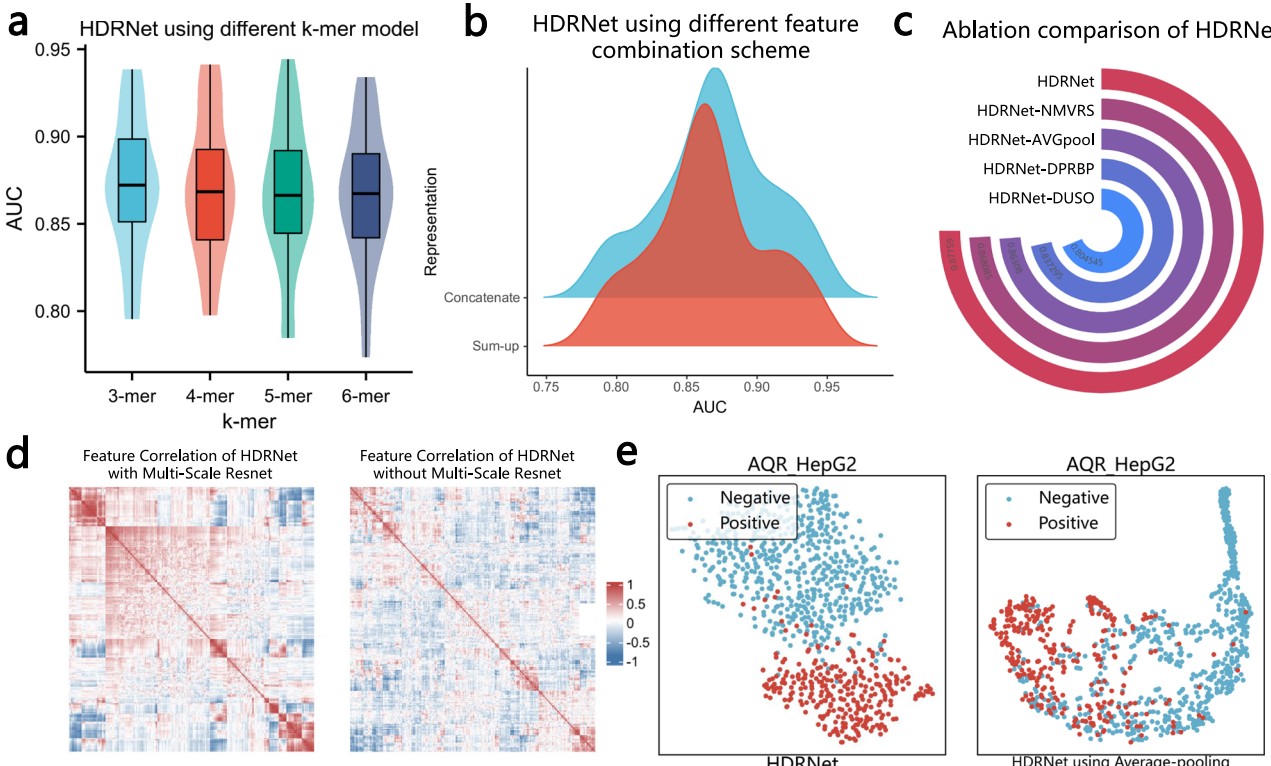

**Fig. 6 | Ablation study of HDRNet. a** The overall performance comparison of HDRNet using different k-mer representations of BERT ($n = 20$ in each group; center line, median; box limits, upper and lower quartiles; whiskers, $1.5 \times$ inter-quartile range). **b** The impact of different feature representations when concatenating and adding. **c** Performance comparison of the various deep network architectures of HDRNet. **d** Visualization of the HDRNet feature map with and without the Hierarchical Multi-scale Residual Network. **e** t-SNE clustering visualization of the output of DPRBP with max-pooling layer and average-pooling layer, respectively. Source data are provided as a Source Data file.

Furthermore, after each DPRBP block, the sequence length is halved using pooling layers that preserve the most important features. We conducted an experiment to investigate the impact of various pooling layers, employing either the maximum pooling layer or the average pooling layer. Figure 6c shows that the average AUC of HDRNet with the maximum pooling layer and the average pooling layer is 0.88 and 0.86, respectively. To further demonstrate the influence of the maximum pooling layer, we projected the embedding layer into two dimensions using t-SNE and annotated them with the true labels in order to display the features extracted by the two pooling layers of HDRNet as summarized in Fig. 6e. We can clearly see that the positive and negative samples were separated, and the clustering effect of HDRNet using the maximum pool was clearly better than that using the average pooling. A potential reason could be that the maximum pooling layer always selects the features of the most prominent tokens, resulting in the most prominent features being kept until the end, whereas the average pooling layer fuses the features of each token together, resulting in the loss of important information. In conclusion, the results indicate the validity and reasonableness of each HDRNet component.

### HDRNet's attenations reveal biologically meaningful interpretable learning patterns and motif inferences

The self-attentive mechanism can capture critical RNA sequence fragments, hence enhancing the ability to recognize motifs[50]. To identify motifs by our HDRNet, we put all of the RNA sequences through BERT's fine-tuning process and generated an attention vector for each sequence. Then, we used the attention vectors to find consecutive high-attention segments using sliding windows and aligned related subregions as the final binding motifs. Once all high-attention sequence fragments were recovered, their nucleotide occurrences

were counted and transformed into position weight matrices (PWMs). As visualized in Fig. 7a, a total of 172 motifs were found that matched the known RNA-binding motifs of RBPs. Interestingly, RBPs involved in the same RNA regulatory pathways were generally grouped together via hierarchical clustering. We then utilized the TOMTOM tool[51] in the MEME Suite to match the authenticated motifs in the ATtRACT database with the motifs discovered by the transformers. As can be seen in Supplementary Table 1, the extracted motifs from the transformers are very comparable to known motifs. We also showed that the dynamic contextual embedding takes into account the location of global words and supports out-of-vocabulary words by implementing a multi-headed self-focus mechanism, thus flexibly adapting to multiple cases of motif pattern extraction.

Moreover, the SHAP tool[47] was employed to extract high-attention dynamic semantic information. We found unexpectedly that HDRNet captured the binding peaks associated with the extracted motifs and identified the structural preferences of the binding events. Taking the input sequence of RBP TIA1 in Hela cells as an example, Supplementary Fig. 13 reflects the impact of each token of the input dynamic contextual feature and the icSHAPE structural information, where HDRNet successfully captured the poly-U binding motifs with structural preference of single-stranded, which is in line with the confirmations of earlier investigations[32] (Supplementary Note 10). In addition, HDRNet can successfully identify specific binding events in different cellular conditions. The saliency maps of the dynamic prediction of RBP LIN28B are presented in Fig. 7b, where the top strip plots the potential binding motifs; the second strip is the heatmap of the sequence attention scores; the third strip indicates the specific sequences; and the bottom strip contains the icSHAPE scores represented as a line plot and structural attention heat-map. From this we understand that HDRNet was capable of identifying dynamic binding

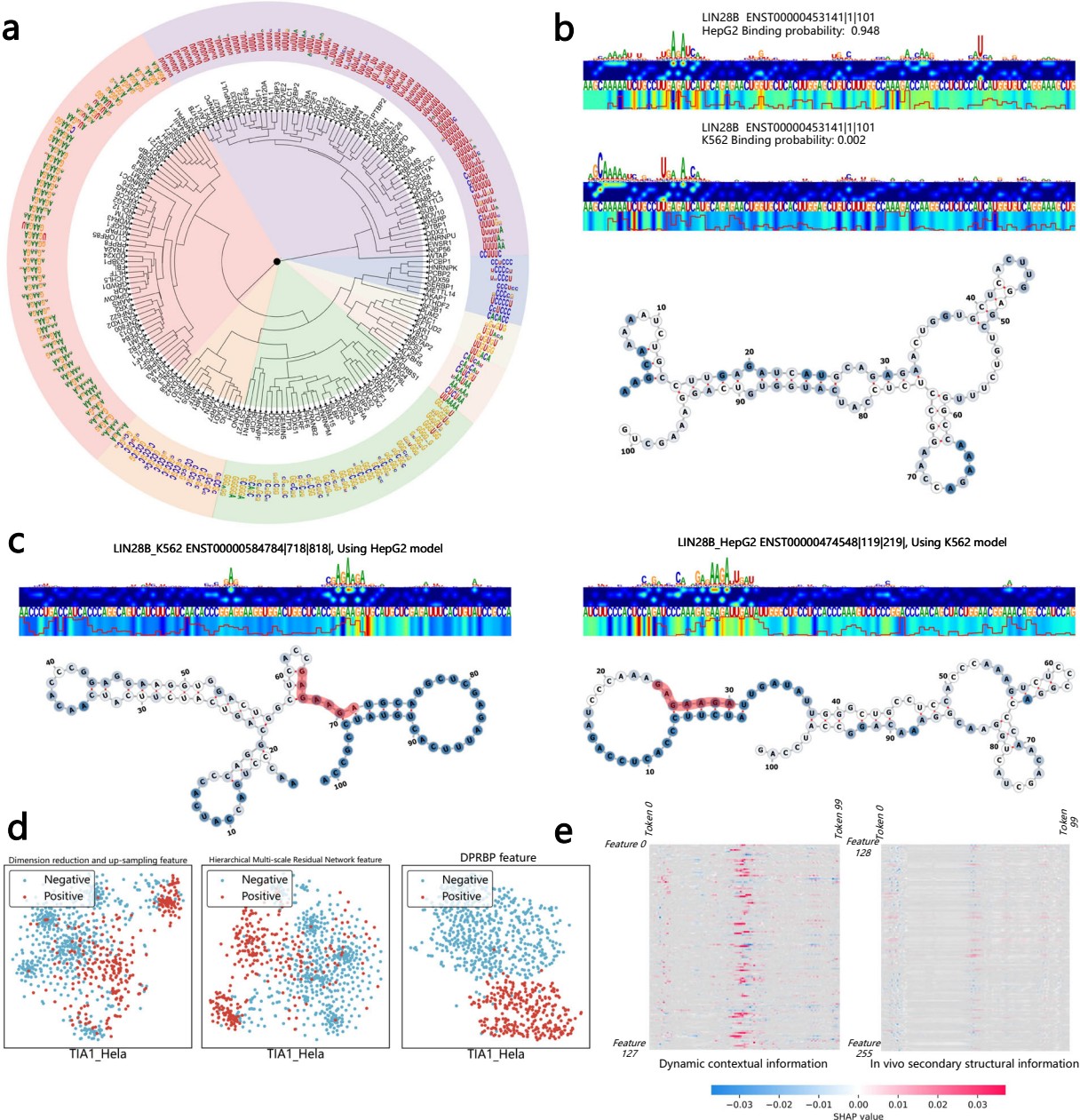

**Fig. 7 | Interpretation study of HDRNet. a** Circos plot of the extracted binding motifs of all 172 RBP datasets. RBPs involved in the same RNA regulatory pathway are generally grouped together via hierarchical clustering. **b** LIN28B binding probabilities and saliency maps in HepG2 and K562 cells and the visualization of the RNA structure. **c** Saliency maps of RBP LIN28B under dynamic prediction tasks, where the dynamic binding motif regions are successfully identified by HDRNet and highlighted. **d** t-SNE clustering results of the output feature map HDRNet. It can be clearly observed that the positive and negative clusters become distinct as HDRNet processes. **e** The attention distribution visualization measured by the SHAP tool, where the potential binding peak is highlighted. Source data are provided as a Source Data file.

events of the same transcript in different cells, as depicted in Fig. 7b, showing that the binding probability of the local region is 0.948 in HepG2 cells and 0.002 in K562 cells, respectively, indicating the possible existence of diverse gene expression in the different cells. Then, as shown in Fig. 7c, we used the HepG2 model to scan the K562 data, and we observed that the LIN28B binding fragment with high attention was highlighted by HDRNet (GAGAAGA). Similarly, we used the K562 model to scan HepG2 data and we obtained the same binding peak as with the K562 data. Moreover, we noticed that LIN28B RBP shares the same structural binding preference (single-stranded) in both K562 and HepG2 cells, since the icSHAPE score of the obtained binding peaks regions was >0.8. These results demonstrate the potential of HDRNet for biological interpretability tasks. We also evaluated the advantages

of adopting the dynamic global contextual embedding, by exploring the distribution of attention weights in BERT from HDRNet. As shown in Supplementary Note 11 and Supplementary Fig. 14, we see that the selected tokens did not decay noticeably with increasing distance, indicating that BERT successfully learned and preserved the long-distance dependencies and short-distance context of the sequence.

In addition, we explored the contribution of the refined in-vivo biological features and the dynamic contextual information for RNA-RBP binding event identification. The analyses results are depicted in Fig. 7d, e. First, we extracted the output feature matrix of each phase of HDRNet during the training process and projected it onto a two-dimensional space using t-SNE to better explain the learning process of HDRNet. As shown in Fig. 7d, the first subplot represents the t-SNE

clustering results of the output features of the dimensionality reduction and up-sampling network, where the prototype of each cluster appeared; the second subplot displays the t-SNE results of the output features of the multi-scale ResNet, where the features have a regular distribution. The last subplot reveals the t-SNE results after processing by DPCNN, where we can clearly observe a distinct clustering result, demonstrating the robustness of the HDRNet architecture. Figure 7e depicts the impact of dynamic contextual information of RNA sequence and icSHAPE structural features on identification of RNA-RBP binding events after the learning of the deep neural network in HDRNet, where the higher SHAP values denote that the particular feature plays a greater role in the final prediction decision, namely the high-attention region. From Fig. 7e, it is evident that the regions of high attention for both dynamic contextual information and secondary structure information were located in the same continuous tokens, forming a binding peak that may represent the final binding point. In addition, we observed that the dynamic contextual information was given greater weight with higher SHAP values than structural information, indicating that RNA sequence information is more relevant than RNA secondary structure information for predicting RNA-RBP interaction events. On this basis, we confirmed the effectiveness of the suggested deep network architecture and the possible biological interpretability of HDRNet.

### Identification and visualization of TDP-43 binding patterns under potential genetic variants

Genetic variants (GVs), primarily including Single-Nucleotide Polymorphisms (SNPs) and Single-Nucleotide Variations (SNVs), are permanent changes in the nucleotides of DNA sequence that makes up a gene. The alternative alleles of a GV may confer different binding specificity to an RBP since each RBP has its own sequence specificity, implying that GVs may disrupt the identification of RNA substrates by the RBP, resulting in allele-specific functional consequences and leading to severe diseases[52].

TARDBP, or TDP-43 (TAR DNA binding protein 43) protein that binds to DNA and RNA, plays an important function in intracellular RNA transcription, selective shearing, and regulation of mRNA stability[53]. Many studies have demonstrated that TDP-43 is associated with cancers[54] and severe neurological disorders such as epilepsy, amyotrophic lateral sclerosis (ALS), frontotemporal dementia (FTLD) and Alzheimer's disease (AD)[55–58]. To further elaborate the relationship between TDP-43 binding properties and disease-related genetic variants, we first used HDRNet to scan the TDP-43 dataset and obtained a total of 10589 TDP-43 binding sites (5231 binding sites in K562 cells and 5298 binding sites in HEK293 cells, respectively). By comparing with the dbSNP[59] and COSMIC databases[60], we found that the TDP-43 binding sites were enriched with a substantial number of SNP and SNV mutant alleles, especially in the high-attention regions identified by HDRNet. To elucidate the correlation between diseases and potential genetic variants identified by HDRNet, we conducted experiments to observe binding behavior after applying a transformation to the variant alleles. As shown in Fig. 8 and Supplementary Fig. 15, we found a decreasing trend of TDP-43 binding to RNA in most of the transcripts, where alteration in the variant resulted in significant changes of binding events in the high-attention region; for example, in transcript *ENST00000547986* from K562 cells shown in Fig. 8a, a mutation of the poly-U binding peak at position 94,971,529 on chromosome 12 resulted in a significant reduction of binding in this local region, leading to a risk of breast and lung tumor; in transcript *ENST00000533549* of K562 cells shown in Fig. 8c, the SNV mutation associated with large intestine tumor also affects the TDP-43 binding event in the adjacent region, suggesting that this genetic variant may affect gene regulation and thus lead to possible disease pathogenesis. Notably, we observed that nucleotide mutations lead to potential RNA structural changes, as demonstrated in Supplementary Note 13 and Supplementary Figs. 16-

17. In addition, as depicted in Fig. 8f in transcript *ENST00000533549* of HEK293 cells, intron variant located at *chr10:100517088* also reduced the binding tendency and is also associated with large intestine tumor.

Notably, we show that the dynamic binding events of Fig. 8c, f located in the *NDUFB8* transcript were highly associated with the large intestine cancer, indicating that the disruption of such dynamic RNA-RBP interactions are potential links with carcinogenesis. In summary, we can conclude that these analyses validate the biological interpretability of HDRNet to identify potential genetic variants and reveal underlying associations of the RBP binding sites with human diseases.

### Characterizing human disease-associated RNA and protein interactors with TDP-43 from a transcriptomic angle

After examining the latent relationship between RBP binding sites and human disease, we continued to study the link between RBP dynamic binding and human disease with a view of functional genomic and to find possible medical interventions from a transcriptomic angle. After obtaining TDP-43 binding sites, we mapped the binding transcript IDs to gene symbols and defined these genes as TDP-43-binding genes. In particular, the shared mapped binding sites were defined as dynamic binding genes. Firstly, we examined the interrelationships between the TDP-43 binding sites in K562 and HEK293 cells. Among the identified binding sites, 1063 binding genes (BGs) for K562 cells and 1894 BGs for HEK293 cells were recognized. The Venn diagram[61] in Fig. 9a shows that the two cell lines shared 544 TDP-43 BGs. Then, we used Enrichr[62] to perform a gene ontology enrichment analysis on the 544 TDP-43 binding genes of K562 and HEK293 cells. This helped us to understand what they might mean biologically and which pathways were enriched. Then, we used the GO database as an annotation source to annotate three types of gene ontology analysis: biological processes, molecular function, and cellular components. The top 10 terms of these three categories based on p-value are depicted in Fig. 9b. For the biological processes, cytoplasmic translation (41 genes) and gene expression (68 genes) were among the top GO terms. Indeed, it has been disclosed that in ALS motor neurons, especially neurons with mislocalized TDP-43, the amount of TARDBP mRNA is increased in the cytoplasm[63], and is involved in other cellular processes such as microRNA biogenesis, apoptosis, and cell division[64]. In the molecular function experiment, we saw RNA binding (220 genes) and mRNA binding (55 genes) were the two top GO terms. It is known that RNA binding proteins (RBPs) are highly associated with neurological diseases[65], corroborating the crucial need for studying and predicting RNA-RBP interactions. The nucleus (265 genes) and intracellular membrane-bounded organelles (283 genes) were the top GO terms for cellular components. Previous research has demonstrated a significant loss of neurons within the lateral part of the pedunculopontine nucleus in individuals with idiopathic Parkinson's disease and in individuals with combined Parkinson's and Alzheimer's diseases[66], revealing the association between the nucleus and the neurological disease. In addition, variations in intracellular membrane-bounded organelles are likewise a potential cause of neurological disorders[67,68].

Further, we performed KEGG pathway analysis on the 544 shared TARDBP binding genes between K562 cells and HEK293 cells, revealing that the organism is capable of reacting to inherent modifications. KEGG pathway analysis can demonstrate the interaction between various diseases through basic molecular or biological processes[69]. The most significant pathways of the genes in K562 cells and HEK293 cells of the TARDBP binding sites evaluated by p-value are summarized in Fig. 9b. Among the top 10 KEGG human pathways listed, we observe that they were highly associated with neurological diseases, including Parkinson's disease (PKD)[70], Amyotrophic lateral sclerosis (ALS)[56], Huntington's Disease (HD)[71], Prion Disease (PD)[72], and Alzheimer's disease (AD)[73].

In another context, we fed the TARDBP binding genes into the STRING[74] to build a protein–protein interaction (PPI) network to

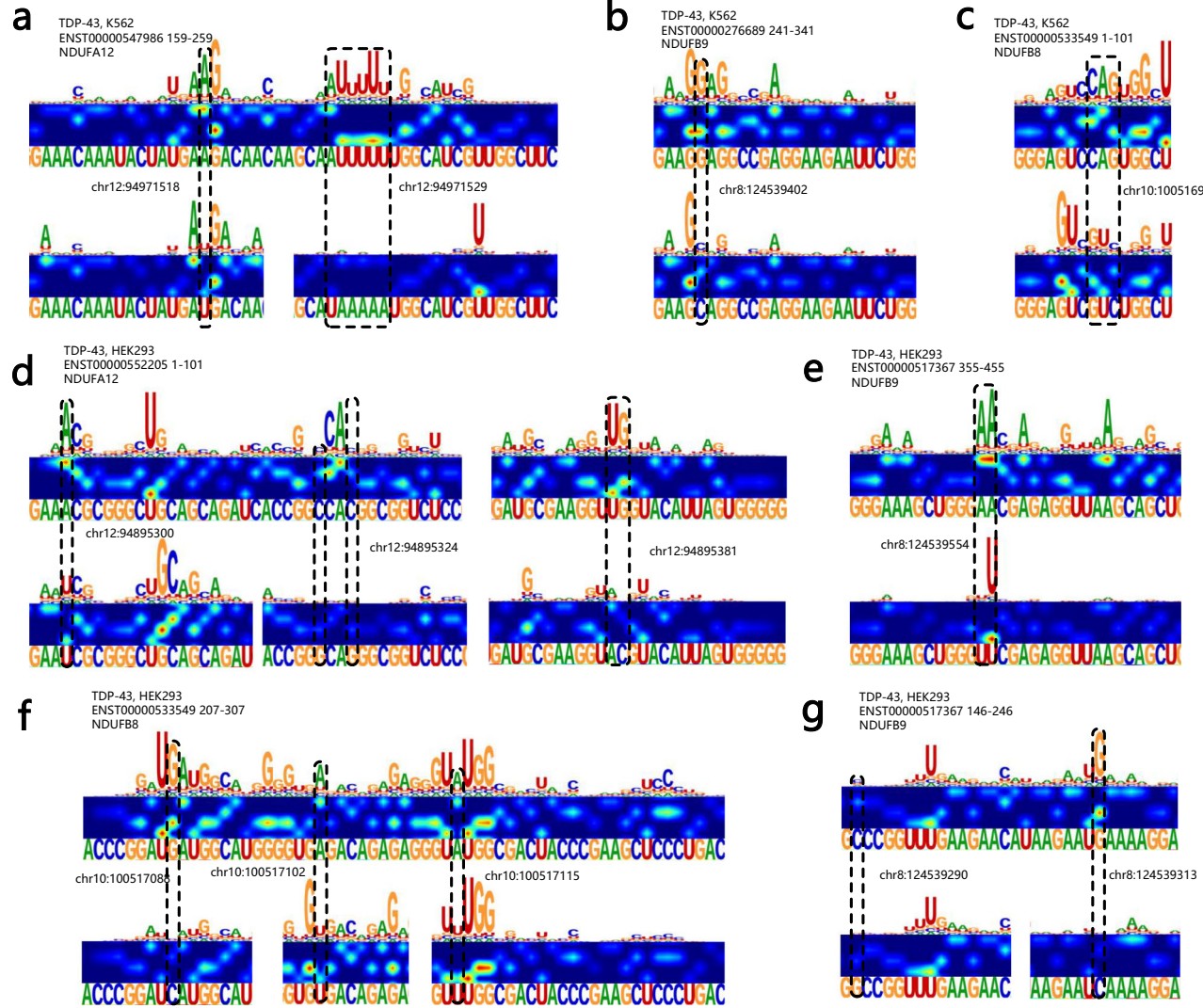

**Fig. 8 | Identification and visualization of potential disease-causing genomic variants. a** Disruption of the TDP-43 binding site in the coding region of *NDUFA12* contributes to the risk of breast and lung tumors, respectively. **b** An SNP variant in the coding region of *NDUFB9* leading to reduced binding affects gene pleiotropy. **c** An SNV variant that alters the coding sequence affects binding events in a local region and plays a role in colorectal tumors. **d** Weakening of the 3'UTR variant at the *NDUFA12* binding site in HEK293 cells. **e** Potential variant affecting the *NDUFB9* binding events. **f** Transcript variants occurring in introns, where the SNV variant located in *chr10:100517115* is associated with large intestine tumors. **g** Variants of the 5' UTR that are associated with large intestine tumors. Source data are provided as a Source Data file.

visualize the interactions and adhesion pathway. As depicted in Supplementary Fig. 18 and Supplementary Note 14, a total of 537 nodes and 1851 edges were obtained, where neurological disease-related genes are highlighted (PKD - red, ALS - blue, HD - green, PD - yellow, AD - pink, a total of 68 genes, see Supplementary Figs. 18 and 19), and we observed that these genes are highly correlated and clustered. Figure 9c depicts the most significant disease-related genes, including a total of 28 genes. Then, the Molecular Complex Detection (Mcode) plugin[75] in Cytoscape was adopted to identify the most important modules, as shown in Fig. 9d. Further, the top three hub genes were selected by the Cytohubba plugin in Cytoscape[76] using the MCC method[76]. As shown in Fig. 9e, the top three hub genes were *NDUFA12, NDUFB9 and NDUFB8*, and mutations in these genes are new causes of complex I deficiency[77–79], which is known to be associated with Parkinson's disease[80]. Moreover, focusing on these top hub TARDBP binding genes, we performed the pan-cancer analyses to characterize the differential expression of these genes in various cancers using TIMER[81] (Supplementary Note 15). As depicted in Supplementary Fig. 20, we observe that the human disease-related TDP-43 binding genes were also significantly differentially expressed between tumors

and normal tissues in human cancers. Notably, some of the cancers identified (Breast Cancer (BRCA), Lung Squamous Cell Carcinoma (LUSC), Lung Adenocarcinoma (LUAD) and Colon adenocarcinoma (COAD)) were consistent with those we found in RBP binding site mutations, thus corroborating the potential association of RBPs with cancer. In addition, Supplementary Fig. 21 in Supplementary Note 16 depicts the hub gene-disease association network, where we found a possible link between liver tumors and subunits of NADH dehydrogenase. Supplementary Figs. 22–23 in Supplementary Note 17 showed the interaction network of TFs and miRNAs and the disease-related genes, and we indeed observed diseases related to the nervous system. Based on the identified disease-linked binding genes, we finally investigated possible drug molecules for treatment of neurological disorders, as tabulated in Supplementary Table 2 and Supplementary Note 18.

## Discussion

Cross-linked immunoprecipitation sequencing technology enables the high-throughput measurements of RNA-binding protein (RBP) binding patterns at the transcript level, accounting for the dynamic cellular

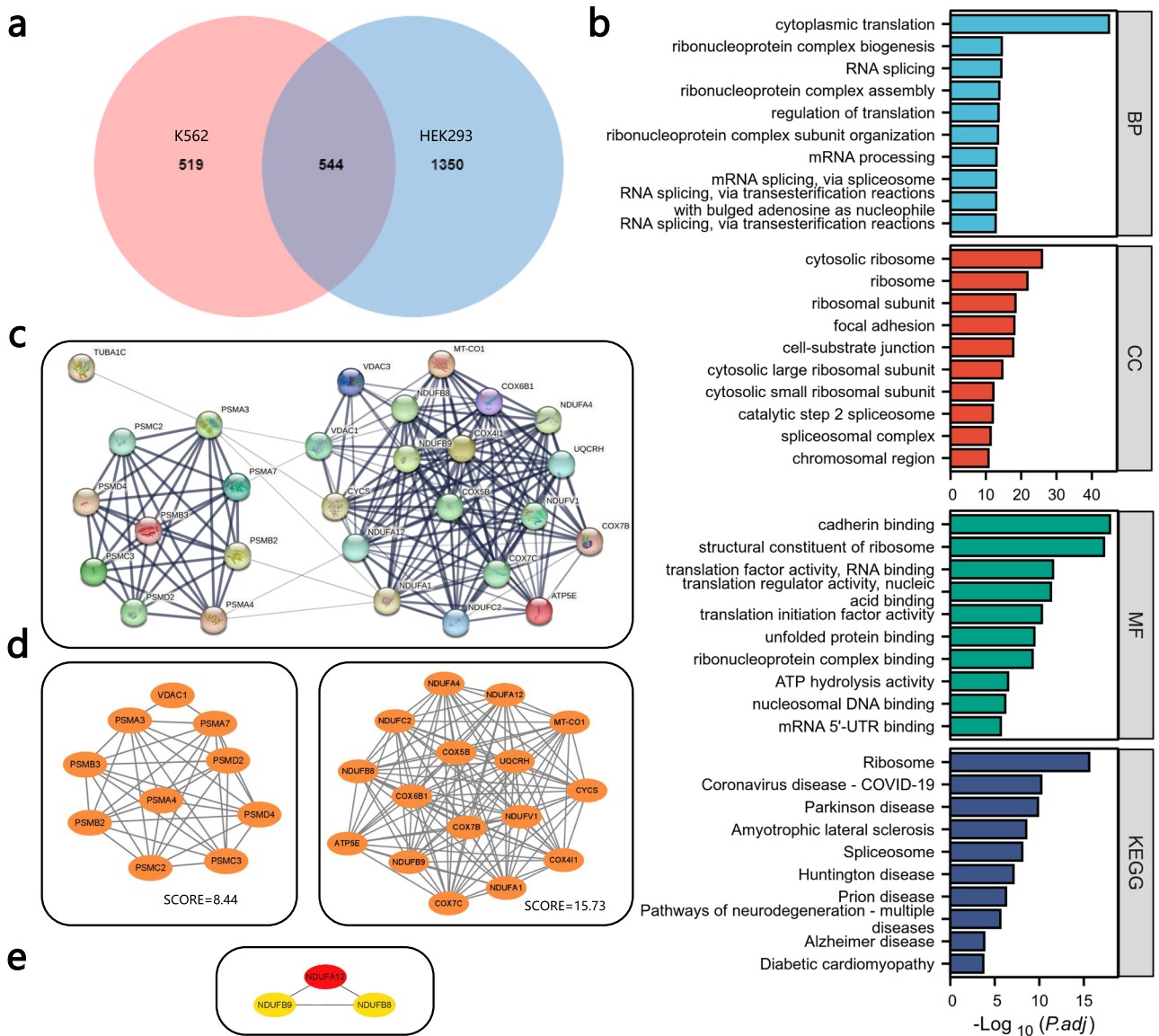

**Fig. 9 | Transcriptomic analysis of human disease-associated RNAs and proteins interacting with TDP-43. a** TARDBP binding site datasets from K562 cells and HEK293 are integrated. The integrated analysis revealed 644 shared TARDBP binding genes between the two cell lines. **b** Bar graph representing the biological process, molecular function, cellular component and KEGG pathway enrichment ontological analysis of shared TARDBP binding genes (two-sided Wilcoxon test and adopt BH to adjust p-values for multiple comparisons). **c** The protein-protein interaction network of the TDP-43 binding genes for neurological diseases localization. **d** PPI networks in MCODE analysis. **e** The top three hub genes in CytoHubba analysis. Source data are provided as a Source Data file.

conditions. However, current computational methods have limitations in considering the diversity of cellular conditions, which poses a significant challenge for predicting the cross-prediction events between RNAs and proteins in different cells. Here, we proposed an end-to-end deep learning-based framework to precisely predict dynamic RBP binding events across diverse cellular conditions. Specifically, we adopted multi-source biological information including the dynamic global contextual embedding and the in-vivo RNA secondary structure profile to characterize both the sequence and structural properties of RNA. Then, a unified alignment of multi-source feature representation was employed to generate potential feature maps with dimensional homogeneity for both biological representation. After that, we proposed a hierarchical multi-scale residual network to comprehend the contextual dependencies between the nucleotides and their structures at varying distances. Finally, a deep protein-RNA binding predictor with a multi-layer feature selector module was developed to learn and select the underlying key nucleotides tokens and employed a sigmoid function to forecast the binding of the proteins to RNA sequences.

We compared HDRNet with five RBP prediction methods in static cellular protein-RNA interaction identification. The experimental results showed that HDRNet had the best prediction performance, outperforming all compared methods on most benchmark datasets. In particular, HDRNet was able to efficiently learn implicit representations of RNA sequences and structural information. Meanwhile, HDRNet brought new insights by characterizing RNA binding events between different cell lines in a dynamic manner. Specifically, we trained HDRNet on the RBP dataset of K562 or HepG2 cells, and then predicted the RBP binding sites in the other cell line as an independent test set. We compared the dynamic prediction performance of HDRNet with the 5 benchmark methods, and found that HDRNet outperformed other methods on both K562 and HepG2 cells. HDRNet successfully predicted all binding sites of RBP HNRNPA1 on the MT-ND5 transcript with no false positives. Moreover, HDRNet pioneeringly accomplished cross-tissue dynamic prediction tasks and successfully highlighted the significant binding regions, illustrating the robustness of HDRNet in dynamic prediction.

In addition, HDRNet can also elucidate the underlying binding motifs from the dynamic global contextual embedding. We obtained a total of 172 motifs and discovered that RBPs involved in the RNA regulatory pathways were generally grouped together via hierarchical clustering. In addition, we found that HDRNet can detect the specific binding-peak by capturing the attention of the input data, thus enabling the identification of specific binding events in different cellular conditions. Then, through comparison with the genomic variants dataset, we identified that HDRNet captured high attention regions enriched with mutable alleles, and that alterations in these variants had significant changes in binding events in these high attention regions, revealing a potential link between RBP binding sites and human diseases. We also investigated the interrelationships between RBP binding sites and human diseases from a transcriptomic perspective by mapping the binding sites onto gene symbols and performing a series of genomic analyses, and provided possible drug molecules for disease treatment.

In summary, HDRNet discards the traditional RNA sequence representation and uses multi-source biological information to characterize the binding patterns of RNA binding proteins. As a new deep-learning method, HDRNet can simultaneously perform both static and dynamic RBP binding prediction, potential feature extraction, binding-peak identification, and interpretability analyses.

## Methods

### Data processing

For each RBP dataset, the resulting peaks were defined as binding sites, and the top 5000 binding sites with the most confident peaks were reserved for training and test sets as positive samples[32]. Among these, the length of each binding site was fixed at 101 nt while a region shorter than 101 nt was extended from the middle to both sides and a region longer than 101 nt was cut off from both sides. In addition, 10000 negative samples were generated by randomly selecting 101 nt from the whole transcriptome. Then, we labeled the positive samples as '1' and negative samples as '0'. For each RBP dataset, we constructed a HDRNet model for each cell line separately. During the training of each RBP dataset, 20% of the samples were randomly selected as an independent test set to evaluate the performance of the model. Among the remaining 80% of the training samples, 20% of the training samples were randomly selected as the validated set to optimize the model parameters, while the rest of the data was considered as the training set. For convenience, we have provided all datasets and the BERT model on the website https://doi.org/10.6084/m9.figshare.24132423.

### In-vivo RNA secondary structure representation

In general, the function of RNA is closely related to its complex folded secondary and tertiary structures. In addition, RNA secondary structure motifs are essential for the regulation of several biological procedures, such as protein binding, subcellular localization, and RNA decay[82]. Therefore, it is essential to effectively identify RNA secondary structure features for precise RBP binding prediction. Although computational methods of structural analysis can also yield predictions of RNA structure from sequences with a certain degree of accuracy, the predicted RNA structures do not reflect the real situation under dynamic cellular conditions, which greatly limits dynamic RBP prediction. In vivo click selective 29-hydroxyl acylation and profiling experiment icSHAPE[83] pipeline allows for the generation of RNA structure profiles, providing insights into the dynamic nature of RNA structure across the entire transcriptome from in vivo experiments, enabling the global and accurate characterization of the relationship between RNA structure and RNA binding protein (RBP) interactions[84,85]. In particular, icSHAPE is a chemical modification-based approach, which is known for providing the natural reflection on RNA folding.[82] Lastly, another key factor is the

widespread acceptance and usage within the research community of icSHAPE[84,85], and this tool has garnered significant support and is backed by high-quality data. Thus, in our study, we adopted the icSHAPE technique to characterize the in vivo RNA secondary structure in different cellular conditions.

Briefly, the icSHAPE structure score $R$ is calculated based on reverse transcription (RT) counts and polymeric sequencing coverage (base density, BD) on individual bases, where each mapped read contributes an RT value for the first base upstream of the starting point of the mapping and a BD value for all bases covered. For each sequence, the window of size $wSize$ is initialized in the 5' direction and slid to the 3' direction with window step $wStep$. Indeed, the RT and BD between two replicates of the DMSO library and the NAI library of each base are combined by direct addition as follows:

$$r_i^C = r_i^{C1} + r_i^{C2} \tag{1}$$

$$r_i^T = r_i^{T1} + r_i^{T2} \tag{2}$$

$$b_i^C = b_i^{C1} + b_i^{C2} \tag{3}$$

$$1 \le i \le wSize \tag{4}$$

where $r_i^C$ is the RT of the DMSO library, $r_i^T$ is the RT of the NAI library, and $b_i^C$ is the BD of the DMSO library. Then, these values are normalized by dividing by the average of the previous 90% to 95% values:

$$r_i^C = r_i^C / r_{q95}^C \tag{5}$$

$$r_i^T = r_i^T / r_{q95}^T \tag{6}$$

$$b_i^C = b_i^C / b_{q95}^C \tag{7}$$

$$1 \le i \le wSize; \tag{8}$$

where $r_{q95}^C$, $r_{q95}^T$, and $b_{q95}^C$ are the calculated normalization factors for each replicate. On this basis, the enrichment signal is computed as follows:

$$e_i = \frac{r_i^T - \alpha \times r_i^C}{b_i^C} \tag{9}$$

where $\alpha$ is the subtraction factor to measure the effect of background noise from the DMSO sample in the signal of the NAI-N3 sample. Finally, the resulting icSHAPE score is normalized to [0, 1] as below:

$$R_i = \begin{cases} \min\left(1, \max\left(0, \frac{e_i - e_{q5}}{e_{q95} - e_{q5}}\right)\right), & b_i^C \ge 200 \\ NULL, & b_i^C < 200 \end{cases} \tag{10}$$

where $e_{q5}$ is the bottom 5% signal value and $e_{q95}$ is the top 95% value. In the final icSHAPE score, it is noteworthy that only nucleotides with more than 200x coverage are considered as effective quality control. Accordingly, we process the RNA sequences using icSHAPE, and each RNA fragment generates an equal-length vector of real numbers representing secondary structural features as the input of the identification of RNA binding events in various cellular conditions.

### Dynamic global contextual embedding

To characterize the sequence information of RNA, traditional one-hot encoding methods for RNA sequence information characterization

focus solely on the sequential alignment of nucleotides, neglecting the biological properties and contextual information of RNA sequences, thus restricting the performance of the model. We thus adopted the Bidirectional Encoder Representations from Transformers (BERT) model[43], a self-attention-based language representation model, within the HDRNet framework to generate the dynamic global contextual information for RNA sequences in diverse cellular conditions. Benefitting from the multi-head self-attention mechanism, BERT has been widely used for capturing the long-range dependencies between tokens in sequential data (discussed in Supplementary Note 19). Inspired by ref. 50, we first converted all the input data into *k*-mer tokens. For instance, given an RNA sequence 'ACGUGA', we can obtain fragments {ACG, CGU, GUG, UGA} after 3-mer processing. Then, we added two special tokens, [CLS] and [SEP], denoting the beginning and end of the sequence respectively, to each RNA sequence. Following this, all the input tokens were embedded into dynamic feature vectors of 768 dimensions, resulting in the construction of a feature matrix *X* for each sequence. To generate the weighted sum of the feature vectors across all tokens, we utilized a multi-head self-attention module. This module employs a multi-headed attention mechanism to capture the dependencies between tokens, resulting in a weighted sum of feature vectors as follows:

$$\textbf{Output} = \text{MultiHead}\,(\mathbf{Q}, \mathbf{K}, \mathbf{V}) = \text{Concat}\,(\textbf{head}_1, ..., \textbf{head}_h)\mathbf{W}^O \quad (11)$$

$$\textbf{head}_i = \text{Attention}\,(\mathbf{Q}, \mathbf{K}, \mathbf{V}) \quad (12)$$

$$\text{Attention}\,(\mathbf{Q}, \mathbf{K}, \mathbf{V}) = \text{softmax}\left(\frac{\mathbf{Q}\mathbf{K}^T}{\sqrt{d_k}}\right) \cdot \mathbf{V} \quad (13)$$

$$\begin{cases} \mathbf{Q} = \mathbf{X} \cdot \mathbf{W}_i^Q \\ \mathbf{K} = \mathbf{X} \cdot \mathbf{W}_i^K \\ \mathbf{V} = \mathbf{X} \cdot \mathbf{W}_i^V \end{cases} \quad (14)$$

where $\mathbf{Q}, \mathbf{K}, \mathbf{V}$ denote Query, Key and Value, respectively, which are projected through *h* diverse transformers encoder. $\sqrt{d_k}$ stands for the scaling factor to control the magnitude of the dot product. $\{\mathbf{W}_i^Q, \mathbf{W}_i^K, \mathbf{W}_i^V\}_{i=0}^h$ are trainable parameters matrices of Query, Key, and Value vectors of the *i*-th **head**, respectively, and $\mathbf{W}^O$ is the learnt weight matrix of the final linear projection of multi-heads. After *L* such transformation layers, the model is capable of learning rich global semantics and encoding tokens into dynamic embedding representations. On this basis, we extracted the hidden states from the final layer of the model as numerical representations and removed the special markers [CLS] and [SEP] that are added at the beginning and end of the sequence. Additionally, the BERT architecture used in our work has 12 transition layers, each of which has 768 hidden units and 12 self-attention heads, and the weights of the model were obtained from ref. 50. This attention mechanism enables the model to dynamically focus on the crucial nucleotide in the sequences under different cellular conditions and capture the contributions of features at individual tokens to facilitate the final prediction.

### Hierarchical deep neural networks

After obtaining multi-source biological characteristics of RNA sequences, an end-to-end hierarchical deep neural network was designed to simultaneously learn feature representation and identification via explicit modeling of those RBP binding data, as depicted in Fig. 1.

### Unified alignment of multi-source feature representation. As expounded in the preceding section, we presented two distinct

methods for characterizing RNA sequences. The first approach involved in vivo secondary structure representation produced by icSHAPE, while the second approach utilized dynamic global contextual sequence embedding generated by BERT. However, as the distributions of the two feature descriptors differ, a unified alignment model was developed to extend and extract structural and sequence features respectively, which can effectively capture global features of the in vivo secondary structure representation and local features of dynamic global contextual sequence embedding.

Considering the structural information, we first embedded the RNA sequences of the RNA-RBP binding datasets into a one-dimensional vector of size *d*, which is defined as $\mathbf{S} = [s_1, s_2, s_3, s_4, \cdots, s_d]$, where $s_i$ denotes the *i*-th in-vivo RNA secondary structure score. We can clearly observe that the structural feature vector has a small number of channels, which limits the global structural information contained in the in-vivo RNA secondary structure representation. To address this limitation, we first fed such one-dimensional vector into a CNN block to enrich the global dependence of the structural information, as follows:

$$\tilde{\mathbf{S}} = F_C(\mathbf{S}) \quad (15)$$

$$F_C(\mathbf{x}) = \text{ReLU}(\text{BN}(\text{Conv}\,(\mathbf{x}))) \quad (16)$$

$$\text{Conv}\,(\mathbf{x}) = \begin{cases} \sum_{i=1}^{l} \mathbf{W}_{c,i} * \mathbf{x}_{n,i-p_l} & \text{if} \quad p_l < i < p_l + l \\ 0 & \text{else} \end{cases} \quad (17)$$

where $F_C$ is the customized sequential CNN block designed to detect the local sensitive regions of RBP binding. Conv(**x**) denotes a 1D-convolutional layer with a learnable convolutional kernel $\mathbf{W} \in R^d$ of *c* channels and a padding $p_l$ to ensure that the input and output sizes are identical. After that, to enhance the ability of the model to detect sensitive regions and to avoid overfitting, we add a batch normalization layer (BN)[86] after the convolution layer as follows:

$$\text{BN}\,(\mathbf{x}) = \gamma\left(\frac{x - E(\mathbf{x})}{\sqrt{\text{Var}\,(\mathbf{x}) + \varepsilon}}\right) + \beta \quad (18)$$

where $\gamma$ and $\beta$ are learnable parameters in the batch normalization layer. $E(\mathbf{x})$ and $\text{Var}(\mathbf{x})$ represent the mini-batch average and variance, respectively, and $\varepsilon$ is added for numerical stability. During the training process, the batch normalization layer continuously updates running estimates of the mean and variance, which are used for normalization during evaluation. These running estimates have a default momentum value of *m*=0.1. The current estimate $\hat{\mathbf{x}}$ is calculated as a combination of the normalized input **x**, the previous estimate $\hat{\mathbf{x}}_p$, and the momentum value *m* using the formula:

$$\mathbf{x} = (1 - m) \times \hat{\mathbf{x}}_p + m \times \hat{\mathbf{x}} \quad (19)$$

Then, to retain and enhance the learned RBP-sensitive regions and to prevent the gradient from vanishing, the Rectified Linear Unit activation function (ReLU)[87] was employed to efficiently transfer the gradient, thus accelerating the convergence of the model as follows:

$$\text{ReLU}\,(x) = \begin{cases} x & \text{if} \quad x > 0 \\ 0 & \text{else} \end{cases} \quad (20)$$

In line with the approach taken for structural features, we applied a similar procedure to the BERT features. However, unlike secondary structure features, BERT features possess a higher number of channels, which presents an issue of global information being overly rich, thus disregarding the local dependence of nucleotides. Therefore, for BERT

features, we also utilized CNN modules with varying numbers of channels to extract and enrich the local information of the BERT features.

**Hierarchical multi-scale residual network.** To capture the hierarchical relationships between the nucleotides and their structure, we developed a hierarchical multi-scale residual network to capture the contextual dependencies between the nucleotides and their structure in an exhaustive manner. In comparison with traditional conventional convolutional neural networks (CNNs), residual neural networks (ResNet) have been shown to improve information flow and prevent the vanishing gradient problem, which often leads to overfitting in deep neural networks[88] (also discussed in Supplementary Note 19). As mentioned previously, the Unified Alignment of Multi-source Feature Representation was utilized to obtain the unified multi-source biological features. To further investigate the potential representations of these multi-source biological features, we constructed a hierarchical multi-scale CNN network with multiple sequential CNN modules, each of which has a different scale, to capture the potential different distance dependencies, which is represented as follows:

$$\text{MultiScale}(\mathbf{x}) = \left| \tilde{O}_1(\mathbf{x}), \tilde{O}_2(\mathbf{x}), \tilde{O}_3(\mathbf{x}), \tilde{O}_4(\mathbf{x}) \right| \quad (21)$$

$$\tilde{O}_j(\mathbf{x}) = \prod_j \left( F_{C_j}(\mathbf{x}) \right) \quad (22)$$

$$F_{C_j}(\mathbf{x}) = F_j * \mathbf{x}_{i+p-1} = \text{ReLU}\left( BN\left( \mathbf{w}_j \cdot \mathbf{x}_{i+p-1} + b \right) \right) \quad (23)$$

$$j \in \{1, 2, 3, 4\} \quad (24)$$

where $|| \cdot ||$ denotes the concat operation, $\tilde{O}_j(\mathbf{x})$ denotes the output of each scale network, $F_C$ denotes a CNN block consisting of a one-dimensional convolutional layer, BN layer and ReLU activation, $\mathbf{w}$ is a learnable convolutional kernel, and $b$ is a bias term. $\prod_j$ denotes $j$ sequential CNN blocks in scale $\tilde{O}_j(\mathbf{x})$, each of which has the identical number of channels.

Through multi-scale learning, the model is able to efficiently learn the different contextual dependencies of structural features at different distances to capture their unique binding properties. After that, we employed a residual mechanism to fuse the uniformly aligned secondary structural or dynamic global contextual embedding features with their corresponding multiscale biological features to enhance their local information, which is formulated as follows:

$$\text{ResidualBlock}(\mathbf{x}) = \text{ReLU}(\mathbf{x} + \text{MultiScale}(\mathbf{x})) \quad (25)$$

**Deep protein-RNA binding predictor.** After applying the structural and sequence information to the hierarchical multi-scale residual network, we first fused the two enriched features by concatenating them to obtain a comprehensive feature matrix. However, the restriction on sequence length still makes it challenging to make precise predictions. To address this challenge, we developed the Deep protein-RNA binding predictor based on deep pyramid convolutional neural network[89], which is capable of capturing global dependence of long-distance nucleotides at the token-level resolution. In our deep protein-RNA binding predictor, we set the sequence length halved after each layer, which specifically carries out a max-pooling layer and produces a new internal representation of the RNA sequence by taking the token-wise maximum over three consecutive internal vectors, which shortens the sequence while maintaining all its information. Since the number of feature channels is fixed, the computational complexity is also halved after each layer as the sequence length is halved. Moreover, unlike the original DPCNN[89], in DPRBP we set up two

independent CNN blocks instead of sharing their parameters to dynamically learn global context dependencies. The specific representation is as follows:

$$t(\mathbf{x}) = F_{C_1}\left( F_{C_2}(\mathbf{x}) \right) \quad (26)$$

$$r(\mathbf{x}) = \text{ReLU}(x + t(\mathbf{x})) \quad (27)$$

where $t$ denotes two consecutive distinct CNN blocks $C_1$ and $C_2$, which are capable of dynamically learning the spatial relationships between sequences and structures. In addition, a residual connection $r$ was added to enhance the perception of DPRBP in the high-attention region of sequences and structures. Subsequently, a max-pooling layer of size 3 with a step size of 2 was utilized to identify the specific binding of RBPs from the token-level loci, as follows:

$$g(\mathbf{x}) = \text{MaxPool}(r(\mathbf{x})) \quad (28)$$

Through this max-pooling layer, the length of the sequence is halved in such a way that the most significant nucleotides in the sequence are retained. Simultaneously, since each CNN layer has the same number of channels, the corresponding computational complexity is also halved. After $\log_2(L)$ layer, the length of the sequence is finally reduced to 1, and a feature vector $\mathbf{h}$ is generated that is enriched with the structure-sequence space relationship and contains the global information of the whole sequence, denoted as follows:

$$\mathbf{h} = \text{ReLU}\left( \prod_{\log_2 l} \left( g_j(\mathbf{x}) \right) \right) \quad (29)$$

Afterwards, to predict the interaction of RBP with RNA, the obtained $\mathbf{h}$ will be fed into a fully connected neural network:

$$P = \sigma\left( \mathbf{W}_d \times \mathbf{h} \right) \quad (30)$$

$$\sigma(x) = \frac{1}{1 + e^{-x}} \quad (31)$$

where $d$ denotes the dimension of the vector $\mathbf{h}$ and $\sigma(\cdot)$ is the sigmoid activation function. Finally, the output $P$ is considered as the probability of whether it is an RBP binding site or not.

## Training of HDRNet

Generally, we trained an HDRNet model for identifying dynamic cellular RNA-binding events, including 261 RBP datasets for 172 human proteins. First, we used RNA sequences as the input data with BERT and icSHAPE secondary structures for multi-source feature representation. During the training process, we randomly selected 20% of the samples as the independent test set for performance evaluation, and the rest were considered as the training set, where 20% of the training set was randomly selected as the validation set. HDRNet was trained to learn the parameters that minimize the binary cross-entropy loss function:

$$\text{Loss}(\mathbf{P}, \mathbf{Y}) = -\frac{1}{N} \sum_{i=1}^{N} [y_i * \log p_i + (1 - y_i) * \log(1 - p_i)] + \lambda |W|_2 \quad (32)$$

where $N$ denotes the mini batch size, $y_i$ is the true label and $p_i$ is the predicted binding probability. $\lambda |W|_2$ denotes the $L_2$ norm on all parameters, which acts as the weight decay term to reduce overfitting in the training model. In addition, to demonstrate the performance of our model, the area under the receiver operating characteristic curve (AUC) is chosen for performance evaluation.

## Parameter settings

In the experimental section, we describe the implementation details of HDRNet. Sequence features obtained from BERT were first reduced in dimensionality using 128 one-dimensional convolutional kernels with a kernel size of 1. In addition, RNA secondary structure features were upsampled using 128 one-dimensional convolutional filters with a kernel size of 3. The resulting sequence and structural features were then separately fed into a Hierarchical multi-scale ResNet with identity maps for information enrichment, which contains one-dimensional convolutional layers with kernel sizes of 1, 3, 5 and 7 and CNN blocks of 1, 2, 3 and 4 in each view, totaling 32 channels. After concatenating the feature maps, they were passed through the deep protein-RNA binding predictor module, which had a kernel size of 5 and 256 channels, followed by a fully connected layer with one hidden unit and a Sigmoid activation function. All CNN layers were arranged in the order of Conv-BN-ReLU and had no bias. The model is trained on PyTorch 1.10 with parameters initialized using *Kaiming initialization*[90]. To prevent overfitting, dropout layers with a rate of 0.3 were added after each activation layer. The network was trained end-to-end using the Adam optimizer with a batch size of 32, a learning rate of 0.001, and a weight decay of 1e-6. In addition, we employed a linearly scaled warm-up scheme to calibrate the learning rate to overcome optimization challenges in the early training. Our model was trained on an NVIDIA GeForce RTX 3090 GPU with 24 GB of memory and early stops were controlled by validation loss, and taking ~80 h on all 261 RBP datasets.

## Competing methods

HDRNet was implemented in Python, and the core model was built on the PyTorch framework that is publicly available at https://github.com/zhuhr213/HDRNet. To elucidate the effectiveness of our proposed model, we compared HDRNet with several deep learning models and machine learning algorithms as follows:

- PrismNet[32] (https://github.com/kuixu/PrismNet) is a recent study that developed a convolutional neural network (CNN)-based deep learning approach, which effectively incorporates in vivo RNA structure data and RNA-binding protein (RBP) binding data to make precise predictions of RBP binding sites. This method applied an "attention" strategy to precisely identify RBP binding nucleotides. Notably, PrismNet is the first tool designed for dynamic prediction tasks.
- PRIESSTESS[30] (https://github.com/kaitlin309/PRIESSTESS) introduces a universal RNA motif-finding/scanning strategy capable of identifying enriched RNA sequences and/or structure motifs. PRIESSTESS consists of two steps. The first step generates a large collection of enriched motifs encompassing both RNA sequence and structure. The second step produces an aggregate model, which combines the motif scores into a single value, and gauges the relative importance of each motif.
- iDeep[28] (https://github.com/xypan1232/iDeep) is proposed and leverages a novel hybrid CNN network and deep belief network to predict the RBP interaction sites and motifs on RNAs by converting the original data into a high-level abstraction feature space using multiple layers of learning blocks, where the shared representations across different domains are integrated.
- DMSK[39] (https://github.com/Rebecca3150/DMSK) is a novel identification method based on multi-view deep learning, subspace learning, and a multi-view classifier for the identification of circRNA-RBP interaction sites involved in computationally predicted RNA secondary structures.
- GraphProt[29] (https://github.com/dmaticzka/GraphProt) models the sequence and structure binding preferences of RBP binding sites using graph kernel features based on sequence and computationally predicted structure information.

- DeepBind[26] (https://github.com/jisraeli/DeepBind) is a CNN-based deep learning model that predicts RBP binding sites based on only RNA sequences.

In addition, we conducted a comprehensive analysis of machine learning methods to compare performance to our proposed HDRNet. Specifically, we evaluated XGBoost[91], Random Forest (RF)[92], Logistic Regression (LR)[93], Artificial Neural Network (ANN)[94], ExtraTreeClassifier (ETC)[95], SGDClassifier (SGDC)[96], and GaussianNB (GNB)[97] using the same feature representation as HDRNet, including the dynamic contextual embedding and the icSHAPE secondary structure information. For the ANN implementation, we used a Pytorch environment and set the size of the hidden layers to 512, 32, and 1, respectively. XGBoost was implemented using the Python package of version 1.5.0. The remaining algorithms were implemented using the scikit-learn package[98]. Meanwhile, we applied the default parameter settings for Logistic Regression, Random Forest, and XGBoost.

## Tissue data pre-processing

The raw data for tissue dataset is stored in the '.bed' format, which records the genomic positions of each binding peak. We employed the 'bedtools getfasta' function or 'getFastaFromBed' to obtain the corresponding nucleotide sequences based on the chromosomal coordinates. Specifically, for human tissue data, we utilized hg38 as the reference genome, while for mouse data, we employed mm9 as reference.

## Motif calculation

The multi-head self-attention mechanism has the capacity to accurately identify and decode significant genomic regions. In our study, we leverage dynamic global contextual embedding to explore the biological functionalities of RNA sequences. More explicitly, we computed the attention score for the *i*-th embedded *k*-mer token, denoted as $Score_i$ through the summation, thereby identifying the transcript fragment in each RNA sequence that offers the most substantial contribution towards downstream classification, which can be formulated as follows:

$$Score_i = \sum_{n=1}^{N} \frac{\exp\left(\mathbf{Q}_{CLS}^T \cdot \mathbf{K}_i / \sqrt{d}\right)}{\sum_{t=1}^{T} \exp\left(\mathbf{Q}_{CLS}^T \cdot \mathbf{K}_t / \sqrt{d}\right)} \tag{33}$$

where $N$ symbolizes the number of attention heads, $T$ represents the number of tokens in a given RNA sequence, $\mathbf{Q}_{CLS}^T$ denotes the query vector of the special tokens [CLS], $\mathbf{K}_i$ denotes the key vector of the *i*-th *k*-mer token with $i \in \{1, 2, \ldots, L\}$ where $L$ denotes the number of input tokens, and $d$ is the dimensionality of the feature vectors. After computing the attention score for each token, we classified RNA fragments as RBP-binding motifs based on the following three criteria: (1) The attention score of the fragment exceeds the average score of the sequence; (2) The attention score of the fragment is 10 times higher than the lowest attention score; and (3) The minimum length of the fragment is 6.

## High attention region visualization

We adopted the SHAP tool[47] to calculate the specific attention score for each token of the input RNA sequence. The 'GradientExplainer' function was employed, where a score matrix of the same size as the input dynamic context matrix coule be obtained. Each numerical value in this matrix represents the attention value at the corresponding position in the input matrix. We then selected the maximum value of each token as the final attention score, and got an attention vector **Att** with length 99. Subsequently, we allocated the attention scores of each

token evenly across every nucleotide, as demonstrated below:

$$Score_i = \begin{cases} \mathbf{Att}_i, & i=1 \\ \frac{\mathbf{Att}_{i-1}+\mathbf{Att}_i}{2}, & i=2 \\ \frac{\mathbf{Att}_{i-1}+\mathbf{Att}_{i-2}}{2}, & i=100 \\ \mathbf{Att}_{i-1}, & i=101 \\ \frac{\mathbf{Att}_i+\mathbf{Att}_{i-1}+\mathbf{Att}_{i-2}}{3}, & else \end{cases} \quad (34)$$

where $Score_i$ denotes the attention score of the $i$-th nucleotide, and $i$ is in range [1, 101].

### Functional enrichment

We used the R package ClusterProfiler[99] to perform KEGG and GO enrichment analysis on the 544 TDP-43 binding genes. We utilized the function 'enrichKEGG' for the KEGG analysis, where the parameters were set to "pAdjustMethod = fdr, pvalueCutoff = 0.01, qvalueCutoff = 0.05". The GO analysis was performed by function 'enrichGO', where the parameters were set to "ont = ALL, pAdjustMethod = BH, pvalueCutoff = 0.01, qvalueCutoff = 0.05".

### PPI network

We performed preliminary PPI network construction on the 544 shared TDP-43 binding genes using STRING[74] and the most significant disease-related genes, including a total of 28 genes were input to Cytoscape for visualization. To identify the most important modules in the PPI network, we adopt MCODE[75] for the network and give the top three modules.

### Statistics and reproducibility

The detailed statistical tests were explained in each figure legend. Sample data were obtained from public repositories. Sample size was not predetermined and is the maximum number of samples available for each datasets. No data were excluded from the analyses. No experimental groups were assigned. Our study does not involve group allocation that requires blinding. To reproduce the results, please find the Source Data file we provided.

### Reporting summary

Further information on research design is available in the Nature Portfolio Reporting Summary linked to this article.

## Data availability

We collected 261 RBP binding sites datasets for cell lines across multiple databases, including 172 RBPs constructed using the same flag-marked technique in K562, HepG2, HEK293, HEK293T, HeLa and H9 cell lines. These datasets include 65 CLLP-seq datasets for 61 RBPs from POSTAR database[45] and 196 eCLIP datasets for 111 RBPs from the ENCODE project[100]. These data have been deposited in[32]. In terms of the RBP binding data in tissues, the processed MBNL2 (Muscleblind Like Splicing Regulator 2) binding peak data in human brain tissues[44] were available in POSTAR database[45] under accession code GSE68890 [https://www.ncbi.nlm.nih.gov/geo/query/acc.cgi?acc=GSE68890]; the DGCR8 and HNRNPU binding data in human adrenal gland were collected from ENCODE project[100]; and the MBNL1 (Muscleblind Like Splicing Regulator 1) binding data in brain, heart, muscle, and myoblasts from mice were obtained from[46] under accession code GSE39911 [https://www.ncbi.nlm.nih.gov/geo/query/acc.cgi?acc=GSE39911]. We have deposited these datasets in FigShare database [https://doi.org/10.6084/m9.figshare.24132423][101]. These datasets can also be downloaded from our HDRNet web-server at http://www.aibio-lab.com:5050/. Source data are provided with this paper.

## Code availability

HDRNet is an open-source tool available at https://github.com/zhuhr213/HDRNet[102], where all packages are implemented in Python.

We provide a user-friendly web server for the HDRNet method at http://www.aibio-lab.com:5050/, which enables users to determine whether a given RNA sequence is a binding site for an RNA-binding protein. Users can choose the precise RBP types and the cell lines by clicking on their corresponding buttons to complete the static and dynamic predictions based on their requirements. To facilitate use, users can enter the query RNA sequences in the input box or upload a text file containing RNA sequences in any format. The submitted jobs and predicted results, including the probability of the RNA sequence binding to the RBP, are then sent to the users' provided contact addresses. In addition, we provide all datasets along with the BERT model used in this study, which can be downloaded directly from the web server and FigShare[101].

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

## Acknowledgements

The work described in this paper was substantially supported by the National Natural Science Foundation of China under (Grant No. 62076109) and the Jilin Province Outstanding Young Scientist Program (Grant No. 20230508098RC), and also funded by "the Fundamental Research Funds for the Central Universities, JLU".

## Author contributions

X.L. conceived and supervised the project. H.Z. developed and implemented the algorithms under the guidance of X.L. and H.Z. wrote the manuscript. H.Z. conducted the experiments. X.L., H.Z., and K.W. did the biological interpretation. H.Z., Y.Y., and Y.W. completed the figures and manuscript. H.Z., X.L., and K.W. revised the manuscript. F.W., Y.H., and Y.C. provided advice on method development. All authors approved the manuscript.

## Competing interests

The authors declare no competing interests.
