## [Peer Review File · Nature Communications]

Dynamic characterization and interpretation for protein–RNA interactions across diverse cellular conditions using HDRNetReviewer #1 (Remarks to the Author):

Predicting RNA-RBP interactions in a dynamic manner across different cell lines is an important challenge for understanding the interactions between RNAs and RBPs. In this manuscript, the authors present a novel deep-learning method called HDRNet to address this problem. The method utilizes multi-source biological features, including dynamic global contextual information and in-vivo RNA secondary structure profiles. The authors compare HDRNet with existing tools in multiple RBP datasets and demonstrate its superior performance in both static and dynamic prediction tasks. Additionally, the authors perform motif analysis and high-attention binding region identification to provide biological interpretability. Furthermore, the authors conduct potential genetic variants analysis and genome-wide functional studies for TDP-43 to better understand the associations between dynamic RNA-RBP binding behavior and human diseases. The overall structure of the article is clear and the research results are quite interesting. However, there are some comments that need to be addressed before publication.

1. The authors employed icSHAPE to obtain in-vivo RNA secondary structures profiles. However, it is difficult to obtain structure profiles from icSHAPE for most of the given RNA sequences. Therefore, the authors should explore other RNA structure representation methods such as RNAfold, a computational tool for predicting RNA secondary structure, as an alternative to icSHAPE and demonstrate their impact on prediction performance.
2. The authors compared HDRNet with other existing deep learning approaches but did not compare it with machine learning algorithms. Therefore, the validity of HDRNet cannot be guaranteed, and the authors should include a comparison with machine learning algorithms.
3. The authors proposed a hierarchical deep neural network to extract sequence and structure features separately. However, they did not combine the multi-source features at the beginning of the network. The authors should explain the advantages of using hierarchical networks and the differences between the learned characteristics of hierarchical and non-hierarchical networks.
4. The motif analysis and biological interpretability provided by the authors are interesting. However, the manuscript lacks a detailed description of the motif calculation method. Therefore, the authors should include this part in an appropriate place.
5. The authors studied the effect of variant alleles within the high-attention binding region on RNA-RBP binding on the TDP-43 dataset. However, they only explored the sequence variation, and it is unclear whether this variation could lead to structural changes in the sequence.
6. The authors provide a user-friendly web server for online static and dynamic RNA-RBP binding prediction, which is helpful for biologists. However, the web server address is a digital IP instead of a domain name, making it unofficial. Additionally, there is a lack of detailed instructions and potential bug previews in the web server. Furthermore, when uploading a .fasta file with 1000 sequences, the results were returned in an acceptable time, but the authors should consider using a more advanced server to increase the speed of operation. Finally, the authors should provide an option to download the results and an optional visualization of the high-attention binding region.

Reviewer #2 (Remarks to the Author):

This manuscript by Zhu et al presents HDRNet, a new deep learning-based approach to predict dynamic RNA binding protein (RBP) in different cellular contexts. The main components of the HDRNet pipeline consist of i) the generation of input multi-source information by characterizing RNA sequence and structure profiles using BERT and the icSHAPE, respectively, ii) the unified alignment of the sequence and structure feature representation maps, and iii) extraction, prediction, and scoring to determine the prominent nucleotide tokens and RNA-protein interactions using the hierarchical multi-scale residual network (HMRN) and the deep protein-RNA binding predictor (DPRBP). By comparing five baseline methods (PrismNet, DMSK, iDeep, DeepBlind and Graphroot) and subsequent evaluations, the authors demonstrate the robustness and better

performance of the proposed HDRNet pipeline. Moreover, HDRNet is employed to reveal the relationship between TDP-43 binding properties and disease-related genetic variants. The approach and results introduced by this study will be valuable for the RNA and related research community. However, a revision will be required to further clarify the key analyses and data interpretation to better illustrate the function and usage of the HDRNet pipeline.

Major comments:

1) Why is icSHAPE pipeline specifically used for generating RNA structure profiles? icSHAPE papers should be referenced properly in the main text. Also, it would be helpful for authors to compare and discuss icSHAPE and other methods (e.g., icSHAPE-MaP, DMS-seq and related). Will integrating RNA structure information from different methods further improve the performance of HDRNet?

2) Are there sub-groups of binding events that are better characterized by HDRNet (e.g., high vs low AUC scores) or improved at different levels by the newly developed HDRNet vs other methods (e.g., PrismNet)? What are the features associated with these sub-groups (e.g., structural complexity or context variations)?

3) Related to 2), FMR1 and FXR2 binding site identification enhancement by HDRNet is highlighted as examples in the main text. Is this only observed in HEK293 cells or also in other cells? What are the specific features associated with FMR1 and FXR2?

4) How does HDRNet perform for RBPs with high and low expression levels and for target RNA events with high and low expression levels in different cellular contexts?

5) In addition to cell lines, how does HDRNet perform for predicting dynamic RNA-protein interactions in vivo tissue contexts (normal or disease conditions)? Is the model trained using cell line data (e.g., K562 and/or HepG2) sufficient for predicting dynamic interactions in tissues? Or additional tissue data are required. Examples, such as CLIP-Seq data from in brain, heart, muscle, and myoblasts (Wang et al. Cell 2012, PMID: 22901804), could be useful to further explore and compare the applications of HDRNet in different contexts?

Minor comments:

6) Some supplementary notes/figures (e.g., 1, 4 ...) are not referenced in the main text.

7) Page 12: "transformation to the variant alleles" -> alleles and improper citation "[cite:bardou2014jvonn]".

8) Further clarification for static and dynamic protein-RNA interactions would be helpful, including the authors' choice of dynamic embedding model BERT and the impact and limitations of input data from different contexts (e.g., different cell lines, tissues, and disease conditions).

9) A better description of the candidate drug prediction analysis in Fig. S10 (also highlighted in Fig. 1e) would be helpful. How many neurological disease-associated genes (68 genes from Supplementary Note 6 or 28 genes from Fig. 8c) are used for the analysis?

Reviewer #3 (Remarks to the Author):

This is a new method for the prediction of protein-RNA interactions in the RNA sequences. It relies on a deep network and several input types. I appreciate a comprehensive nature of the empirical comparisons and the downstream analysis of the results. The ablation analysis is also a plus. However, I have a few concerns.

PrismNet was published over 2 years ago and may no longer represent state of the art in the field. Authors heavily rely on this approach as motivation for their own solution. More recent methods should be surveyed and considered. For instance, a recently released relevant framework, PRIESTESS (<https://academic.oup.com/nar/article/50/19/e111/6677328>), should be investigated. Authors should compare against it and consider this type of approach as a useful

resource to augment inputs to their network.

While the proposed predictive architecture is an improvement over PrismNet (with the above caveat), the use of BERT, deep residual networks and self-attention is not particularly innovative. The literature is already populated with similar "end-to-end" solutions. The authors need to provide much more complete accounting of modern methods and more compelling explanation of the novel aspects of their method.

The survey of tools that is included in the introduction has missed an entire large community that performs these predictions on the protein side, by predicting RNA-binding proteins and RNA-binding amino acids.

On a minor note, quality of writing needs substantial improvements. There are grammatical errors (even in the abstract), some passages/sentences are poorly phrased, problems with articles, and other issues. The article should be carefully copyedited by a skilled native speaker.

Reviewer #1:

Predicting RNA-RBP interactions in a dynamic manner across different cell lines is an important challenge for understanding the interactions between RNAs and RBPs. In this manuscript, the authors present a novel deep-learning method called HDRNet to address this problem. The method utilizes multi-source biological features, including dynamic global contextual information and in-vivo RNA secondary structure profiles. The authors compare HDRNet with existing tools in multiple RBP datasets and demonstrate its superior performance in both static and dynamic prediction tasks. Additionally, the authors perform motif analysis and high-attention binding region identification to provide biological interpretability. Furthermore, the authors conduct potential genetic variants analysis and genome-wide functional studies for TDP-43 to better understand the associations between dynamic RNA-RBP binding behavior and human diseases. The overall structure of the article is clear and the research results are quite interesting. However, there are some comments that need to be addressed before publication.

Response: We would like to thank you for your insightful comments and suggestions. For the past 4 months, we have duly followed your comments and suggestions to carefully address all the raised concerns as elaborated in the following responses. In particular, we would like to thank you for your constructive comments that enabled us to improve our manuscript in numerous ways. Thank you very much for your precious time and kind attention.

1. *The authors employed icSHAPE to obtain in-vivo RNA secondary structures profiles. However, it is difficult to obtain structure profiles from icSHAPE for most of the given RNA sequences. Therefore, the authors should explore other RNA structure representation methods such as RNAfold, a computational tool for predicting RNA secondary structure, as an alternative to icSHAPE and demonstrate their impact on prediction performance.*

Response: Thank you very much for your constructive comments. We agree that it is difficult to obtain structure profiles from icSHAPE for most of the given RNA sequences. Therefore, to follow your suggestions, we conducted additional experiments to explore other RNA secondary structure representation models including RNAfold (1) and two other in-vivo secondary structural characterization methods: DMS-seq (2) and DMS-MaPSeq (3). In addition, we also visualized the secondary structure of RNA using those diverse structural features to illustrate the differences in RNA structural representations.

Supplementary Note 2: Comparing icSHAPE with other in vivo and computationally predicted secondary structure profiles

To demonstrate the effectiveness of icSHAPE in our HDRNet model further, we conducted an experiment to compare icSHAPE with other RNA secondary structure representation models, namely RNAfold (1) and two in-vivo secondary structural characterization methods: DMS-seq (2) and DMS-MaPSeq (3). To ensure a fair comparison, we kept all other modules of HDRNet, substituting only icSHAPE with the structural features derived from the comparative methods. The experimental results are

summarized in **Supplementary Fig. 2**. As depicted in **Supplementary Fig. 2a**, the HDRNet model with icSHAPE for structural representation outperformed HDRNet models with other structural features in identifying protein-RNA interactions. We speculate that computationally predicted secondary structures could be obsolete and hence prone to artifacts. In contrast, in-vivo secondary structure scores, representing the probabilities of nucleotide pairing, could potentially offer rich structural information. Taking the SLTM protein dataset from K562 cells as an example, we subsequently utilized t-SNE to visualize the embedding representation of HDRNet with various structural information. As illustrated in **Supplementary Fig. 2b**, icSHAPE demonstrated a distinct division between positive and negative samples, with each group clearly situated on opposite sides. However, for DMS-seq and DMS-MaPseq, positive and negative samples are considerably overlapped between the clusters, whereas the performance of RNAfold was deemed suboptimal due to insufficient sample separation.

Supplementary Fig. 2. (a) Overall results of HDRNet using different structure information. HDRNet with icSHAPE performs best in both static and dynamic prediction tasks. (b) The latent embedding of the learned features by HDRNet using different types of input structure features. (c) Left: The high attention binding region identified by HDRNet with different input structure characterization. Right: The structural models predicted by RNAfold with the corresponding structural technology scores as constraints.

Supplementary Fig. 2c provides a visual representation of the high-attention binding regions identified by different RNA secondary structure representation models. Notably, icSHAPE and RNAfold, when used as the structural features within the

HDRNet model, successfully captured continuous high-attention regions. Conversely, DMS-seq and DMS-MaPseq exhibited a considerable number of missing positions, resulting in a lack of continuity in the regions they identify. Moreover, the RNA models with those secondary structure features exhibit significant differences; for instance, icSHAPE could predict RNA models with greater structural heterogeneity and a higher number of nucleotide pairings. Conversely, RNAfold structures resembled those constrained using DMS-MaPseq, suggesting that the performance of DMS-MaPseq in structure prediction may be vulnerable to missing positions. On this basis, we can conclude that these analyses not only highlight the effectiveness of icSHAPE features but also underscores potential limitations. Since the acquisition of icSHAPE data relies on sequencing experiments, RNAfold or other secondary structure prediction algorithms may serve as suitable alternatives for prediction tasks. These observations emphasize the importance of carefully selecting the most appropriate RNA secondary structure representation model based on the specific requirements of the task at hand.

We also conducted further experiments to investigate and compare the performance of HDRNet by integrating RNA structure information from different methods. Specifically, we explored multiple secondary structure features obtained from different methods and then employed a concatenation strategy to integrate it into the HDRNet model. As demonstrated in **Supplementary Fig. 3**, our observations indicate that the performance of HDRNet was not further improved but slightly decreased by integrating other secondary structure features. This observation indicates that those in vivo secondary structure information provides a more accurate depiction of the RNA environment than the computationally predicted secondary structure information (RNAfold), thus ensuring the performance of HDRNet. These results support the reliability of HDRNet and suggest that the original HDRNet is sufficiently robust as a standalone model.

Supplementary Fig. 3. Ablation comparison results of integrating RNA structure information from different methods. It can be observed that combining other features didn't further improve the performance of HDRNet.

2. The authors compared HDRNet with other existing deep learning approaches but did not compare it with machine learning algorithms. Therefore, the validity of HDRNet cannot be guaranteed, and the authors should include a comparison with machine learning algorithms.

Response: Thank you very much for your valuable and insightful comments. We agree that further efforts are necessary to ensure the validity of HDRNet. We have followed your comments to conduct an experiment comparing HDRNet with seven machine learning algorithms, namely XGBoost (4), Random Forest (RF) (5), Logistic Regression (LR) (6), Artificial Neural Network (ANN) (7), ExtraTreeClassifier (ETC) (8), SGDClassifier (SGDC) (9), and GaussianNB (GNB) (10) in **Supplementary Note 1**. These algorithms are implemented using scikit-learn package (11).

Supplementary Note 1: HDRNet provides better performance than classic machine learning algorithms

To further demonstrate the effectiveness of our proposed HDRNet, we compared HDRNet with seven machine learning algorithms, namely XGBoost (12), Random Forest (RF) (13), Logistic Regression (LR) (6), Artificial Neural Network (ANN) (7), ExtraTreeClassifier (ETC) (8), SGDClassifier (SGDC) (9), and GaussianNB (GNB) (10). These algorithms were implemented in scikit-learn package (14). The experimental results are shown in **Supplementary Fig. 1**. From the comparison results, we have observed that HDRNet outperformed those machine learning algorithms across a substantial proportion of the RBP datasets in both static and dynamic prediction tasks. As illustrated in **Supplementary Fig. 1a**, HDRNet achieved the highest AUC in the static prediction task. It is worth noting that, among the compared machine learning algorithms, XGBoost exhibited superior performance, whereas ETC performed the worst. ANN also produced comparable results, likely due to the proficiency of the neural network framework for extracting features. However, ANN performed extremely poorly on certain RBP datasets, with an AUC of 0.5, whereas HDRNet achieved an AUC exceeding 0.8. On the other hand, as depicted in **Supplementary Fig. 1b**, HDRNet consistently outperformed all compared machine learning algorithms in dynamic prediction tasks. Notably, the dynamic prediction performance of ANN was degraded, compared to the static prediction performance. Overall, these results highlight the benefits and advantages of the proposed HDRNet architecture, as the neural network framework alone performs poorly in dynamic prediction.

3. The authors proposed a hierarchical deep neural network to extract sequence and structure features separately. However, they did not combine the multi-source features at the beginning of the network. The authors should explain the advantages of using hierarchical networks and the differences between the learned characteristics of hierarchical and non-hierarchical networks.

Response: Thank you very much for your insights. We agree with you and have followed your comments to explain the advantages on our chosen network architecture and differences. In HDRNet, we leverage multiple features to predict the binding sites of RBP, including the sequence features generated by BERT, along with *in vivo* secondary structure information derived from icSHAPE. From the perspective of feature dimensions, sequence features have 768 channels, whereas the secondary structure features are vectors, corresponding in length to the sequence and possessing a single channel. Given the disparity in channel dimensionality between those two types of features, we were concerned that integrating them at the beginning of the model could potentially attenuate the representation of the secondary structure features, thereby affecting prediction performance and

Supplementary Fig. 1. (a) Scatter plot comparing AUC scores of HDRNet with other machine learning algorithms. (b) Comparative analysis of the overall dynamic prediction performance between HDRNet and other machine learning methods, where "****" denotes a p-value ≤ 0.001 , "****" signifies a p-value ≤ 0.01 , and "***" indicates a p-value ≤ 0.05 .

the identification of high-attention binding regions. To address this concern, we conducted an experiment to investigate the impact of hierarchical networks versus non-hierarchical networks on the predictive performance of our proposed HDRNet. After that, we visualized the latent embedding derived from the ablation of HDRNet to discern the disparities learned by hierarchical and non-hierarchical networks.

To demonstrate the effectiveness of our chosen network architecture, we initially constructed a non-hierarchical HDRNet, referred to as HDRNet-nonhier, which combined multi-source features at the network's inception. After that, a comparative analysis was conducted between our proposed HDRNet and HDRNet-nonhier, evaluating their performance in both static and dynamic prediction tasks. The comparative results are summarized in **Supplementary Fig. 6a**, revealing that the HDRNet, characterized by its hierarchical structure, exhibited significantly superior prediction performance compared to its non-hierarchical counterpart (HDRNet-nonhier). Moreover, HDRNet, with its hierarchical architecture, demonstrated superior performance in intercellular dynamic prediction. Furthermore, we performed feature correlation analysis and visualization across different versions of HDRNet. For illustration purpose, we employed the TBRG4 protein dataset in HepG2 cells as a representative sample, as depicted in **Supplementary Fig. 6b**. We can observe that HDRNet with a hierarchical structure has a more distinct feature correlation and hierarchy than the non-hierarchical one. Therefore, sequence information and secondary structure profiles can be normalized and fused, enriching the features learned by HDRNet to be robust.

Supplementary Fig. 6. (a) Performance comparison of HDRNet with different hierarchical structures. The results demonstrate that the proposed hierarchical network achieves superior performance in both static and dynamic prediction tasks. (b) Feature correlation analysis of different HDRNet versions. Notably, HDRNet exhibits substantial feature correlation and feature hierarchy.

4. The motif analysis and biological interpretability provided by the authors are interesting. However, the manuscript lacks a detailed description of the motif calculation method. Therefore, the authors should include this part in an appropriate place.

Response: Thank you very much for the constructive comments. We agree with you and have followed your comments to provide a comprehensive description of the motif calculation method in **Supplementary Note 20**, which is copied below for easy reference:

Supplementary Note 20: Motif calculation

The multi-head self-attention mechanism has the capacity to accurately identify and decode significant genomic regions. In our study, we leverage dynamic global contextual embedding to explore the biological functionalities of RNA sequences. More explicitly, we computed the attention score for the i -th embedded k -mer token, denoted as $Score_i$ through the summation, thereby identifying the transcript fragment in each RNA sequence that offers the most substantial contribution towards downstream classification, which can be formulated as follows:

$$Score_i = \sum_{n=1}^N \frac{\exp(Q_{CLS}^T \cdot K_i / \sqrt{d})}{\sum_{t=1}^T \exp(Q_{CLS}^T \cdot K_t / \sqrt{d})} \quad (1)$$

where N symbolizes the number of attention heads, T represents the number of tokens in a given RNA sequence, Q_{CLS}^T denotes the query vector of the special tokens [CLS], K_i denotes the key vector of the i -th k -mer token with $i \in \{1, 2, \dots, L\}$ where L denotes the number of input tokens, and d is the dimensionality of the feature vectors. After computing the attention score for each token, we classified RNA fragments as RBP-binding motifs based on the following three criteria: (1) The attention score of the fragment exceeds the average score of the sequence; (2) The attention score of the fragment is 10 times higher than the lowest attention score; and (3) The minimum length of the fragment is 6.

5. The authors studied the effect of variant alleles within the high-attention binding region on RNA-RBP binding on the TDP-43 dataset. However, they only explored the sequence variation, and it is unclear whether this variation could lead to structural changes in the sequence.

Response: Thank you very much for the insightful comments. Indeed, sequence homology and its relationships to structural homology are always intriguing. To address your concerns, we have followed your comments to discuss the potential structural changes led by sequence variations.

As depicted in **Supplementary Fig. 16 from Supplementary Note 13**, we visualized RNA models with significant structural alterations. In particular, we employed RNAfold to predict and visualize the possible RNA structure after locus mutation, and then compared it with the predicted secondary structures constrained by icSHAPE. **Supplementary Fig. 16a** summarizes the high-attention binding regions of the NDUFA12-binding transcript variant (ENST00000547986) of TDP-43 identified by HDRNet in K562 cells, both before and after mutations, along with the corresponding predicted RNA secondary structure models. From this figure, we can observe that the mutations occurring at positions 94971518 and 94971529 on chromosome 12 could

potentially lead to considerable change in the overall structure of the segment after the local variation; for instance, an A->U mutation at position 25 of the sequence results in a loss of pairing with U at position 61, instead pairing with A at a closer position, i.e. 47. Additionally, the AUUUUU fragment at positions 37 to 42 forms an expanded loop structure post-mutation.

Similar structural changes were also identified in transcript ENST00000517367, the NDUFB9 binding gene of TDP-43 in HEK293 cells. As demonstrated in **Supplementary Fig. 16b**, the mutations at positions 124539290 and 124539313 on chromosome 8 also lead to structural variants, where the C->G mutation at position 2 and the G->C mutation at position 25 result in pairings that would not ordinarily occur in that segment: the G at position 2 paired with the C at position 100, and the C at position 25 paired with the G at position 10. Furthermore, **Supplementary Fig. 17** provides the influence of RNA base mutations on global structural changes. As depicted in **Supplementary Fig. 17a**, the G->C mutation at position 40 within the ENST00000276689 transcript does not modify the pairing behavior at that particular position; however it leads to a significant alteration in the overall structure of the segment. Similarly, **Supplementary Fig. 17b** highlights the impact of the AA->UU mutation at positions 47-48 in ENST00000517367, inducing a structural shift. For instance, the C at position 10 no longer pairs with the G at position 23 but instead aligns with the G at position 70, while preserving its pairing behavior at the mutation site. Overall, the above analysis illustrates that sequence variants may lead to substantial structural alterations within RNA sequences, which in turn affects the binding behavior.

6. The authors provide a user-friendly web server for online static and dynamic RNA-RBP binding prediction, which is helpful for biologists. However, the web server address is a digital IP instead of a domain name, making it unofficial. Additionally, there is a lack of detailed instructions and potential bug previews in the web server. Furthermore, when uploading a .fasta file with 1000 sequences, the results were returned in an acceptable time, but the authors should consider using a more advanced server to increase the speed of operation. Finally, the authors should provide an option to download the results and an optional visualization of the high-attention binding region.

Response: Thank you very much for your thoughtful comments. We apologize that our web server currently utilizes a digital IP rather than a domain name, which may detract from its perceived formality. We have obtained a domain name at <http://www.aibio-lab.com:5050/> for our server to enhance its credibility and ease of access. Then, we have followed your comments to implement a help webpage, which is now linked in the heading section of the website, providing instructions and potential bug previews. Its screenshot is copied below for your easy reference:

To improve the speed and the download process of the results, we have replaced the previous server with a more advanced one, resulting in enhanced processing speed and efficiency. Furthermore, the results will be sent to users via emails, allowing users to access the predicted results and the visualization of the high-attention binding region.

(a)

TDP-43, K562
ENST00000547986 159-259
NDUFA12

chr12:94971518

chr12:94971529

(b)

TDP-43, HEK293
ENST00000517367 146-246
NDUFB9

chr8:124539290

chr8:124539313

Supplementary Fig. 16. Nucleotides mutation can potentially lead to local structural variations in RNA.

(a)

TDP-43, K562
ENST00000276689 241-341
NDUFB9

(b)

TDP-43, HEK293
ENST00000517367 355-455
NDUFB9

Supplementary Fig. 17. Alterations in nucleotides can potentially induce variations in global RNA structure.

HDRNet web-server is a user-friendly deep-learning platform for RNA-RBP binding sites identification from the level of RNA sequences. We provide a one-stop-shop prediction service for researchers under the Predict sub_tab, where users can select different RBP-cell line combinations according to their own needs to accomplish RBP binding site prediction under different contextual conditions. We also provide all 261 RBP binding site datasets and the pre-trained BERT model for characterizing RNA sequences under the Download sub_tab.

1. Prediction context selection.

• In order to perform RBP binding site prediction, users should first select the prediction context according to their own needs. Specifically, users should select a particular RBP (e.g., AARS) and the cell line environment (e.g., K562) in which the RBP is located. Note that if users are not sure which RBP should be predicted, or in which cell line, or simply wants to predict if it is a general locus, we provide an "ALL" option to use the HDRNet model trained on all RBP datasets (general model) for prediction. When any of the "ALL" options in the RBP or Cell line is selected, we use the general model to make predictions.

• The Prediction type option allows the user to choose between static or dynamic prediction. In the static prediction, the prediction will be performed on the specific RBP and cell line selected by the user. While in dynamic prediction, the prediction is performed on other cell lines of the selected RBP. Note that due to the limitations of the datasets availability, dynamic prediction can currently only be performed between K562 and HepG2 cell lines for 62 RBPs, including:

AGGF1,AKAPI1,AQR,AAUH,BUD13,CSTF2T,DDX3K,DDX52,DDX55,DDX56,DGCR8,DHX30,DROSHA,EFTUD2,FAM120A,FASTKD2,FTO,FXR2,GRWD1,GTZF1,HLLTF,HN RNP1,HNRNP1,HNRNP2,HNRNP3,HNRNP4,HNRNP5,HNRNP6,HNRNP7,HNRNP8,HNRNP9,HNRNP10,HNRNP11,HNRNP12,HNRNP13,HNRNP14,HNRNP15,HNRNP16,HNRNP17,HNRNP18,HNRNP19,HNRNP20,HNRNP21,HNRNP22,HNRNP23,HNRNP24,HNRNP25,HNRNP26,HNRNP27,HNRNP28,HNRNP29,HNRNP30,HNRNP31,HNRNP32,HNRNP33,HNRNP34,HNRNP35,HNRNP36,HNRNP37,HNRNP38,HNRNP39,HNRNP40,HNRNP41,HNRNP42,HNRNP43,HNRNP44,HNRNP45,HNRNP46,HNRNP47,HNRNP48,HNRNP49,HNRNP50,HNRNP51,HNRNP52,HNRNP53,HNRNP54,HNRNP55,HNRNP56,HNRNP57,HNRNP58,HNRNP59,HNRNP60,HNRNP61,HNRNP62,HNRNP63,HNRNP64,HNRNP65,HNRNP66,HNRNP67,HNRNP68,HNRNP69,HNRNP70,HNRNP71,HNRNP72,HNRNP73,HNRNP74,HNRNP75,HNRNP76,HNRNP77,HNRNP78,HNRNP79,HNRNP80,HNRNP81,HNRNP82,HNRNP83,HNRNP84,HNRNP85,HNRNP86,HNRNP87,HNRNP88,HNRNP89,HNRNP90,HNRNP91,HNRNP92,HNRNP93,HNRNP94,HNRNP95,HNRNP96,HNRNP97,HNRNP98,HNRNP99,HNRNP100,HNRNP101,HNRNP102,HNRNP103,HNRNP104,HNRNP105,HNRNP106,HNRNP107,HNRNP108,HNRNP109,HNRNP110,HNRNP111,HNRNP112,HNRNP113,HNRNP114,HNRNP115,HNRNP116,HNRNP117,HNRNP118,HNRNP119,HNRNP120,HNRNP121,HNRNP122,HNRNP123,HNRNP124,HNRNP125,HNRNP126,HNRNP127,HNRNP128,HNRNP129,HNRNP130,HNRNP131,HNRNP132,HNRNP133,HNRNP134,HNRNP135,HNRNP136,HNRNP137,HNRNP138,HNRNP139,HNRNP140,HNRNP141,HNRNP142,HNRNP143,HNRNP144,HNRNP145,HNRNP146,HNRNP147,HNRNP148,HNRNP149,HNRNP150,HNRNP151,HNRNP152,HNRNP153,HNRNP154,HNRNP155,HNRNP156,HNRNP157,HNRNP158,HNRNP159,HNRNP160,HNRNP161,HNRNP162,HNRNP163,HNRNP164,HNRNP165,HNRNP166,HNRNP167,HNRNP168,HNRNP169,HNRNP170,HNRNP171,HNRNP172,HNRNP173,HNRNP174,HNRNP175,HNRNP176,HNRNP177,HNRNP178,HNRNP179,HNRNP180,HNRNP181,HNRNP182,HNRNP183,HNRNP184,HNRNP185,HNRNP186,HNRNP187,HNRNP188,HNRNP189,HNRNP190,HNRNP191,HNRNP192,HNRNP193,HNRNP194,HNRNP195,HNRNP196,HNRNP197,HNRNP198,HNRNP199,HNRNP200,HNRNP201,HNRNP202,HNRNP203,HNRNP204,HNRNP205,HNRNP206,HNRNP207,HNRNP208,HNRNP209,HNRNP210,HNRNP211,HNRNP212,HNRNP213,HNRNP214,HNRNP215,HNRNP216,HNRNP217,HNRNP218,HNRNP219,HNRNP220,HNRNP221,HNRNP222,HNRNP223,HNRNP224,HNRNP225,HNRNP226,HNRNP227,HNRNP228,HNRNP229,HNRNP230,HNRNP231,HNRNP232,HNRNP233,HNRNP234,HNRNP235,HNRNP236,HNRNP237,HNRNP238,HNRNP239,HNRNP240,HNRNP241,HNRNP242,HNRNP243,HNRNP244,HNRNP245,HNRNP246,HNRNP247,HNRNP248,HNRNP249,HNRNP250,HNRNP251,HNRNP252,HNRNP253,HNRNP254,HNRNP255,HNRNP256,HNRNP257,HNRNP258,HNRNP259,HNRNP260,HNRNP261,HNRNP262,HNRNP263,HNRNP264,HNRNP265,HNRNP266,HNRNP267,HNRNP268,HNRNP269,HNRNP270,HNRNP271,HNRNP272,HNRNP273,HNRNP274,HNRNP275,HNRNP276,HNRNP277,HNRNP278,HNRNP279,HNRNP280,HNRNP281,HNRNP282,HNRNP283,HNRNP284,HNRNP285,HNRNP286,HNRNP287,HNRNP288,HNRNP289,HNRNP290,HNRNP291,HNRNP292,HNRNP293,HNRNP294,HNRNP295,HNRNP296,HNRNP297,HNRNP298,HNRNP299,HNRNP300,HNRNP301,HNRNP302,HNRNP303,HNRNP304,HNRNP305,HNRNP306,HNRNP307,HNRNP308,HNRNP309,HNRNP310,HNRNP311,HNRNP312,HNRNP313,HNRNP314,HNRNP315,HNRNP316,HNRNP317,HNRNP318,HNRNP319,HNRNP320,HNRNP321,HNRNP322,HNRNP323,HNRNP324,HNRNP325,HNRNP326,HNRNP327,HNRNP328,HNRNP329,HNRNP330,HNRNP331,HNRNP332,HNRNP333,HNRNP334,HNRNP335,HNRNP336,HNRNP337,HNRNP338,HNRNP339,HNRNP340,HNRNP341,HNRNP342,HNRNP343,HNRNP344,HNRNP345,HNRNP346,HNRNP347,HNRNP348,HNRNP349,HNRNP350,HNRNP351,HNRNP352,HNRNP353,HNRNP354,HNRNP355,HNRNP356,HNRNP357,HNRNP358,HNRNP359,HNRNP360,HNRNP361,HNRNP362,HNRNP363,HNRNP364,HNRNP365,HNRNP366,HNRNP367,HNRNP368,HNRNP369,HNRNP370,HNRNP371,HNRNP372,HNRNP373,HNRNP374,HNRNP375,HNRNP376,HNRNP377,HNRNP378,HNRNP379,HNRNP380,HNRNP381,HNRNP382,HNRNP383,HNRNP384,HNRNP385,HNRNP386,HNRNP387,HNRNP388,HNRNP389,HNRNP390,HNRNP391,HNRNP392,HNRNP393,HNRNP394,HNRNP395,HNRNP396,HNRNP397,HNRNP398,HNRNP399,HNRNP400,HNRNP401,HNRNP402,HNRNP403,HNRNP404,HNRNP405,HNRNP406,HNRNP407,HNRNP408,HNRNP409,HNRNP410,HNRNP411,HNRNP412,HNRNP413,HNRNP414,HNRNP415,HNRNP416,HNRNP417,HNRNP418,HNRNP419,HNRNP420,HNRNP421,HNRNP422,HNRNP423,HNRNP424,HNRNP425,HNRNP426,HNRNP427,HNRNP428,HNRNP429,HNRNP430,HNRNP431,HNRNP432,HNRNP433,HNRNP434,HNRNP435,HNRNP436,HNRNP437,HNRNP438,HNRNP439,HNRNP440,HNRNP441,HNRNP442,HNRNP443,HNRNP444,HNRNP445,HNRNP446,HNRNP447,HNRNP448,HNRNP449,HNRNP450,HNRNP451,HNRNP452,HNRNP453,HNRNP454,HNRNP455,HNRNP456,HNRNP457,HNRNP458,HNRNP459,HNRNP460,HNRNP461,HNRNP462,HNRNP463,HNRNP464,HNRNP465,HNRNP466,HNRNP467,HNRNP468,HNRNP469,HNRNP470,HNRNP471,HNRNP472,HNRNP473,HNRNP474,HNRNP475,HNRNP476,HNRNP477,HNRNP478,HNRNP479,HNRNP480,HNRNP481,HNRNP482,HNRNP483,HNRNP484,HNRNP485,HNRNP486,HNRNP487,HNRNP488,HNRNP489,HNRNP490,HNRNP491,HNRNP492,HNRNP493,HNRNP494,HNRNP495,HNRNP496,HNRNP497,HNRNP498,HNRNP499,HNRNP500,HNRNP501,HNRNP502,HNRNP503,HNRNP504,HNRNP505,HNRNP506,HNRNP507,HNRNP508,HNRNP509,HNRNP510,HNRNP511,HNRNP512,HNRNP513,HNRNP514,HNRNP515,HNRNP516,HNRNP517,HNRNP518,HNRNP519,HNRNP520,HNRNP521,HNRNP522,HNRNP523,HNRNP524,HNRNP525,HNRNP526,HNRNP527,HNRNP528,HNRNP529,HNRNP530,HNRNP531,HNRNP532,HNRNP533,HNRNP534,HNRNP535,HNRNP536,HNRNP537,HNRNP538,HNRNP539,HNRNP540,HNRNP541,HNRNP542,HNRNP543,HNRNP544,HNRNP545,HNRNP546,HNRNP547,HNRNP548,HNRNP549,HNRNP550,HNRNP551,HNRNP552,HNRNP553,HNRNP554,HNRNP555,HNRNP556,HNRNP557,HNRNP558,HNRNP559,HNRNP560,HNRNP561,HNRNP562,HNRNP563,HNRNP564,HNRNP565,HNRNP566,HNRNP567,HNRNP568,HNRNP569,HNRNP570,HNRNP571,HNRNP572,HNRNP573,HNRNP574,HNRNP575,HNRNP576,HNRNP577,HNRNP578,HNRNP579,HNRNP580,HNRNP581,HNRNP582,HNRNP583,HNRNP584,HNRNP585,HNRNP586,HNRNP587,HNRNP588,HNRNP589,HNRNP590,HNRNP591,HNRNP592,HNRNP593,HNRNP594,HNRNP595,HNRNP596,HNRNP597,HNRNP598,HNRNP599,HNRNP600,HNRNP601,HNRNP602,HNRNP603,HNRNP604,HNRNP605,HNRNP606,HNRNP607,HNRNP608,HNRNP609,HNRNP610,HNRNP611,HNRNP612,HNRNP613,HNRNP614,HNRNP615,HNRNP616,HNRNP617,HNRNP618,HNRNP619,HNRNP620,HNRNP621,HNRNP622,HNRNP623,HNRNP624,HNRNP625,HNRNP626,HNRNP627,HNRNP628,HNRNP629,HNRNP630,HNRNP631,HNRNP632,HNRNP633,HNRNP634,HNRNP635,HNRNP636,HNRNP637,HNRNP638,HNRNP639,HNRNP640,HNRNP641,HNRNP642,HNRNP643,HNRNP644,HNRNP645,HNRNP646,HNRNP647,HNRNP648,HNRNP649,HNRNP650,HNRNP651,HNRNP652,HNRNP653,HNRNP654,HNRNP655,HNRNP656,HNRNP657,HNRNP658,HNRNP659,HNRNP660,HNRNP661,HNRNP662,HNRNP663,HNRNP664,HNRNP665,HNRNP666,HNRNP667,HNRNP668,HNRNP669,HNRNP670,HNRNP671,HNRNP672,HNRNP673,HNRNP674,HNRNP675,HNRNP676,HNRNP677,HNRNP678,HNRNP679,HNRNP680,HNRNP681,HNRNP682,HNRNP683,HNRNP684,HNRNP685,HNRNP686,HNRNP687,HNRNP688,HNRNP689,HNRNP690,HNRNP691,HNRNP692,HNRNP693,HNRNP694,HNRNP695,HNRNP696,HNRNP697,HNRNP698,HNRNP699,HNRNP700,HNRNP701,HNRNP702,HNRNP703,HNRNP704,HNRNP705,HNRNP706,HNRNP707,HNRNP708,HNRNP709,HNRNP710,HNRNP711,HNRNP712,HNRNP713,HNRNP714,HNRNP715,HNRNP716,HNRNP717,HNRNP718,HNRNP719,HNRNP720,HNRNP721,HNRNP722,HNRNP723,HNRNP724,HNRNP725,HNRNP726,HNRNP727,HNRNP728,HNRNP729,HNRNP730,HNRNP731,HNRNP732,HNRNP733,HNRNP734,HNRNP735,HNRNP736,HNRNP737,HNRNP738,HNRNP739,HNRNP740,HNRNP741,HNRNP742,HNRNP743,HNRNP744,HNRNP745,HNRNP746,HNRNP747,HNRNP748,HNRNP749,HNRNP750,HNRNP751,HNRNP752,HNRNP753,HNRNP754,HNRNP755,HNRNP756,HNRNP757,HNRNP758,HNRNP759,HNRNP760,HNRNP761,HNRNP762,HNRNP763,HNRNP764,HNRNP765,HNRNP766,HNRNP767,HNRNP768,HNRNP769,HNRNP770,HNRNP771,HNRNP772,HNRNP773,HNRNP774,HNRNP775,HNRNP776,HNRNP777,HNRNP778,HNRNP779,HNRNP780,HNRNP781,HNRNP782,HNRNP783,HNRNP784,HNRNP785,HNRNP786,HNRNP787,HNRNP788,HNRNP789,HNRNP790,HNRNP791,HNRNP792,HNRNP793,HNRNP794,HNRNP795,HNRNP796,HNRNP797,HNRNP798,HNRNP799,HNRNP800.

• We also provide the visualization of the salient map for high-attention region of the input sequences (which might be slow). Due to the limitations of our server storage, the top 3 sequences with the highest prediction scores will be plotted, and users can attach the results after the job is done.

Select RBP

ALL ▼

Select Cell Line

ALL ▼

Prediction Type

Static Dynamic Prediction context selection

Plot high attention region

No Yes Visualization

2. Date input.

• After determining the prediction context, users can input the sequences to be predicted either by typing or by uploading files. All input sequences should be in .fasta format, with the name followed by the sequence and each sequence should better be fixed with length 101. Note that if the length of the input sequence is greater than 101, we will slice the sequence from both sides; and if the length is less than 101, we will make random nucleotide from both sides. We offer an Example button, where users can check out the correct input format by an easy clicking. We currently do not impose any restrictions on the number of sequences you can upload, but having too many sequences may result in longer prediction times.

Input Sequence Example

>Sequence_name
AGGTGGATTAGAGTAACTAGAGTAGAGTCTTGGTTGACAGGGCCCTGAAGGCGGTACACACCCTCCCTCAAGTACTTCMAAGGA

Or Upload Your .fasta File

Upload

Fig. 1. The newly developed help-webpage.

3. Task submission.

- In order for users to get the prediction results (including the probability of binding sites and the salient map) correctly and on time, users should fill in a name and a correct email address so that we can deliver the results to the user via email. Note that the name is not necessarily required, "friend" will be used as default. Once all content is filled out, users can click the **Submit** button in order to upload the prediction task to the HDRNet web service.

Name
Enter your Name

Email
Enter a valid email address

Submit

4. Results access.

- Once the prediction task is finished, a .zip archive containing the prediction results and the salient map (if the user has chosen to visualize the salient map) will be sent to the user via the email address he/she has provided. For each sequence, we give the probability that it is a binding site for the corresponding RBP. An example of the returned output is shown in the following figures:

Potential bug previews

- Users may encounter situations where the selected RBP-cell line context cannot be predicted. As previously mentioned, some combinations of prediction contexts may not be available due to limitations in dataset accessibility. The same issues may also occur with dynamic predictions. Users can determine the specific reason for prediction failures by examining the contents of the returned result file. Users can also download the dataset provided on the download page to obtain available RBP-cell line combinations. We also provide an "ALL" selection where the prediction will be performed on the HDRNet model trained on the mixture of all datasets.
- Due to varying network conditions, some users may experience submission failures or not receive prediction results. Users can either resubmit their requests or contact us for individual handling.

Fig. 2. The newly developed help-webpage.

Reviewer #2:

This manuscript by Zhu et al. presents HDRNet, a new deep learning-based approach to predict dynamic RNA binding protein (RBP) in different cellular contexts. The main components of the HDRNet pipeline consist of i) the generation of input multi-source information by characterizing RNA sequence and structure profiles using BERT and the icSHAPE, respectively, ii) the unified alignment of the sequence and structure feature representation maps, and iii) extraction, prediction, and scoring to determine the prominent nucleotide tokens and RNA-protein interactions using the hierarchical multi-scale residual network (HMRN) and the deep protein-RNA binding predictor (DPRBP). By comparing five baseline methods (PrismNet, DMSK, iDeep, DeepBind and Graphprot) and subsequent evaluations, the authors demonstrate the robustness and better performance of the proposed HDRNet pipeline. Moreover, HDRNet is employed to reveal the relationship between TDP-43 binding properties and disease-related genetic variants. The approach and results introduced by this study will be valuable for the RNA and related research community. However, a revision will be required to further clarify the key analyses and data interpretation to better illustrate the function and usage of the HDRNet pipeline.

Response: We would like to thank you for your insightful and helpful comments. For the past 4 months, we have duly followed your comments and suggestions to carefully address all the raised concerns as elaborated in the following responses. In particular, we would like to thank you for your constructive comments that enabled us to improve our manuscript in numerous ways. Thank you very much for your precious time and kind attention.

1. *Why is icSHAPE pipeline specifically used for generating RNA structure profiles? icSHAPE papers should be referenced properly in the main text. Also, it would be helpful for authors to compare and discuss icSHAPE and other methods (e.g., icSHAPE-MaP, DMS-seq and related). Will integrating RNA structure information from different methods further improve the performance of HDRNet?*

Response: Thank you very much for your insightful comments. We carefully considered multiple factors when selecting icSHAPE. One of the primary reasons is that the icSHAPE (15) pipeline allows for the generation of RNA structure profiles, providing insights into the dynamic nature of RNA structure across the entire transcriptome from *in vivo* experiments, enabling the global and accurate characterization of the relationship between RNA structure and RNA binding protein (RBP) interactions (16, 17). This is essential for the design of our predictive models, as the RNA-protein interactions we are trying to predict are fundamentally affected by these structural features. As such, our need for encapsulating these elements naturally led us towards icSHAPE.

In particular, icSHAPE is a chemical modification-based approach which is known for providing the natural reflection on RNA folding (18). Lastly, another key factor in our selection of icSHAPE is its widespread acceptance and usage within the research community (16, 17). This tool has garnered significant support and is backed by high-quality data. The availability and quality of these data, combined with icSHAPE's well-documented track record, convinced us of the tool's suitability for our application and strengthened our choice (15). In addition, we recognize the importance of acknowledging and properly referencing the foundational works that underpin our research. We have appropriately cited the relevant icSHAPE papers with 4 citations (15–18) in our revised manuscript.

Meanwhile, to address your second comment, we have followed your comment to conduct an experiment to explore other RNA secondary structure representation models including RNAfold (1) and two other *in-vivo* secondary structural characterization methods: DMS-seq (2) and DMS-MaPseq (3). It is worth noting that we excluded icSHAPE-MaP from our comparison group, as it is specifically designed for small RNA and does not provide RNA secondary structure information with sufficient length coverage (19). Therefore, it was not suitable for our RBP datasets. In addition, we decided to include DMS-MaPseq data, an evolved version of DMS-seq, in our analysis. DMS-MaPseq has proven to be highly effective in genome-wide and targeted RNA structure probing, thereby providing a broad and comprehensive landscape of RNA structures (3). Finally, we compared and discussed icSHAPE and other RNA secondary structure representation methods (RNAfold, DMS-seq, and DMS-MaPseq) in our HDRNet model to demonstrate its effectiveness.

Supplementary Note 2: Comparing icSHAPE with other in vivo and computationally predicted secondary structure profiles

To demonstrate the effectiveness of icSHAPE in our HDRNet model further, we conducted an experiment to compare icSHAPE with other RNA secondary structure representation models, namely RNAfold (1) and two *in-vivo* secondary structural characterization methods: DMS-seq (2) and DMS-MaPseq (3). To ensure a fair comparison, we kept all other modules of HDRNet, substituting only icSHAPE with the structural features derived from the comparative methods. The experimental results are summarized in **Supplementary Fig. 2**. As depicted in **Supplementary Fig. 2a**, the HDRNet model with icSHAPE for structural representation outperformed HDRNet models with other structural features in identifying protein-RNA interactions. We speculate that computationally predicted secondary structures could be obsolete and hence prone to artifacts. In contrast, *in-vivo* secondary structure scores, representing the probabilities of nucleotide pairing, could potentially offer rich structural information. Taking the SLTM protein dataset from K562 cells as an example, we subsequently utilized t-SNE to visualize the embedding representation of HDRNet with various structural information. As illustrated in **Supplementary Fig. 2b**, icSHAPE demonstrated a distinct division between positive and negative samples, with each group clearly situated on opposite sides. However, for DMS-seq and DMS-MaPseq, positive and negative samples are considerably overlapped between the clusters, whereas the performance of RNAfold was deemed suboptimal due to insufficient sample separation.

Supplementary Fig. 2c provides a visual representation of the high-attention binding regions identified by different RNA secondary structure representation models. Notably, icSHAPE and RNAfold, when used as the structural features within the HDRNet model, successfully captured continuous high-attention regions. Conversely, DMS-seq and DMS-MaPseq exhibited a considerable number of missing positions, resulting in a lack of continuity in the regions they identify. Moreover, the RNA models with those secondary structure features exhibit significant differences; for instance, icSHAPE could predict RNA models with greater structural heterogeneity and a higher number of nucleotide pairings. Conversely, RNAfold structures resembled those constrained using DMS-MaPseq, suggesting that the performance of DMS-MaPseq in structure prediction may be vulnerable to missing positions. On this basis, we can conclude that these analyses not only highlight the effectiveness of icSHAPE features but also underscores potential limitations. Since the acquisition of icSHAPE data relies on sequencing experiments, RNAfold or other secondary structure prediction algorithms may serve as suitable alternatives for prediction tasks. These observations emphasize the importance of carefully selecting the most appropriate RNA secondary structure representation model

Supplementary Fig. 2. (a) Overall results of HDRNet using different structure information. HDRNet with icSHAPE performs best in both static and dynamic prediction tasks. (b) The latent embedding of the learned features by HDRNet using different types of input structure features. (c) Left: The high attention binding region identified by HDRNet with different input structure characterization. Right: The structural models predicted by RNAfold with the corresponding structural technology scores as constraints.

based on the specific requirements of the task at hand.

We also conducted further experiments to investigate and compare the performance of HDRNet by integrating RNA structure information from different methods. Specifically, we explored multiple secondary structure features obtained from different methods and then employed a concatenation strategy to integrate it into the HDRNet model. As demonstrated in **Supplementary Fig. 3**, our observations indicate that the performance of HDRNet was not further improved but slightly decreased by integrating other secondary structure features. This observation indicates that those in vivo secondary structure information provides a more accurate depiction of the RNA environment than the computationally predicted secondary structure information (RNAfold), thus ensuring the performance of HDRNet. These results support the reliability of HDRNet and suggest that the original HDRNet is sufficiently robust as a standalone model.

Supplementary Fig. 3. Ablation comparison results of integrating RNA structure information from different methods. It can be observed that combining other features didn't further improve the performance of HDRNet.

2. Are there sub-groups of binding events that are better characterized by HDRNet (e.g., high vs low AUC scores) or improved at different levels by the newly developed HDRNet vs other methods (e.g., PrismNet)? What are the features associated with these sub-groups (e.g., structural complexity or context variations)?

Response: Thank you very much for the constructive suggestions. Yes, there exists sub-groups of binding events that are better characterized by HDRNet (e.g., high vs low AUC scores) or improved at different levels by the newly developed HDRNet vs other methods (e.g. PrismNet).

In this part, we conducted a comparison between HDRNet and PrismNet, focusing on their performance across various cell

lines, and the detailed results of this comparison are presented in **Supplementary Fig. 7a**. As demonstrated in this figure, we can observe that HDRNet was consistently superior to PrismNet in all cell lines evaluated, with significant enhancement observed in HEK293, HepG2, and K562 cells. To investigate it, we selected datasets from these three cell lines in which HDRNet was at least 5% better than PrismNet. We then analyzed the performance gap between the proposed HDRNet and PrismNet models for these datasets, as demonstrated in **Supplementary Fig. 7b**. We found that HDRNet showed a more significant performance improvement in predicting multiple datasets in HEK293 cell line. For illustration purpose, we further explored those RBPs in HEK293 cell lines. To explore their relationships, we have mapped these RBPs into STRING (20) and then used the Markov Cluster Algorithm (MCL) with an inflation rate of 2.5 to cluster them, as demonstrated in **Supplementary Fig. 7c**. As can be seen from the figure, RBPs with similar biological functions are clustered together, while RBP datasets exhibiting significant performance improvements are consistently grouped into the same functional clusters, e.g., 15% for FMR2 and 13% for FXR1, in the subclusters represented by the red nodes highlighted. Moreover, in this cluster, previous studies have demonstrated that several RBPs (e.g., FXR2, FMR1, LIN28B) have the coexpression in the same tissues (21), which further indicates that these RBPs in the cluster may have similar context or structures.

With the distinct subgroups identified, we proceeded with an analysis of the structural complexity of these RBPs. Initially, we utilized HDRNet to scan the RBP datasets and extracted the high-attention 6-mer fragments that HDRNet identified most frequently. Similar to PrismNet, we then calculated the structural complexity of these 6-mer fragments, generated the PWM matrix and represented their structural motifs where the structure component using the labels “U” for unpaired nucleotide and “P” for paired. As shown in **Supplementary Fig. 7d**, we observe that RBPs in the same subgroup show structural similarities. For instance, the subgroup of NUDT21, CPSF2, and CPSF4 exhibited a prevalent preference for binding to paired structures, whereas RBPs in the subgroup containing LIN28A demonstrated a higher tendency to interact with complex structure fragments. Meanwhile, the binding region of FBL usually does not pair. These phenomena demonstrate the potential structural differences between different subgroups.

To investigate context variations further, we explored the context variations between RBP clusters using 3-mer analysis, tailored to the tokenization used by our dynamic global contextual embedding approach. In specific, we calculated the relative content of each 3-mer token within each RBP dataset. Subsequently, we employed hierarchical clustering to group RBPs based on their 3-mer token content, thereby assigning RBPs with similar profiles to the same cluster. As shown in **Supplementary Fig. 7e**, it can be seen that the results are in close agreement with the clusters we obtained through the STRING database; for instance, RBPs such as FMR1, FXR2, LIN28A, and LIN28B were consistently assigned to the same cluster, while CPSF2 and CPSF4 were in a distinct cluster. These findings further support the notion that the 3-mer token with contextual information can serve as an informative feature to identify different sub-groups of RBPs.

Supplementary Fig. 7. (a) Performance comparison of HDRNet and PrismNet through different cell lines. (b) Performance gap between HDRNet and PrismNet through HEK293, HepG2 and K562 cell lines. (c) Identified subgroups of RBP datasets in HEK293 cell line. (d) The integrative motifs identified by HDRNet on the RBPs within different subgroups. The top half presents the high-attention 6-mer fragment identified by HDRNet most frequently, while the lower half displays the structural motifs of the 6-mer sequence, where 'P' stands for paired, and 'U' indicates unpaired. (e) Heatmap representing the relative content of each 3-mer in the RBP binding site. RBPs with similar relative contents were grouped into a cluster by hierarchical clustering.

3. Related to 2), FMR1 and FXR2 binding site identification enhancement by HDRNet is highlighted as examples in the main text. Is this only observed in HEK293 cells or also in other cells? What are the specific features associated with FMR1 and FXR2?

Response: Thank you very much for the constructive comments. According to your **Comments 2**, we observe that HDRNet outperformed PrismNet across different cell lines. In particular, HDRNet's superiority is not confined to HEK293 cells alone; it also demonstrates excellent performance in HepG2 and K562 cells; for instance, in the FMR1 dataset of the K562 cell line, the performance of HDRNet improved by 5%, and in the FXR2 dataset of K562 and HepG2 cell lines, the performance of HDRNet was boosted by 6% and 2%, respectively. To further explore the potential characteristics of FMR1 and FXR2, we also conducted additional explorations into FMR1 and FXR2 to elucidate the specific features associated with them.

Considering the intrinsic properties of FMR1 and FXR2 proteins, as previously mentioned, both FMR1 and FXR2 are members of the fragile X mental retardation protein (FMRP) family and are co-expressed in the cytoplasm of specific differentiated neurons (22). Despite being encoded by different genes located on chromosome X and chromosome 17, FMR1 and FXR2 share considerable similarity, with 60% sequence identity. Moreover, they display overlapping tissue distribution and have been found to interact *in vivo* (23). Additionally, FMRP proteins exhibit substantial functional overlap in terms of translational regulation and RNA-binding specificity (24), suggesting functional homology between FMR1 and FXR2 (25).

Furthermore, we investigated the potential properties of FMR1 and FXR2 in the context of RBP binding motifs. Firstly, we focused on the 6-mer distribution of the binding sites of FMR1 and FXR2 across different cell lines. As detailed in **Supplementary Fig. 8a**, we identified the top ten 6-mers with the highest enrichment in each dataset. Notably, the content of these 6-mers demonstrated remarkable similarities, with CAGCAG being highly represented across datasets. Moreover, a predominance of AG or CUG components within these 6-mers was noted, suggesting potential roles of AG and CUG in the binding domains of FMR1 and FXR2.

To further investigate the binding characteristics of FMR1 and FXR2, we applied HDRNet to scan these datasets and tabulated the 6-mer tokens with the maximum attention score in each sequence. The top ten high-attention 6-mers, most frequently identified by HDRNet, are depicted in **Supplementary Fig. 8b**. This analysis revealed a consensus high-attention 6-mer, CATGGA, in FMR1 datasets, while CTGCTG was consistently found across FXR2 datasets. Noteworthy observations include the presence of AGAAGA in the FMR1 dataset and AAGAAG in the FXR2 dataset, corroborating the binding motif previously suggested in (26). For a more precise visualization of binding properties, salient regions flagged by HDRNet are depicted in **Supplementary Fig. 8c**, where AG-rich and CUG-rich regions are highlighted within FMR1 and FXR2 datasets respectively. Interestingly, within the FMR1_HEK293 dataset, HDRNet accentuated the GGA segment, in spite of the prominence of the CATGGA 6-mer. Simultaneously, in the FXR2_HepG2 dataset, along with the underlined CUG region, HDRNet discerned the AAGAAG motif. Correspondingly, the attention distribution derived from the dynamic global contextual representation model further highlighted the AG-rich regions, as depicted in **Supplementary Fig. 8d**, where the constituent tokens comprise AGAAGA, coinciding with the motif we previously identified. This demonstrates the preferential focus of HDRNet on the regions enriched with AG, suggesting a potentially significant role of these AG-rich regions in the RNA binding process of

FMR1 and FXR2.

Supplementary Fig. 8. (a) Bar chart of 6-mer contents for FMR1 and FXR2 across different cell lines. (b) Statistics of the most significant 6-mer regions identified by HDRNet on the FMR1 and FXR2 datasets. (c) Visualization of the salient map detected by HDRNet, highlighting specific binding regions within the FMR1 and FXR2 datasets. (d) Visualization of attention distribution using dynamic global contextual embedding, with attention concentrated in A/G-rich regions.

4. How does HDRNet perform for RBPs with high and low expression levels and for target RNA events with high and low expression levels in different cellular contexts?

Response: Thank you very much for your insightful comments. To address your comments, we have conducted two additional analyses: (1) Firstly, to evaluate the effectiveness of HDRNet across various expression levels of RBPs in both the same and different cellular context, we obtained the expression levels of RBPs in HepG2 and K562 cell lines from the reference (27) published in *Nature*, which provides a comprehensive collection of human RBPs in K562 and HepG2 cells from the Encyclopedia of DNA Elements (ENCODE) project phase III, including the expression levels of RBPs across different cell lines within the eCLIP dataset. After that, based on these expression levels, we discussed the performance of HDRNet on RBPs with both high and low expression levels, where the average was set as the threshold border. Additionally, we further validated

the effectiveness of HDRNet by examining differential RBPs between HepG2 and K562 cells; (2) To investigate the target expression levels of RBPs, we identified RBPs with significant differences in target expression levels after RBP knockdown experiments. Based on these RBPs, we analyzed the performance of HDRNet in capturing the changes in expression levels of RBP targets. The specific analysis is copied below for your easy reference:

To observe the validity of HDRNet across various expression levels of RBPs in both the same and different cellular contexts, we compared the performance of HDRNet with baseline methods separately on K562 and HepG2 cell lines. As illustrated in **Supplementary Fig. 9a**, we observed that HDRNet exhibited superior static prediction performance for RBPs with varying expression levels in different cell lines. Furthermore, we noted a slight improvement in HDRNet's performance on RBPs with higher expression levels compared to those with lower expression levels within the same cell line. Meanwhile, to further investigate the relationship between RBP expression levels and the performance of HDRNet, we calculated their correlation. As depicted in **Supplementary Fig. 9b**, we found a weak positive correlation between the performance of HDRNet, as measured by metrics such as the AUC and RBP expression levels, suggesting that the binding sites of RBPs with higher expression levels were more likely to be accurately recognized by HDRNet.

Then, we analyzed the relationship between the performance of HDRNet and RBP expression levels from the perspective of dynamic prediction. As demonstrated in **Supplementary Fig. 9c**, we first computed the correlation of RBP expression levels in the K562 and HepG2 cell lines. We found a strong positive correlation between the RBP expression levels in these two cell lines, indicating similar RBP expression patterns between them. Therefore, we evaluated the dynamic prediction performance of HDRNet based on these categorizations. As depicted in **Supplementary Fig. 9d**, consistent with the static prediction results, HDRNet exhibited the most accurate dynamic predictive performance for these RBPs. However, it is worth noting that there are differentially expressed RBPs between the two cell lines. To identify these differentially expressed RBPs, we employed the DESeq2 package, as described in the reference (27). A total of 35 RBPs were identified, of which 15 were differentially expressed RBPs in K562 cells, and 20 were differentially expressed RBPs in HepG2 cells. **Supplementary Fig. 9e** visualizes the performance of HDRNet on these differentially expressed RBPs, confirming its superior performance in both cases. Based on these results, we further investigated whether the expression levels of RBPs influenced the dynamic prediction performance of HDRNet. As displayed in **Supplementary Fig. 9f**, we observed that the binding sites of RBPs with higher expression levels were more easily identified during the dynamic prediction process.

On the other hand, we conducted further analysis to evaluate the performance of HDRNet on RBPs with different target expression levels. RBPs play a crucial role in regulating gene expression by interacting with RNA and forming ribonucleoprotein complexes. However, the ENCODE project data (27) for RBPs in K562 and HepG2 cells does not provide expression values for their target RNAs, making it challenging to directly measure the high and low expression levels of target RNA events. Nevertheless, they have employed RNA sequencing after depleting RBPs using short hairpin RNA (shRNA) or CRISPR to investigate RBP target binding site functionality and examined expression level differences before and after knockdown (27). On this basis, there are 4 RBPs including DD3X3, DDX6, LARP4 and RBM15 that have previously been identified as RNA decay factors. After RBP-knockdown experiments, their target genes exhibited upregulation in specific cell conditions (knockdown-increased), indicating that these RBPs are associated with low expression levels of their target genes. Similarly, there are 6 RBPs including AKAP1, DDX55, APOBEC3C, FMR1, CPSF6, and IGF2BP3 that have been previously recognized

Supplementary Fig. 9. (a) Static prediction performance comparison of HDRNet and baseline models on different expression levels of RBPs. (b) The correlation between HDRNet's performance and RBP expression levels. (c) The correlation of RBP expression levels between K562 and HepG2 cell lines. (d) The performance comparison of HDRNet on RBPs with identical expression levels across cell lines. (e) The performance comparison of HDRNet on differentially expressed RBPs across cell lines. (f) The correlation between RBP expression levels and HDRNet's dynamic prediction performance.

to increase the stability of RNA targets; and their target genes showed downregulation after RBP knockdown in specific cell contexts (knockdown-decreased), indicating that these RBPs are associated with higher expression levels of their target genes. With these target RNA events with high and low expression levels, we first examined the static predictive performance of HDRNet on the specific cell line data for these RBPs. As depicted in **Supplementary Fig. 10a,b**, HDRNet consistently achieved the best performance and showed significant improvements compared to other baseline methods. For instance, in the RBM15_K562 dataset, HDRNet shows its performance gain of 8% over PrismNet; in the FMR1_K562 dataset, only HDRNet's AUC metric exceeded 0.9. These results indicate that HDRNet is capable of accommodating RBPs with different target expression levels and effectively identifying their binding characteristics. Furthermore, we selected those RBPs within this cohort that allowed for dynamic prediction and compared HDRNet's dynamic predictive performance. As demonstrated in **Supplementary Fig. 10c,d**, HDRNet also exhibited superior performance in dynamic prediction as well. For example, we observed a performance improvement of over 10% for RBM15 when comparing HDRNet to PrismNet. Notably, other baseline methods displayed considerable instability in the dynamic prediction tasks; for instance, DMSK failed to accurately identify the binding sites of DDX3X, and PrismNet performed poorly on AKAP1, with an AUC below 0.7. These results further validate the adaptability and robustness of HDRNet to RBPs with different target expression levels.

5. *In addition to cell lines, how does HDRNet perform for predicting dynamic RNA-protein interactions in vivo tissue contexts (normal or disease conditions)? Is the model trained using cell line data (e.g., K562 and/or HepG2) sufficient for predicting dynamic interactions in tissues? Or additional tissue data are required. Examples, such as CLIP-Seq data from in brain, heart, muscle, and myoblasts (Wang et al. Cell 2012, PMID: 22901804), could be useful to further explore and compare the applications of HDRNet in different contexts?*

Response: Thank you very much for the constructive comments. We agree that additional tissue data are needed to further explore and compare the applications of HDRNet across different contexts. To address your concerns, we have followed your comments to conduct three additional experiments to explore and compare the predictive capabilities of HDRNet for dynamic RNA-protein interactions across different tissue contexts: (1) We first collected MBNL2 HITS-CLIP binding profile in brain to explore the dynamic prediction performance of HDRNet in normal and disease conditions; (2) We obtained two additional eCLIP datasets, DGCR8 and HNRNPU, from adrenal tissues provided by ENCODE to validate whether HDRNet, trained on K562 or HepG2 cell line data, could accurately predict binding behavior in tissue contexts; (3) Following your recommendations, we compiled CLIP-seq data of MBNL1 from multiple tissues in mice to further validate the efficacy of HDRNet for dynamic prediction across various tissues.

To demonstrate the effectiveness of HDRNet in both normal and disease conditions, we have meticulously curated the MBNL2 (Muscleblind Like Splicing Regulator 2) binding peak data (28) (GEO accession: GSE68890) in human brain tissues from POSTAR (29). This data source is comprised of five distinct datasets, including autopsy tissues (hippocampus and frontal cortex) from the patients with myotonic dystrophy type 1 (DM1, 2 datasets), myotonic dystrophy type 2 (DM2, 1 dataset of hippocampus), and control subjects (2 datasets), where DM1 and DM2 are progressive and multi-systemic neuromuscular disorders, originating from the aberrant sequestration and activation of RNA processing factors and RAN translation. Then, we consolidated replicate data within each dataset and standardized the binding peaks to a length of 101 nucleotides. Subsequent

Supplementary Fig. 10. (a) The performance comparison of HDRNet on RBPs with lower target expression levels. (b) The performance comparison of HDRNet on RBPs with higher target expression levels. (c) The comparison of HDRNet's dynamic prediction performance on RBPs with lower target expression levels. (d) The comparison of HDRNet's dynamic prediction performance on RBPs with higher target expression levels.

evaluations of the dynamic prediction performance on these tissue datasets involved HDRNet, along with baseline models including PrismNet, DMSK, iDeep, GraphProt, DeepBind, and PRIESSTESS. Firstly, the evaluations focused on cross-tissue dynamic prediction experiments under single conditions, such as normal-normal and DM1-DM1 predictions. As shown in **Fig. 4a**, HDRNet consistently outperformed the other models across both conditions. Notably, we observed a significant performance gap between HDRNet and PrismNet in the cross-tissue prediction task, while PRIESSTESS failed to identify the binding sites of MBNL2 in the DM1 frontal cortex dataset, rendering it incapable of performing the dynamic prediction task. Then, to validate the efficacy of HDRNet in capturing dynamic tissue conditions between normal and diseases, we conducted experiments on cross-condition dynamic predictions, such as, we used the model trained on normal control tissue data to predict binding sites in disease tissues. As shown in **Fig. 4b**, HDRNet maintained strong dynamic prediction capabilities in cross-condition dynamic prediction. It achieved the highest AUC metric of 0.8, with an AUC difference exceeding 10% compared to PrismNet. These results demonstrate that HDRNet can effectively learn the binding patterns of RBPs across diverse conditions. Additionally, as highlighted in **Fig. 4c and Supplementary Fig. 11a**, HDRNet exhibited a robust capacity to detect and accurately capture disease-related high-attention binding regions. These regions represent the critical interaction sites between MBNL2 and DM1 extended CUG repeats and DM2 CCUG expansion RNAs, which are pathophysiological hallmarks of myotonic dystrophies, as previously revealed in (28, 30). By successfully capturing these regions, HDRNet not only aids in identifying key molecular interactions but also enhances our understanding of disease mechanisms.

To further investigate the predictive capabilities of our proposed model regarding dynamic interactions in tissues, we conducted an experiment to predict two additional eCLIP RBP datasets from adrenal tissues obtained from ENCODE, namely DGCR8 and HNRNPU. The experiment results are summarized in **Supplementary Fig. 11b**, where our model achieved the highest AUC metric. As depicted in this figure, we can observe that HDRNet achieved identical performance in dynamically predicting RBP binding in both K562 and HepG2 cell lines, while other baseline methods exhibited variability in performance when using models trained on different cell lines for dynamic prediction. This observation further highlights the ability of HDRNet to identify RBP binding patterns across different sources of RBP binding data. Interestingly, we also observed that PrismNet, although slightly inferior to HDRNet, showed improved performance on the new eCLIP data compared to the previous MBNL2 data. We speculate that this improvement is due to the fact that the PrismNet model was specifically designed based on eCLIP data. In particular, the static encoding within PrismNet limits its performance on other platforms. In addition, our observations revealed that the HDRNet model, trained on cell line data, successfully highlighted significant binding regions in the tissue binding data which were functionally relevant; for example, focusing on the DGCR8 protein, it has been reported to bind to extended CGG repeat sequences, leading to partial sequestration of DGCR8 in CGG RNA aggregates, consequently reducing the processing of miRNAs (31). In **Supplementary Fig. 11c**, we can observe that HDRNet effectively identifies the dynamic CGG repeat binding region, indicating its capability to dynamically identify RBP binding patterns. Additionally, HDRNet accurately identified the G-rich binding region of HNRNPU, as depicted in **Fig. 4d**. Notably, upon visualizing potential RNA models, we discovered that the identified G-rich binding regions predominantly corresponded to G-quadruplex structures, which are stable secondary structures formed by G-G base pairs. Indeed, as reported by (32), HNRNPU exhibits a preference for recognizing G-quadruplexes in RNAs as binding motifs. These results highlight the robustness of HDRNet on RBP data across different platforms, demonstrating that HDRNet with K562 and HepG2 cell line data is sufficient to accurately predict dynamic interactions in tissues.

(a) Dynamic prediction under single conditions.

(b) Dynamic prediction between normal and diseases conditions

(c)

(d)

(e)

Dynamic prediction based on model trained on C2C12

Fig. 4. (a) Performance comparison of MBNL2 dynamic prediction in different tissues in human brain. (b) MBNL2 dynamic binding prediction performance comparison across normal and disease conditions. (c) The high attention binding region of MBNL2 in DM2 hippocampus dataset captured by HDRNet. HDRNet successfully identifies the disease-related RNA repeats. (d) HDRNet identifies the salient DGCR8 binding region of G-rich segment with the G-quadruplex structure. (e) HDRNet outperforms other baselines in the dynamic prediction tasks on mice tissues, using model trained on C2C12 dataset.

(a)

(b)

(c)

Supplementary Fig. 11. (a) The salient map of the high attention binding region captured by HDRNet. HDRNet successfully identifies the disease-related RNA repeats. (b) Performance comparison of DGCR8 and HNRNPU dynamic prediction, using the model trained on cell line data. (c) HDRNet identifies the CGG-rich region of DGCR8 binding patterns.

Lastly, we have also followed your comments to add the direct regulatory targets of MBNL1 (Muscleblind Like Splicing Regulator 1) in brain, heart, muscle, and myoblasts from mice (Wang et al. Cell 2012, PMID: 22901804) (30) from the CLIP-Seq data to further explore the performance of HDRNet in different contexts. In particular, we collected a total of five datasets (GEO accession: GSE39911), including two from the brain (129Brain, B6Brain), one from muscle (B6Muscle), one from heart (B6Heart), and one from myoblasts (C2C12 cells), where 129 and B6 are individual mouse ID numbers. The experimental results, as illustrated in **Supplementary Fig. 12a and Fig. 4e**, demonstrate that HDRNet provided performance improvements compared to other baseline methods, resulting in an increase in AUC from 5% to 10%. Meanwhile, it is important to note that MBNL1 and MBNL2 are both members of the muscleblind-like (MBNL) protein family, and therefore MBNL1 shares similar binding patterns with MBNL2, as discussed previously, showing binding specificity to CUG and CCUG pathological expansions (28, 30). As illustrated in **Supplementary Fig. 12b**, HDRNet can dynamically distinguish these diseases-related RNA repeats sequences using models trained on different MBNL1 binding data, indicating that HDRNet can effectively extract RBP binding properties from the RBP binding data in different physiological environments. In summary, these results demonstrate the comprehensiveness of HDRNet in the task of dynamic prediction of RBP binding sites in multiple tissues.

Minor comments:

1. *Some supplementary notes figures (e.g., 1, 4 ...) are not referenced in the main text.*

Response: Thank you very much for the valuable comments. We apologize for some supplementary notes and figures that not referenced in the main text. We have referenced the supplementary notes and figures in the appropriate sections of the main text. Then, we (the first author and corresponding authors) have checked the revised manuscript in multiple rounds again to avoid this kind of situation.

2. *Page 12: "transformation to the variant alleles" -> alleles and improper citation "[cite:bardou2014jvonn]"*.

Response: Thank you very much for your careful comments. We have corrected the spelling errors and improper citations there. In addition, we (the first author and corresponding author) conducted several more rounds of proofreading the manuscript to ensure that these errors would not reappear.

3. *Further clarification for static and dynamic protein-RNA interactions would be helpful, including the authors' choice of dynamic embedding model BERT and the impact and limitations of input data from different contexts (e.g., different cell lines, tissues, and disease conditions).*

Response: Thank you very much for the constructive comments. We apologize for the insufficiently detailed description about the static and dynamic protein-RNA interactions. We appreciate your suggestions and have followed your comments to describe them in **Introduction**, which is copied below for your easy reference:

(a)

Dynamic prediction based on model trained on 129Brain

(b)

Supplementary Fig. 12. (a) Performance comparison of MBNL2 dynamic prediction performance using model trained on 129Brain date of mouse. (b) Salient binding regions identified by HDRNet under dynamic conditions.

Introduction

In parallel, the establishment of links between RBPs and their targets from the perspective of RNA sequences enables understandings on the regulation mechanism. A variety of efforts have been developed to address it in identical cellular conditions, referred to as static protein-RNA interactions; for instance, Deepbind (33) was developed to understand RBP binding preferences in RNA sequence data using a deep neural network (DNN). Ilan et al. developed DLPRB (34), a new DNN approach based on convolutional neural networks (CNN) and recurrent neural networks (RNN) for learning intrinsic RBP binding preferences and predicting novel interactions. Pan et al. developed iDeep (35), a hybrid convolutional neural network and deep belief network-based model to predict the RBP interaction sites and motifs on RNAs. Daniel et al. proposed GraphProt (36) that integrates sequence and computationally-predicted RNA secondary structure information into graph-kernel features. Lavery et al. introduced PRIESTEES (37), a universal RNA motif-finding/scanning strategy capable of identifying enriched RNA sequence and/or structure motifs that are subsequently reduced to a set of core motifs by logistic regression with LASSO regularization. However, these methods only account for a specific cellular condition and are hence limited in their ability to predict RNA-protein interactions in other cell lines while considering the dynamic contexts.

Indeed, the binding behavior of RBP-RNA interactions has been demonstrated to be dynamic in different cell types, as it is influenced by different cellular or tissue environments (38). In other words, the binding sites of the RBP that are exclusively present in specific cells or tissues can be designated as dynamic protein-RNA interactions. Recently, a new deep learning-based method called PrismNet (26), was developed to accurately predict dynamic RBP binding in various cellular conditions by integrating *in vivo* experimental RNA structure information (39). Nonetheless, unfortunately, we conceive that there are still room for improvements; for instance, the architectural design of the deep neural network with in PrismNet can still be investigated in the context of neural network architecture search, indicating the potential for further improvements in RBP-RNA binding prediction. In addition, the one-hot encoding representation in PrismNet exhibits limitations that may hinder its ability to capture RNA features globally. Furthermore, the heterogeneity of transcripts results in widely sequential relationships across different contexts, such as various cell lines, tissues, and normal or disease conditions. The one-hot encoding may overlook this contextual information, treating each position independently and potentially missing crucial sequence patterns. Therefore, it is crucial to address these limitations by exploring alternative coding methods to leverage the contextual and sequential nature of RNA sequences.

The contextual relationship has been a focus of increasing attention, as word embedding techniques (40–42) have proven to be effective frameworks for automatically encoding RNA sequences due to their syntactic, lexical, and semantic similarities to human language (43). On this basis, much effort has been devoted to the application of advanced NLP techniques in RNA-related problems (44–46). However, most existing research on the application of NLP techniques in RNA-related problems has relied on static embedding models such as Word2Vector (47), GloVe (41), Doc2Vector (42) and FastText (48). Similar to one-hot representation, static embedding methods maintain the same nucleotide encoding across all contexts, which leads to fail to capture the context-based dynamic semantic information of RNA sequences and therefore cannot represent RNA sequence features globally. Moreover, static embedding does not capture the underlying information of nucleotide sequences in different

cellular contexts, thus limiting the ability of dynamic prediction in other cellular conditions. To address this knowledge gap, we propose to adopt and customize the Bidirectional Encoder Representations from Transformers (BERT) model (49) to learn the context-dependent information of RNA sequences, which can generate robust expressions containing the global contextual information by pre-training with a large-scale unlabeled text corpus in a self-supervised fashion. In contrast to static embeddings where nucleotide embeddings remain the same regardless of the context, the primary advantage of dynamic embeddings lies in their ability to generate context-specific features in different nucleotide contexts. Moreover, considering the heterogeneity of transcripts under different cellular, tissue, or physiological conditions (e.g., normal or diseased), dynamic embeddings can overcome the limitation of a single representation for diverse contextual sequences, enriching the global features of the sequences and surpassing the performance bottleneck in dynamically predicting RBP binding sites. Therefore, by leveraging the advantages of the transformer model, we adopt the BERT model to encode RNA sequences in dynamic embedding vectors that then contain rich, global contextual semantic information for identifying dynamic RNA-binding events between different cell lines, tissues, or physiological conditions (normal or diseased)."

4. A better description of the candidate drug prediction analysis in Fig. S10 (also highlighted in Fig. 1e) would be helpful. How many neurological disease-associated genes (68 genes from Supplementary Note 6 or 28 genes from Fig. 8c) are used for the analysis?

Response: Thank you very much for the critical comments. We used the identified the 68 neurological disease-associated genes from **Supplementary Note 6 (now Supplementary Note 14)** for the candidate drug prediction analysis. We have clarified it and have followed your comments to make a more detailed descriptions about the candidate drug analyses, which is copied below for your easy reference:

Considering all 68 identified genes associated with neurological diseases as potential drug targets, we employed Enrichr to access the Drug Signatures Database (DSigDB) and conducted an analysis to identify potential drug molecules for the treatment of neurological diseases, using transcriptomic signatures. As shown in Supplementary Table 2, we extracted the top 10 chemical compounds that could be potential candidates for neurological diseases therapies based on the p -Value < 0.001; for instance, we identified ambroxol, a mucolytic compound predominantly prescribed for respiratory conditions like chronic bronchitis (50). Interestingly, recent studies have uncovered ambroxol's role as a pharmacological chaperone for glucosylceramidase (GCCase), an enzyme that is often found to have reduced activity in the brains of Parkinson's disease patients, potentially contributing to the degeneration of dopaminergic neurons (51). Given ambroxol's ability to augment GCCase levels, it holds promise as a potential disease-modifying intervention for Parkinson's disease (52–54). Moreover, ambroxol is also a candidate drug for Amyotrophic Lateral Sclerosis as it promotes and protects motor units and improves axonal plasticity (55).

We also extracted Metformin hydrochloride, a well-known antidiabetic medication commonly used to manage type 2 diabetes. However, emerging studies have suggested that metformin may have a role in the regulation of neurological systems (56). Indeed, research has indicated that Metformin has the potential to delay the progression of Alzheimer's disease by improving insulin sensitivity and reducing inflammatory responses (57, 58). Moreover, evidence suggests that metformin could provide neuroprotection by activating the AMP-activated protein kinase (AMPK) pathway, which could help alleviate symptoms of

Parkinson's disease, as in Katila et al. (59). Another compound we extracted was Carfilzomib, a specific proteasome inhibitor primarily used for the treatment of multiple myeloma, particularly in cases that are refractory to other treatment modalities or have relapsed (60). Extensive research has been conducted on the relationship between the proteasome and neurodegenerative diseases, such as Alzheimer's disease, Parkinson's disease, and Huntington's disease (61–63). Under these conditions, the abnormal accumulation of specific proteins could lead to neuronal damage and death (63, 64). Intriguingly, Carfilzomib functions as an epidermal growth factor receptor (EGFR) inhibitor, which has been reported to be associated with susceptibility to Parkinson's disease (65, 66).

In conclusion, our candidate drug prediction analysis identified several promising drugs with potential value for the treatment of neurological disorders. These findings warrant further investigation and provide possible insights into novel therapeutic strategies for those complex conditions.

Reviewer #3:

This is a new method for the prediction of protein-RNA interactions in the RNA sequences. It relies on a deep network and several input types. I appreciate a comprehensive nature of the empirical comparisons and the downstream analysis of the results. The ablation analysis is also a plus. However, I have a few concerns.

Response: Thank you very much for the critical and helpful comments. For the past 4 months, we have duly followed your comments and suggestions to carefully address all the concerns as elaborated in the following responses. In particular, we would like to thank you for your critical comments which have improved the manuscript in numerous ways. Thank you very much for your precious time and kind attention.

1. *PrismNet* was published over 2 years ago and may no longer represent state of the art in the field. Authors heavily rely on this approach as motivation for their own solution. More recent methods should be surveyed and considered. For instance, a recently released relevant framework, *PRIESSTESS* (<https://academic.oup.com/nar/article/50/19/e111/6677328>), should be investigated. Authors should compare against it and consider this type of approach as a useful resource to augment inputs to their network.

Response: Thank you very much for your insightful comments and for bringing the *PRIESSTESS* framework (37) to our attention. We have added and cited it to the reference list on the revised manuscript. We appreciate your suggestion to consider recent methods in our study, and agree that it is crucial to stay updated with the latest advancements in the field. Meanwhile, we have thoroughly investigated the *PRIESSTESS* framework and found that it indeed offers a unique approach for feature representation, which complements the methodology used in *PrismNet* and our proposed model. We appreciate the strengths of *PRIESSTESS*, particularly its two-step process in motif generation and aggregate model creation, which provides a comprehensive representation of RNA sequence and structure. Its citation has been added on the revised manuscript.

To address your concerns, we have added the *PRIESSTESS* method to our comparison group on those 261 RBP datasets across both static and dynamic cellular conditions. In addition, we have considered the potential of integrating aspects of the *PRIESSTESS* framework to augment the inputs of our network. It is copied below for your easy reference:

"..... Circos plot in Fig. 2a shows the overall experimental results of HDRNet compared to the other methods across the 261 RBP datasets. As depicted in this figure, we notice that HDRNet consistently outperformed other approaches. In particular, our model substantially enhanced the performance in identifying FMR1 and FXR2 binding sites in HEK293 cells; for instance, FMR1: PrismNet = 0.67 vs. HDRNet = 0.80; FXR2: PrismNet = 0.71 vs. HDRNet = 0.87. Moreover, we conducted additional analyses on the associated characteristics of FMR1 and FXR2, identifying their specific features (Supplementary Note 6, Supplementary Fig. 8). Meanwhile, we notice that HDRNet outperformed *PRIESSTESS* on almost all datasets. This discrepancy may arise from the fact that *PRIESSTESS* is highly dependent on the motif extracting process in the initial stage, and may not be able to identify salient patterns across the datasets with insufficient binding patterns, resulting in a limited training feature set. Moreover, the size of the dataset also plays a crucial role in influencing the efficiency of motif recognition. When the dataset is relatively small, it can potentially result in inaccuracies or the failure to recognize motifs, consequently diminishing

the predictive performance. In addition, the logistic regression model based on LASSO regularization within PRIESSTESS could yield suboptimal performance when confronted with nonlinear decision boundaries, while deep learning methodologies tend to exhibit more favorable outcomes. Although PRIESSTESS demonstrated suboptimal performance, its unique Motif extraction process could potentially enhance HDRNet, as illustrated in Supplementary Note 3 and Supplementary Figs. 4-5. In the violin plot of Fig. 2b, we observe that our proposed HDRNet performed more consistently and had better prediction performance on the majority of the datasets than the competing approaches. The reason for the improved performance may come not only from the self-supervised capability of the transformer that captures the global contextual and semantic information of the RNA sequences but also from the ability of the proposed network architecture to learn and transform long-range dependencies. We also provide the receiver operating characteristic (ROC) curve of the first four datasets that are plotted in Fig. 2c. The ROC curve analyses demonstrated that HDRNet had a higher true positive rate (TPR) compared to the other methods, indicating that HDRNet has a higher sensitivity for identifying RBP binding sites."

New insights by characterizing RNA binding events between different cell lines in a dynamic manner

..... Fig. 3a provides the heatmap of the dynamic prediction results, from which we can see that the proposed HDRNet outperformed the other methods for both K562 and HepG2 predictions while Deepbind performed the worst across the majority of datasets, with either the lowest AUC value or being incapable of making dynamic predictions. When predicting the RBP sites in K562 cells after training on HepG2 cells, the average AUC result of HDRNet was 0.79, which is 4% higher than PrismNet. In particular, the AUC results of RBM15 and XRN2 RBPs are 12% and 9% higher than PrismNet, respectively. Similarly, predicting HepG2 cells binding sites after training on K562 cells, the average AUC result of HDRNet was also 4% better than PrismNet. The results for RBM15, SF3B4 and SLTM also showed significant improvement over PrismNet, with the performance gains of 9%, 11% and 9%, respectively. Meanwhile, we also notice that HDRNet substantially outperformed PRIESSTESS for the dynamic tasks with higher AUC values while PRIESSTESS outperformed Deepbind, GraphProt, iDeep and DMSK in terms of dynamic prediction. Fig. 3b shows the predicted vs observed binding sites of HNRNPA1 on the MT-ND5 transcript. According to eCLIP, the MT-ND5 transcript contains 11 HNRNPA1 binding sites in HepG2 cells, and 6 binding sites in K562 cells. We found that HDRNet correctly predicted all 11 binding sites within the MT-ND5 transcript in HepG2 cells with no false positives, by using the model trained on K562 cells. In contrast, DeepBind and GraphProt, correctly predicted only 2 of the 11 sites, and iDeep and DMSK correctly predicted 4 of the 11 sites, and PrismNet and PRIESSTESS correctly predicted 7 of the 11 binding sites. Fig. 3c depicts the ROC curves of the first two datasets. Similar to the before AUC results, HDRNet had superior dynamic prediction than the other methods supported by better TPR values. "

For the last comment, we understood your concerns and have followed your comments to consider the PRIESSTESS method as the feature representation to augment inputs to our model. In the original study, PRIESSTESS was introduced as a universal RNA motif-finding/scanning strategy capable of identifying enriched RNA sequence and/or structure motifs. To incorporate the PRIESSTESS feature representation into our model, we employed a step-by-step approach. First, we used RNAfold to predict the secondary structure of each sequence and annotated them using the seven alphabets annotation described in PRIESSTESS. Then, we utilized the STREME tool to extract enriched motifs within each alphabets annotation, generating Position Frequency Matrices (PFMs) for each motif. For each annotation, we scanned the obtained PFMs on the data. To create an augmentation of HDRNet, we calculated the sum of the top four scores per sequence for each PFM. This augmentation, derived mainly from the

Fig. 2. HDRNet predicts RBP binding events more accurately than other state-of-the-art methods. (a) The Circos heatmap of the respective AUC scores of HDRNet vs. other methods, including PrismNet, PRIESSTESS, DMSK, iDeep, GraphProt, and DeepBind on all 261 RBP datasets. (b) The violin plot of the overall results of HDRNet and the other methods. (c) The ROC curve on the first 4 RBP datasets using HDRNet and the other methods. (d) t-SNE clustering results of the output features of HDRNet and the other baseline methods.

Fig. 3. HDRNet successfully performs dynamic RBP binding predictions in both K562 cells and HepG2 cells. (a) The heatmap of the respective dynamic prediction AUC scores of HDRNet and other baseline methods, including PrismNet, PRIESSTESS, iDeep, DMSK, GraphProt, and deepbind. (b) Predicted vs observed binding sites of HNRNPA1 on the MT-ND5 transcript. Green/Brown, observed binding sites in K562/HepG2 cells by eCLIP, used as the training/ground truth reference data; Blue and red indicate, respectively, true positive and false positive predictions in HepG2 cells, based on the models trained using K562 data. (c) The dynamic prediction ROC curve of the first two RBP datasets using HDRNet and the other methods. (d) t-SNE clustering results of the output *features* of HDRNet and the other baseline methods in a dynamic manner.

sequence-structure motifs, was integrated into the icSHAPE feature to enhance the expression of the structure profiles. Specifically, we employed a fully connected network to adjust the feature length to match with that of icSHAPE and then summed them together.

After that, we conducted experiments to verify whether the augmented features can further impact the performance of HDRNet. As depicted in **Supplementary Fig. 4**, our results demonstrate that the average performance of HDRNet after feature enhancement was equal to the original features in static prediction task and showed a very slight decrease in the dynamic prediction task, as illustrated in **Supplementary Fig. 5**. However, it is worth noting that we observed significant improvements in the performance of the model after data augmentation on certain datasets. For instance, in the C17ORF85_HEK293 and TNRC6A_HEK293 datasets, the performance of static prediction was enhanced by 2.2% and 2.4%, respectively. Additionally, we also observed a 2% performance improvement in the dynamic prediction task on the DDX52 dataset. These findings support the notion that the motif extraction and scoring based on the multi-sequence-structure annotation approach utilized by PRIESSTESS significantly contribute to the identification of RNA-binding protein (RBP) binding sites and can be an effective approach to enhance the performance of HDRNet when the binding site motif patterns of interest are well-conserved.

Supplementary Fig. 4. Performance comparison of HDRNet and HDRNet with feature augmentation on static prediction task.

Supplementary Fig. 5 Performance comparison of HDRNet and HDRNet with feature augmentation on dynamic prediction task.

In addition, we also emphasize, present, and add explicit citations to the PRIESSTESS algorithm in Section **Competing methods**. It is copied below for your easy reference:

Competing methods

- PRIESSTESS (37) (<https://github.com/kaitlin309/PRIESSTESS>), is a universal and recent RNA motif-finding/scanning strategy capable of identifying enriched RNA sequence and/or structure motifs. PRIESSTESS consists of two steps. The first step generates a large collection of enriched motifs encompassing both RNA sequence and structure. The second step produces an aggregate model, which combines the motif scores into a single value, and gauges the relative importance of each motif.

2. *While the proposed predictive architecture is an improvement over PrismNet (with the above caveat), the use of BERT, deep residual networks and self-attention is not particularly innovative. The literature is already populated with similar "end-to-end" solutions. The authors need to provide much more complete accounting of modern methods and more compelling explanation of the novel aspects of their method.*

Response: Thank you very much for your critical comments. We agree with you. Your comments have provided us with an opportunity to further refine our work and better articulate its unique contributions. We have followed your comments to provide much more complete accounting of modern methods and more compelling explanation of the novel aspects. Indeed, numerous "end-to-end" methods have been developed to identify RNA-binding protein (RBP) binding sites. However, those methods are often designed and optimized for specific cellular conditions, limiting their broader applicability. We believe the distinct advantage of our proposed HDRNet architecture is its capacity to predict **dynamic** RBP binding events **across different cell lines and tissues**. Therefore, while BERT, deep residual networks, and self-attention are prevalent tools, the novelty of our work lies in their particular amalgamation and adaption. In addition, through the substantial revisions with 3 independent reviewers in the past four months, we are also confident that the resultant experimental results in this study can also serve as an independent and comprehensive benchmark beyond the methodology contributions.

In terms of **technical contribution**, the key advantage of HDRNet is its dynamic identification and interpretation of RNA-binding protein (RBP) binding sites. This dynamic approach sets it apart from many other models and forms the core of its novelty. To learn the potential representation of the sequence for the protein–RNA interactions effectively, we have narrowed our scope to a deep learning model and found that PrismNet is also a deep learning model to predict dynamic RBP binding events. In the following, we have endeavored to highlight the unique aspects and advantages of HDRNet by comparing it directly with PrismNet. The distinguishing features and strengths of HDRNet, in contrast to PrismNet, are outlined in the following sections:

1. Firstly, a critical limitation of PrismNet is its reliance on traditional static one-hot encoding. This static encoding fails to capture the dynamic semantic information of RNA sequences and their global features. Moreover, it struggles to capture basic information of nucleotide sequences under varying cellular or tissue contexts, hindering dynamic prediction abilities. To address this, HDRNet utilizes dynamic global context representation, encoding nucleotides differently depending on the context. This dynamic representation can reflect the context relationships of nucleotides and further enhance the

ability of HDRNet to recognize the specific RBP binding sites. Moreover, **no existing research employs dynamic context representation as a foundation for dynamic prediction tasks**, positioning HDRNet at the forefront of innovation in this area, as discussed in **Supplementary Note 19, Fig. 5**.

2. Secondly, traditional deep learning algorithms, including PrismNet, often face a discrepancy in the dimensions of sequence and secondary structure features. These models typically concatenate these two disparate feature types at the input stage, a strategy that can inadvertently lead to an uneven feature distribution. This imbalance can obstruct the model's ability to fully comprehend and learn from both the sequence and secondary structure features, hindering the model's ability to grasp the intricacies of RBP binding sites effectively. To rectify this, **HDRNet implements a dimension unifying architecture at the onset**, harmonizing the dimensions of sequence and structural features and providing a balanced feature representation. Indeed, HDRNet employs convolutional neural network (CNN) blocks with a variety of kernel sizes to achieve this balance. As an adaptive mechanism, these CNN blocks effectively equalize the dimensionality of the feature sets. This results in a uniformity between the sequence and physicochemical features, allowing the model to assimilate and learn the characteristics of both sets of features more efficiently, as depicted in **Fig. 6 and Supplementary Fig. 6**.
3. In contrast to PrismNet, which employs a basic residual structure to process the fused features of sequences and secondary structures, **HDRNet introduces a novel hierarchical multiscale residual network (HMRN) structure**. This approach potentially resolves the challenge faced by PrismNet, which often struggles to accurately identify the combined patterns of sequences and structures due to its reliance on a basic residual structure. Our HMRN structure incorporates a new dual-path parallel structure that separately extracts the combined patterns of sequences and structures, ensuring that HDRNet is able to capture the specific binding preferences of an individual RBP from the sequence and structure perspective, respectively. In addition, we also employ a multi-scale CNN design into HDRNet, using CNN kernels of varying sizes, further enhancing its ability to dynamically identify RBP binding sites, as demonstrated in **Supplementary Fig. 6**.
4. Finally, like many deep learning algorithms, PrismNet merely averages the features in the final prediction stage and uses a fully connected layer to generate the final prediction score. This leads to a blending of global features, potentially hindering the extraction of the most significant binding regions. Here, **we propose the Deep Protein-RBP Binding Predictor (DPRBP) in HDRNet** to address this issue. DPRBP is a pyramid-shaped deep structure where the length of the nucleotide sequence decreases at each successive layer, retaining the nucleotide fragments that most significantly influence the accurate prediction of RBP binding sites, as illustrated in **Fig. 6**. By progressively shortening the sequence length at each layer, DPRBP can preserve the most significant local regions, thus providing HDRNet with a robust mechanism for precisely capturing RBP binding preferences. This architecture distinguishes HDRNet from other deep learning algorithms, offering an advanced solution for predicting RBP binding sites.

As demonstrated in our experiments, this model formulation allows us to accomplish the following tasks successfully:

1. **Static cellular protein–RNA interaction identification.** HDRNet utilizes dynamic global context representation and icSHAPE secondary structure information to generate robust binding site representations through a novel hierarchical network architecture and deep protein-RNA binding predictor. In a comprehensive comparison across 261 benchmark datasets using several state-of-the-art baseline algorithms, HDRNet outperforms its counterparts in static prediction tasks

within specific cellular contexts. Moreover, through visualizing the latent features learned by the model, HDRNet can reveal distinct data clusters, highlighting its robustness, as elucidated in **Fig. 2**.

2. **Dynamic prediction of cellular protein–RNA interaction across different cell lines.** HDRNet’s foremost advantage is its superior capability for dynamic prediction of RBP binding sites, through a comprehensive learning of RBP binding characteristics from the dynamic global contextual representation by the unique architecture of HDRNet. We thoroughly validated HDRNet’s dynamic predictive ability across datasets from various cell lines of 68 RBPs. Importantly, HDRNet can also generate robust latent features for RBP binding site data from other cell lines during the dynamic prediction process, which facilitates HDRNet to accurately discern the binding properties of RBPs. Moreover, HDRNet maintains its superior performance consistently across RBPs of varying expression levels and those targeting different expression levels, as shown in **Fig. 3, Supplementary Figs. 9-10**.
3. **HDRNet’s capacity for dynamic prediction is not limited to cell lines.** HDRNet stands as the **first** tool capable of predicting RBP binding sites across in vivo tissues. We applied HDRNet on the human brain, adrenal gland and mice muscle dataset and demonstrate a remarkable ability to accurately discern RBP binding sites in both normal and diseased conditions. Additionally, HDRNet exhibits a unique adaptive ability, enabling it to utilize models trained on cell line data to predict the binding sites on tissue data. The dynamic prediction capability of HDRNet on tissue data provides new insights for studying the regulation of RBP in vivo in a dynamic manner, showcasing its unique value in this field, as depicted in **Fig. 4, Supplementary Figs. 11-12**.
4. **HDRNet exhibits explainable interpretability.** HDRNet can initially discern potential RBP binding motifs by leveraging BERT as the feature encoder with self-attention mechanism. It then refines this understanding by capturing precise RBP binding preferences from the dynamic context embeddings. This two-step approach proficiently illuminates significant sequence regions indicative of RBP binding characteristics or disease associations. For example, HDRNet accurately demarcates the CUG-rich region of MBNL in human brain dataset and the G-rich region with G-quadruplex structure of HNRNPU in human adrenal gland dataset. Furthermore, through the statistical analysis of HDRNet-identified salient regions and visualization of salient maps, we gleaned deeper insights into the binding preferences of FMR1 and FXR2. With its robust ability to correlate computational predictions with biological understanding, the advanced interpretability of HDRNet holds immense potential for providing novel insights and guiding future research in RNA biology and disease, as highlighted in **Fig. 7, Supplementary Figs. 8, 11-15, Supplementary Table 1**.
5. **HDRNet also exhibits the capability to notice and highlight potential mutation spots.** HDRNet can identify high-attention binding regions enriched with mutable nucleotide sites. After comparing with database entries and manually mutating the corresponding nucleotides, HDRNet’s local binding behavior identification exhibits significant alterations, thereby underscoring its capacity to detect potential mutant binding sites (**Fig. 8, Supplementary Figs. 15-17**). This unique attribute of HDRNet aids in the study of the role of mutations in RBP binding events, thus contributing to a broader understanding of disease pathogenesis and progression. Moreover, we conducted an in-depth study from a transcriptomic viewpoint on the interrelationship between TDP-43 binding genes and human diseases, and performed a series of genomic analyses. This led to the successful identification of potential candidate drugs for the treatment of neurodegenerative diseases, as elucidated in **Supplementary Figs. 18-23, Supplementary Table 2**.

Lastly, it is essential to highlight that HDRNet is the premier deep learning model that utilizes dynamic context embedding representation to tackle dynamic prediction tasks across various cell lines. Furthermore, leveraging the valuable insights received from our reviewers on tissue prediction, we wish to underline that HDRNet has emerged as the first reliable tool for **dynamic prediction between normal and diseased conditions in in vivo tissue data**, and **the trained HDRNet model on cell line data is sufficient to predict dynamic binding behaviour in tissue**, which are extremely rare in the current field of dynamic predictions of RBP binding sites (to the best of our knowledge). These analyses further confirm the effectiveness and robustness of HDRNet. In an effort to assist the broader research community, we provide an accessible HDRNet webserver at <http://www.aibio-lab.com:5050/> and open-source HDRNet code at <https://github.com/zhuhr213/HDRNet>. These resources will facilitate users in predicting RBP sites and identifying significant binding regions.

In addition, we also provide a more comprehensive overview of modern methods including self-attention, BERT and deep residual networks, for protein-RNA interactions in **Supplementary Note 19** to differentiate our work from existing studies. It is copied below for your easy reference:

Self-attention mechanisms have gained popularity across various fields for their ability to highlight important features in sequences while downplaying irrelevant ones. In the context of protein-RNA interactions, the self-attention model enable the focus on key nucleotides and their interactions. Several end-to-end models have already incorporated self-attention mechanisms; for instance, Wang et al. proposed SA-NET (67), which encodes nucleotide sequences into 4-mer tokens and identifies binding sites through a self-attention layer. Similarly, Pan et al. introduced CRMSNet (68), a deep learning model that leverages convolution and residual multi-head self-attention blocks to predict RBPs' binding sites on RNA sequence. These methods excessively rely on the self-attention mechanism, leading to an insufficient diversity of feature extraction and a high demand for data for training (69). To address this limitation, BERT, short for "Bidirectional Encoder Representations from Transformers", was proposed by designing a deep learning model based on self-attention mechanism. Its strength lies in understanding the contextual relationships of words within sentences by analyzing the text data bidirectionally. Given the similarity between nucleotide sequences and textual sequences, BERT has been adopted for the identification of RBP binding sites. For example, Yamada et al. proposed BERT-RBP as a model to predict RNA-RBP interactions by adapting the BERT architecture pre-trained on a human reference genome (70). The principal shortcoming of this kind of BERT-based method lies in the fact that its fine-tuning process cannot significantly optimize the parameters of the BERT model, leading to inaccuracies in capturing the binding characteristics of RBPs, particularly under dynamic cellular conditions. Nevertheless, the pre-training process of BERT, which involves learning from a vast amount of transcriptome data, empowers it to comprehensively grasp the contextual relationships within the genome. Therefore, such advantages make BERT an attractive option to be leveraged as a generative model for creating dynamic contextual representations of nucleotide sequences. Several end-to-end deep learning models based on this dynamic contextual representations have been developed, including HCRNet (71), CircSSNN (72), and JLCRB (73). However, most of these end-to-end methods were designed to identify RBP binding sites on circular RNAs under a singular cellular condition. To the best of our knowledge, the dynamic contextual representation of nucleotide sequences has not yet been characterized for dynamic prediction of RBP binding sites on linear RNA (across different cell lines or in vivo tissues), which motivates our further research in this field.

ResNets, initially introduced in the field of computer vision (74), have brought about a revolutionary impact. Now, these ResNets are applied to the field of RBP binding sites identification. ResNets can solve the vanishing gradient problem common in deep neural networks, allowing for the successful training of very deep networks. Numerous end-to-end models have adopted the residual structure to predict RBP binding sites, including CRBPDL (75), CRMSS (76), iCircRBP-DHN (44), and CircSSNN. Unfortunately, these methods are primarily focused on predicting RBP binding sites on circular RNAs in specific cell lines, limiting their broader applicability. Furthermore, one common aspect in these methods, including PrismNet, is the use of a basic ResNet structure, which integrates multiple features before inputting them into the residual network. This basic architecture possesses a risk as the amalgamation of features might lead to an inability to distinguish between sequence-specific and structural binding characteristics, thus restricting the performance of RBP binding site identification. To mitigate this risk, we propose a hierarchical multi-scale residual structure, comprising a parallel, dual-pathway framework that can extract the latent features of RNA sequences and secondary structures separately, ensuring the optimal performance in our unique dynamic prediction task. Moreover, the multi-scale structure enables HDRNet to learn binding patterns across varying sequence lengths.

3. *The survey of tools that is included in the introduction has missed an entire large community that performs these predictions on the protein side, by predicting RNA-binding proteins and RNA-binding amino acids.*

Response: Thank you very much for the suggestions. We appreciate your suggestion on the tools related to the prediction of RNA-binding proteins and RNA-binding amino acids in the survey of tools included in the introduction. Therefore, we have followed your comments to carefully investigated the recent literature on the prediction of RBP-RNA binding from a protein perspective to fill the absence of this part in the **Introduction**. It is copied below for your easy reference:

Introduction

..... Thanks to the development of cross-linked immunoprecipitation sequencing technology (21, 77), many RBP-RNA binding targets have been uncovered (78, 79), enabling us to develop effective data-driven computational methods (80). These methods, can be broadly classified into two categories (81), either predicting RNA-binding sites on the protein surface (82) or modeling the preferred RNA sequences of RNA-binding proteins (83). From the protein perspective, several computational tools have been developed to predict the RBP-RNA binding sites at the protein level. For instance, SCRIBER, uses predictions of binding residues for several partner types to effectively reduce cross-prediction of the output protein-binding residues, combining novel and previously used input types (84), while aPRBind, developed by Liu et al., combines protein sequence and structural features for RNA-binding residue prediction (85). DRNAPred (86), a fast sequence-based method that accurately predicts and discriminates DNA- and RNA-binding residues was proposed by Yan et al., with regression that penalizes cross-predictions, and a two-layered architecture. However, these statistical or machine learning-based algorithms often encounter performance limitations as the size of the dataset continues to grow. This is primarily due to their inability to effectively capture complex patterns and relationships in large-scale data, which can lead to unsatisfactory predictive performance. Recently, to address such limitations, deep learning methods have developed; for instance, DeepSite employs a 3D deep convolutional neural network to predict the binding site using protein structure (87). Xia et al. proposed GraphBind, an end-to-end graph

neural network that uses hierarchical graph neural networks to identify nucleic-acid-binding residues on proteins (88). Zhang et al. presented DeepDISOBind(89), a deep multi-task architecture that accurately predicts DNA-, RNA- and protein-binding regions from protein sequences. Most recently, Lam et al. introduced NucleicNet, a deep learning model that predicts the binding preference of RNA backbone constituents and different bases from local physicochemical characteristics of the protein structure surface (90). These methods have collectively enhanced our understanding of the binding properties of RBPs at the protein surface level.

In parallel, the establishment of links between RBPs and their targets from the perspective of RNA sequences enables understandings on the regulation mechanism. A variety of efforts have been developed to address it in identical cellular conditions, referred to as static protein-RNA interactions; for instance, Deepbind (33) was developed to understand RBP binding preferences in RNA sequence data using a deep neural network (DNN). Ilan et al. developed DLPRB (34), a new DNN approach based on convolutional neural networks (CNN) and recurrent neural networks (RNN) for learning intrinsic RBP binding preferences and predicting novel interactions. Pan et al. developed iDeep (35), a hybrid convolutional neural network and deep belief network-based model to predict the RBP interaction sites and motifs on RNAs. Daniel et al. proposed GraphProt (36) that integrates sequence and computationally-predicted RNA secondary structure information into graph-kernel features. Laverty et al. introduced PRIESSTESS (37), a universal RNA motif-finding/scanning strategy capable of identifying enriched RNA sequence and/or structure motifs that are subsequently reduced to a set of core motifs by logistic regression with LASSO regularization. However, these methods only account for a specific cellular condition and are hence limited in their ability to predict RNA-protein interactions in other cell lines while considering the dynamic contexts.

Indeed, the binding behavior of RBP-RNA interactions has been demonstrated to be dynamic in different cell types, as it is influenced by different cellular or tissue environments (38). In other words, the binding sites of the RBP that are exclusively present in specific cells or tissues can be designated as dynamic protein-RNA interactions. Recently, a new deep learning-based method called PrismNet (26), was developed to accurately predict dynamic RBP binding in various cellular conditions by integrating in vivo experimental RNA structure information (39). Nonetheless, unfortunately, we conceive that there are still room for improvements; for instance, the architectural design of the deep neural network with in PrismNet can still be investigated in the context of neural network architecture search, indicating the potential for further improvements in RBP-RNA binding prediction. In addition, the one-hot encoding representation in PrismNet exhibits limitations that may hinder its ability to capture RNA features globally. Furthermore, the heterogeneity of transcripts results in widely sequential relationships across different contexts, such as various cell lines, tissues, and normal or disease conditions. The one-hot encoding may overlook this contextual information, treating each position independently and potentially missing crucial sequence patterns. Therefore, it is crucial to address these limitations by exploring alternative coding methods to leverage the contextual and sequential nature of RNA sequences.

The contextual relationship has been a focus of increasing attention, as word embedding techniques (40–42) have proven to be effective frameworks for automatically encoding RNA sequences due to their syntactic, lexical, and semantic similarities to human language (43). On this basis, much effort has been devoted to the application of advanced NLP techniques in RNA-related problems (44–46). However, most existing research on the application of NLP techniques in RNA-related problems

has relied on static embedding models such as Word2Vector (47), GloVe (41), Doc2Vector (42) and FastText (48). Similar to one-hot representation, static embedding methods maintain the same nucleotide encoding across all contexts, which leads to fail to capture the context-based dynamic semantic information of RNA sequences and therefore cannot represent RNA sequence features globally. Moreover, static embedding does not capture the underlying information of nucleotide sequences in different cellular contexts, thus limiting the ability of dynamic prediction in other cellular conditions. To address this knowledge gap, we propose to adopt and customize the Bidirectional Encoder Representations from Transformers (BERT) model (49) to learn the context-dependent information of RNA sequences, which can generate robust expressions containing the global contextual information by pre-training with a large-scale unlabeled text corpus in a self-supervised fashion. In contrast to static embeddings where nucleotide embeddings remain the same regardless of the context, the primary advantage of dynamic embeddings lies in their ability to generate context-specific features in different nucleotide contexts. Moreover, considering the heterogeneity of transcripts under different cellular, tissue, or physiological conditions (e.g., normal or diseased), dynamic embeddings can overcome the limitation of a single representation for diverse contextual sequences, enriching the global features of the sequences and surpassing the performance bottleneck in dynamically predicting RBP binding sites. Therefore, by leveraging the advantages of the transformer model, we adopt the BERT model to encode RNA sequences in dynamic embedding vectors that then contain rich, global contextual semantic information for identifying dynamic RNA-binding events between different cell lines, tissues, or physiological conditions (normal or diseased)."

Minor comments:

1. *On a minor note, quality of writing needs substantial improvements. There are grammatical errors (even in the abstract), some passages/sentences are poorly phrased, problems with articles, and other issues. The article should be carefully copy-edited by a skilled native speaker.*

Response: Thank you very much for the helpful comments. We apologize for the grammatical errors and typos. To correct those errors, we have asked a native English speaker to proofread the revised manuscript. In addition, we (the first author and the corresponding authors) have proofread the manuscript for multiple rounds again. English grammatical errors and typos have been fixed to our best effort.

References

1. Andreas R Gruber, Ronny Lorenz, Stephan H Bernhart, Richard Neuböck, and Ivo L Hofacker. The vienna rna websuite. *Nucleic acids research*, 36(suppl_2):W70–W74, 2008.
2. Silvi Rouskin, Meghan Zubradt, Stefan Washietl, Manolis Kellis, and Jonathan S Weissman. Genome-wide probing of rna structure reveals active unfolding of mrna structures in vivo. *Nature*, 505(7485):701–705, 2014.
3. Meghan Zubradt, Paromita Gupta, Sitara Persad, Alan M Lambowitz, Jonathan S Weissman, and Silvi Rouskin. Dms-mapseq for genome-wide or targeted rna structure probing in vivo. *Nature methods*, 14(1):75–82, 2017.
4. Tianqi Chen and Carlos Guestrin. Xgboost: A scalable tree boosting system. In *Proceedings of the 22nd acm sigkdd international conference on knowledge discovery and data mining*, pages 785–794, 2016.
5. Leo Breiman. Random forests. *Machine learning*, 45:5–32, 2001.
6. David W Hosmer Jr, Stanley Lemeshow, and Rodney X Sturdivant. *Applied logistic regression*, volume 398. John Wiley & Sons, 2013.
7. Anil K Jain, Jianchang Mao, and K Moidin Mohiuddin. Artificial neural networks: A tutorial. *Computer*, 29(3):31–44, 1996.
8. Pierre Geurts, Damien Ernst, and Louis Wehenkel. Extremely randomized trees. *Machine learning*, 63:3–42, 2006.
9. Bianca Zadrozny and Charles Elkan. Transforming classifier scores into accurate multiclass probability estimates. In *Proceedings of the eighth ACM SIGKDD international conference on Knowledge discovery and data mining*, pages 694–699, 2002.

10. Hajer Kamel, Dahir Abdulah, and Jamal M Al-Tuwaijari. Cancer classification using gaussian naive bayes algorithm. In *2019 International Engineering Conference (IEC)*, pages 165–170. IEEE, 2019.
11. F. Pedregosa, G. Varoquaux, A. Gramfort, V. Michel, B. Thirion, O. Grisel, M. Blondel, P. Prettenhofer, R. Weiss, V. Dubourg, J. Vanderplas, A. Passos, D. Cournapeau, M. Brucher, M. Perrot, and E. Duchesnay. Scikit-learn: Machine learning in Python. *Journal of Machine Learning Research*, 12:2825–2830, 2011.
12. Tianqi Chen and Carlos Guestrin. Xgboost: A scalable tree boosting system. In *Proceedings of the 22nd acm sigkdd international conference on knowledge discovery and data mining*, pages 785–794, 2016.
13. Leo Breiman. Random forests. *Machine learning*, 45(1):5–32, 2001.
14. Fabian Pedregosa, Gaël Varoquaux, Alexandre Gramfort, Vincent Michel, Bertrand Thirion, Olivier Grisel, Mathieu Blondel, Peter Prettenhofer, Ron Weiss, Vincent Dubourg, et al. Scikit-learn: Machine learning in python. *the Journal of machine Learning research*, 12:2825–2830, 2011.
15. Pan Li, Ruoyao Shi, and Qiangfeng Cliff Zhang. icshape-pipe: A comprehensive toolkit for icshape data analysis and evaluation. *Methods*, 178:96–103, 2020.
16. Dalen Chan, Chao Feng, and Robert C Spitalé. Measuring rna structure transcriptome-wide with icshape. *Methods*, 120:85–90, 2017.
17. Lu Chen, Howard Y Chang, and Steven E Artandi. Analysis of rna conformation in endogenously assembled rnps by icshape. *STAR protocols*, 2(2):100477, 2021.
18. Ryan A Flynn, Qiangfeng Cliff Zhang, Robert C Spitalé, Byron Lee, Maxwell R Mumbach, and Howard Y Chang. Transcriptome-wide interrogation of rna secondary structure in living cells with icshape. *Nature protocols*, 11(2):273–290, 2016.
19. Qing-Jun Luo, Jinsong Zhang, Pan Li, Qing Wang, Yue Zhang, Biswajoy Roy-Chaudhuri, Jianpeng Xu, Mark A Kay, and Qiangfeng Cliff Zhang. Rna structure probing reveals the structural basis of dicer binding and cleavage. *Nature communications*, 12(1):3397, 2021.
20. Damian Szklarczyk, Annika L Gable, David Lyon, Alexander Junge, Stefan Wyder, Jaime Huerta-Cepas, Milan Simonovic, Nadezhda T Doncheva, John H Morris, Peer Bork, et al. String v11: protein–protein association networks with increased coverage, supporting functional discovery in genome-wide experimental datasets. *Nucleic acids research*, 47(D1):D607–D613, 2019.
21. Eric L Van Nostrand, Gabriel A Pratt, Alexander A Shishkin, Chelsea Gelboin-Burkhart, Mark Y Fang, Balaji Sundararaman, Steven M Blue, Thai B Nguyen, Christine Surka, Keri Elkins, et al. Robust transcriptome-wide discovery of rna-binding protein binding sites with enhanced clip (eclip). *Nature methods*, 13(6):508–514, 2016.
22. Filippo Tamanini, Rob Willemsen, Leontine van Unen, Carola Bontekoe, Hans Galjaard, Ben A Oostra, and André T Hoogveen. Differential expression of fmr1, fxr1 and fxr2 proteins in human brain and testis. *Human molecular genetics*, 6(8):1315–1322, 1997.
23. Yan Zhang, J Patrick O'Connor, Mikiko C Siomi, Sudha Srinivasan, Amalia Dutra, Robert L Nussbaum, and Gideon Dreyfuss. The fragile x mental retardation syndrome protein interacts with novel homologs fxr1 and fxr2. *The EMBO Journal*, 14(21):5358–5366, 1995.
24. Corinne M Spencer, Ekaterina Serysheva, Lisa A Yuva-Paylor, Ben A Oostra, David L Nelson, and Richard Paylor. Exaggerated behavioral phenotypes in fmr1/fxr2 double knockout mice reveal a functional genetic interaction between fragile x-related proteins. *Human molecular genetics*, 15(12):1984–1994, 2006.
25. Carola JM Bontekoe, Kellie L Mcllwain, Ingeborg M Nieuwenhuizen, Lisa A Yuva-Paylor, Anna Nellis, Rob Willemsen, Zhe Fang, Laura Kirkpatrick, Cathy E Bakker, Robin McAninch, et al. Knockout mouse model for fxr2: a model for mental retardation. *Human molecular genetics*, 11(5):487–498, 2002.
26. Lei Sun, Kui Xu, Wenzhe Huang, Yucheng T Yang, Pan Li, Lei Tang, Tuanlin Xiong, and Qiangfeng Cliff Zhang. Predicting dynamic cellular protein–rna interactions by deep learning using in vivo rna structures. *Cell research*, 31(5):495–516, 2021.
27. Eric L Van Nostrand, Peter Freese, Gabriel A Pratt, Xiaofeng Wang, Xintao Wei, Rui Xiao, Steven M Blue, Jia-Yu Chen, Neal AL Cody, Daniel Dominguez, et al. A large-scale binding and functional map of human rna-binding proteins. *Nature*, 583(7818):711–719, 2020.
28. Marianne Goodwin, Apoorva Mohan, Ranjan Batra, Kuang-Yung Lee, Konstantinos Charizanis, Francisco José Fernández Gómez, Sabiha Eddarkaoui, Nicolas Sergeant, Luc Bué, Takashi Kimura, et al. Mbnl sequestration by toxic rnas and rna misprocessing in the myotonic dystrophy brain. *Cell reports*, 12(7):1159–1168, 2015.
29. Yumin Zhu, Gang Xu, Yucheng T Yang, Zhiyu Xu, Xinduo Chen, Binbin Shi, Daoxin Xie, Zhi John Lu, and Pengyuan Wang. Postar2: deciphering the post-transcriptional regulatory logics. *Nucleic acids research*, 47(D1):D203–D211, 2019.
30. Eric T Wang, Neal AL Cody, Sonali Jog, Michela Biancolella, Thomas T Wang, Daniel J Treacy, Shujun Luo, Gary P Schroth, David E Housman, Sita Reddy, et al. Transcriptome-wide regulation of pre-mrna splicing and mrna localization by muscleblind proteins. *Cell*, 150(4):710–724, 2012.
31. Chantal Sellier, Franck Freyermuth, Ricardos Tabet, Tuan Tran, Fang He, Frank Ruffenach, Violaine Alunni, Herve Moine, Christelle Thibault, Adeline Page, et al. Sequestration of drosha and dgcr8 by expanded cgg rna repeats alters microRNA processing in fragile x-associated tremor/ataxia syndrome. *Cell reports*, 3(3):869–880, 2013.
32. Aaron R Haeussler, Christopher J Donnelly, Goran Periz, Eric AJ Simko, Patrick G Shaw, Min-Sik Kim, Nicholas J Maragakis, Juan C Troncoso, Akhilesh Pandey, Rita Sattler, et al. C9orf72 nucleotide repeat structures initiate molecular cascades of disease. *Nature*, 507(7491):195–200, 2014.
33. Babak Alipanahi, Andrew Delong, Matthew T Weirauch, and Brendan J Frey. Predicting the sequence specificities of dna-and rna-binding proteins by deep learning. *Nature biotechnology*, 33(8):831–838, 2015.
34. Ilan Ben-Bassat, Benny Chor, and Yaron Orenstein. A deep neural network approach for learning intrinsic protein-rna binding preferences. *Bioinformatics*, 34(17):i638–i646, 2018.
35. Xiaoyong Pan and Hong-Bin Shen. Rna-protein binding motifs mining with a new hybrid deep learning based cross-domain knowledge integration approach. *BMC bioinformatics*, 18(1):1–14, 2017.
36. Daniel Maticzka, Sita J Lange, Fabrizio Costa, and Rolf Backofen. Graphprot: modeling binding preferences of rna-binding proteins. *Genome biology*, 15(1):1–18, 2014.
37. Kaitlin U Laverty, Arttu Jolma, Sara E Pour, Hong Zheng, Debashish Ray, Quaid Morris, and Timothy R Hughes. Priesstess: interpretable, high-performing models of the sequence and structure preferences of rna-binding proteins. *Nucleic Acids Research*, 50(19):e1111–e1111, 2022.
38. Mallory A Freeberg, Ting Han, James J Moresco, Andy Kong, Yu-Cheng Yang, Zhi John Lu, John R Yates, and John K Kim. Pervasive and dynamic protein binding sites of the mrna transcriptome in *saccharomyces cerevisiae*. *Genome biology*, 14:1–20, 2013.
39. Robert C Spitalé, Ryan A Flynn, Qiangfeng Cliff Zhang, Pete Crisalli, Byron Lee, Jong-Wha Jung, Hannes Y Kuchelmeister, Pedro J Batista, Eduardo A Torre, Eric T Kool, et al. Structural imprints in vivo decode rna regulatory mechanisms. *Nature*, 519(7544):486–490, 2015.
40. Simon Du, Jason Lee, Yuandong Tian, Aarti Singh, and Barnabas Poczos. Gradient descent learns one-hidden-layer cnn: Don't be afraid of spurious local minima. In *International Conference on Machine Learning*, pages 1339–1348. PMLR, 2018.
41. Jeffrey Pennington, Richard Socher, and Christopher D Manning. Glove: Global vectors for word representation. In *Proceedings of the 2014 conference on empirical methods in natural language processing (EMNLP)*, pages 1532–1543, 2014.

42. Quoc Le and Tomas Mikolov. Distributed representations of sentences and documents. In *International conference on machine learning*, pages 1188–1196. PMLR, 2014.
43. Sai Zhang, Jingtian Zhou, Hailin Hu, Haipeng Gong, Ligong Chen, Chao Cheng, and Jianyang Zeng. A deep learning framework for modeling structural features of rna-binding protein targets. *Nucleic acids research*, 44(4):e32–e32, 2016.
44. Yuning Yang, Zilong Hou, Zhiqiang Ma, Xiangtao Li, and Ka-Chun Wong. icircrbp-dhn: identification of circrna-rbp interaction sites using deep hierarchical network. *Briefings in Bioinformatics*, 22(4):bbab274, 2021.
45. Hui Li, Zhaohong Deng, Haitao Yang, Xiaoyong Pan, Zhisheng Wei, Hong-Bin Shen, Kup-Sze Choi, Lei Wang, Shitong Wang, and Jing Wu. circrna-binding protein site prediction based on multi-view deep learning, subspace learning and multi-view classifier. *Briefings in Bioinformatics*, 23(1):bbab394, 2022.
46. Zhengfeng Wang and Xiujuan Lei. Prediction of rbp binding sites on circrnas using an lstm-based deep sequence learning architecture. *Briefings in Bioinformatics*, 22(6):bbab342, 2021.
47. Tomas Mikolov, Kai Chen, Greg Corrado, and Jeffrey Dean. Efficient estimation of word representations in vector space. *arXiv preprint arXiv:1301.3781*, 2013.
48. Piotr Bojanowski, Edouard Grave, Armand Joulin, and Tomas Mikolov. Enriching word vectors with subword information. *Transactions of the association for computational linguistics*, 5:135–146, 2017.
49. Jacob Devlin, Ming-Wei Chang, Kenton Lee, and Kristina Toutanova. Bert: Pre-training of deep bidirectional transformers for language understanding. *arXiv preprint arXiv:1810.04805*, 2018.
50. Mario Malerba and Beatrice Ragnoli. Amroxol in the 21st century: pharmacological and clinical update. *Expert opinion on drug metabolism & toxicology*, 4(8):1119–1129, 2008.
51. Matthew E Gegg and Anthony HV Schapira. The role of glucocerebrosidase in parkinson disease pathogenesis. *The FEBS Journal*, 285(19):3591–3603, 2018.
52. CRA Silveira, J MacKinley, K Coleman, Z Li, E Finger, R Bartha, SA Morrow, J Wells, M Borrie, RG Tirona, et al. Amroxol as a novel disease-modifying treatment for parkinson's disease dementia: Protocol for a single-centre, randomized, double-blind, placebo-controlled trial. *BMC neurology*, 19:1–10, 2019.
53. Alisdair McNeill, Joana Magalhaes, Chengguo Shen, Kai-Yin Chau, Derrilyn Hughes, Atul Mehta, Tom Foltynie, J Mark Cooper, Andrey Y Abramov, Matthew Gegg, et al. Amroxol improves lysosomal biochemistry in glucocerebrosidase mutation-linked parkinson disease cells. *Brain*, 137(5):1481–1495, 2014.
54. Stephen Mullin, Laura Smith, Katherine Lee, Gayle D'Souza, Philip Woodgate, Josh Effein, Jenny Hällqvist, Marco Toffoli, Adam Streeter, Joanne Hosking, et al. Amroxol for the treatment of patients with parkinson disease with and without glucocerebrosidase gene mutations: a nonrandomized, noncontrolled trial. *JAMA neurology*, 77(4):427–434, 2020.
55. Alexandra Bouscary, Cyril Quessada, Althéa Mosbach, Noëlle Callizot, Michael Spedding, Jean-Philippe Loeffler, and Alexandre Henriques. Amroxol hydrochloride improves motor functions and extends survival in a mouse model of familial amyotrophic lateral sclerosis. *Frontiers in pharmacology*, 10:883, 2019.
56. Jing Wang, Denis Gallagher, Loren M DeVito, Gonzalo I Cancino, David Tsui, Ling He, Gordon M Keller, Paul W Frankland, David R Kaplan, and Freda D Miller. Metformin activates an atypical pkc-cbp pathway to promote neurogenesis and enhance spatial memory formation. *Cell stem cell*, 11(1):23–35, 2012.
57. Jared M Campbell, Matthew D Stephenson, Barbora De Courten, Ian Chapman, Susan M Bellman, and Edoardo Aromataris. Metformin use associated with reduced risk of dementia in patients with diabetes: a systematic review and meta-analysis. *Journal of Alzheimer's Disease*, 65(4):1225–1236, 2018.
58. Amit Gupta, Bharti Bisht, and Chinmoy Sankar Dey. Peripheral insulin-sensitizer drug metformin ameliorates neuronal insulin resistance and alzheimer's-like changes. *Neuropharmacology*, 60(6):910–920, 2011.
59. Nikita Katila, Sunil Bhurtel, Sina Shadfar, Sunil Srivastav, Sabita Neupane, Uttam Ojha, Gil-Saeng Jeong, and Dong-Young Choi. Metformin lowers α -synuclein phosphorylation and upregulates neurotrophic factor in the mptp mouse model of parkinson's disease. *Neuropharmacology*, 125:396–407, 2017.
60. David S Siegel, Thomas Martin, Michael Wang, Ravi Vij, Andrzej J Jakubowiak, Sagar Lonial, Suzanne Trudel, Vishal Kukreti, Nizar Bahlis, Melissa Alsina, et al. A phase 2 study of single-agent carfilzomib (px-171-003-a1) in patients with relapsed and refractory multiple myeloma. *Blood, The Journal of the American Society of Hematology*, 120(14):2817–2825, 2012.
61. Qiuyang Zheng, Timothy Huang, Lishan Zhang, Ying Zhou, Hong Luo, Huaxi Xu, and Xin Wang. Dysregulation of ubiquitin-proteasome system in neurodegenerative diseases. *Frontiers in aging neuroscience*, 8:303, 2016.
62. Zaira Ortega and Jose J Lucas. Ubiquitin-proteasome system involvement in huntington's disease. *Frontiers in molecular neuroscience*, 7:77, 2014.
63. Chris McKinnon and Sarah J Tabrizi. The ubiquitin-proteasome system in neurodegeneration. *Antioxidants & redox signaling*, 21(17):2302–2321, 2014.
64. Christopher A Ross and Michelle A Poirier. Protein aggregation and neurodegenerative disease. *Nature medicine*, 10(Suppl 7):S10–S17, 2004.
65. Jianing Jin, Li Xue, Xinling Bai, Xiaona Zhang, Qingwu Tian, and Anmu Xie. Association between epidermal growth factor receptor gene polymorphisms and susceptibility to parkinson's disease. *Neuroscience Letters*, 736:135273, 2020.
66. Pusheng Quan, Kai Wang, Shi Yan, Shirong Wen, Chengqun Wei, Xinyu Zhang, Jingwei Cao, and Lifan Yao. Integrated network analysis identifying potential novel drug candidates and targets for parkinson's disease. *Scientific Reports*, 11(1):1–9, 2021.
67. Xinyi Wang, Mingyang Zhang, Chunlin Long, Lin Yao, and Min Zhu. Self-attention based neural network for predicting rna-protein binding sites. *IEEE/ACM Transactions on Computational Biology and Bioinformatics*, 20(2):1469–1479, 2022.
68. Zhenshen Pan, Shusen Zhou, Hailin Zou, Chanjuan Liu, Mujun Zang, Tong Liu, and Qingjun Wang. Crmsnet: A deep learning model that uses convolution and residual multi-head self-attention block to predict rbps for rna sequence. *Proteins: Structure, Function, and Bioinformatics*, 2023.
69. Haixin Lv, Jinglong Chen, Tongyang Pan, Tianci Zhang, Yong Feng, and Shen Liu. Attention mechanism in intelligent fault diagnosis of machinery: A review of technique and application. *Measurement*, page 111594, 2022.
70. Keisuke Yamada and Michiaki Hamada. Prediction of rna-protein interactions using a nucleotide language model. *Bioinformatics Advances*, 2(1):vbac023, 2022.
71. Yuning Yang, Zilong Hou, Yansong Wang, Hongli Ma, Pingping Sun, Zhiqiang Ma, Ka-Chun Wong, and Xiangtao Li. Hcnet: high-throughput circrna-binding event identification from clip-seq data using deep temporal convolutional network. *Briefings in Bioinformatics*, 23(2), 2022.
72. Chao Cao, Shuhong Yang, Mengli Li, and Chungui Li. Circssnn: circrna-binding site prediction via sequence self-attention neural networks with pre-normalization. *BMC bioinformatics*, 24(1):220, 2023.
73. Xiuquan Du and Zhigang Xue. Jlcrb: A unified multi-view-based joint representation learning for circrna binding sites prediction. *Journal of Biomedical Informatics*, 136:104231, 2022.
74. Kaiming He, Xiangyu Zhang, Shaoqing Ren, and Jian Sun. Deep residual learning for image recognition. In *Proceedings of the IEEE conference on computer vision and pattern recognition*, pages 770–778, 2016.
75. Mengting Niu, Quan Zou, and Chen Lin. Crbpd1: Identification of circrna-rbp interaction sites using an ensemble neural network approach. *PLoS computational biology*, 18(1):e1009798, 2022.
76. Lishen Zhang, Chengqun Lu, Min Zeng, Yaohang Li, and Jianxin Wang. Crmss: predicting circrna-rbp binding sites based on multi-scale characterizing sequence and structure features. *Briefings*

- in *Bioinformatics*, 24(1):bbac530, 2023.
77. Jernej Ule, Kirk B Jensen, Matteo Ruggiu, Aldo Mele, Aljaz Ule, and Robert B Darnell. Clip identifies nova-regulated rna networks in the brain. *Science*, 302(5648):1212–1215, 2003.
 78. Jun-Hao Li, Shun Liu, Hui Zhou, Liang-Hu Qu, and Jian-Hua Yang. starbase v2. 0: decoding mirna-erna, mirna-ncrna and protein-rna interaction networks from large-scale clip-seq data. *Nucleic acids research*, 42(D1):D92–D97, 2014.
 79. Gerd Anders, Sebastian D Mackowiak, Marvin Jens, Jonas Maaskola, Andreas Kuntzagk, Nikolaus Rajewsky, Markus Landthaler, and Christoph Dieterich. dorina: a database of rna interactions in post-transcriptional regulation. *Nucleic acids research*, 40(D1):D180–D186, 2012.
 80. Jingna Si, Jing Cui, Jin Cheng, and Rongling Wu. Computational prediction of rna-binding proteins and binding sites. *International journal of molecular sciences*, 16(11):26303–26317, 2015.
 81. Junkang Wei, Siyuan Chen, Licheng Zong, Xin Gao, and Yu Li. Protein-rna interaction prediction with deep learning: structure matters. *Briefings in bioinformatics*, 23(1):bbab540, 2022.
 82. Jing Yan, Stefanie Friedrich, and Lukasz Kurgan. A comprehensive comparative review of sequence-based predictors of dna-and rna-binding residues. *Briefings in bioinformatics*, 17(1):88–105, 2016.
 83. Zhichao Miao and Eric Westhof. A large-scale assessment of nucleic acids binding site prediction programs. *PLoS computational biology*, 11(12):e1004639, 2015.
 84. Jian Zhang and Lukasz Kurgan. Scriber: accurate and partner type-specific prediction of protein-binding residues from proteins sequences. *Bioinformatics*, 35(14):i343–i353, 2019.
 85. Yang Liu, Weikang Gong, Yanpeng Zhao, Xueqing Deng, Shan Zhang, and Chunhua Li. aprbind: protein-rna interface prediction by combining sequence and i-tasser model-based structural features learned with convolutional neural networks. *Bioinformatics*, 37(7):937–942, 2021.
 86. Jing Yan and Lukasz Kurgan. Drnapred, fast sequence-based method that accurately predicts and discriminates dna-and rna-binding residues. *Nucleic acids research*, 45(10):e84–e84, 2017.
 87. José Jiménez, Stefan Doerr, Gerard Martínez-Rosell, Alexander S Rose, and Gianni De Fabritiis. Deepsite: protein-binding site predictor using 3d-convolutional neural networks. *Bioinformatics*, 33(19):3036–3042, 2017.
 88. Ying Xia, Chun-Qiu Xia, Xiaoyong Pan, and Hong-Bin Shen. Graphbind: protein structural context embedded rules learned by hierarchical graph neural networks for recognizing nucleic-acid-binding residues. *Nucleic acids research*, 49(9):e51–e51, 2021.
 89. Fuhao Zhang, Bi Zhao, Wenbo Shi, Min Li, and Lukasz Kurgan. Deepdisobind: accurate prediction of rna-, dna-and protein-binding intrinsically disordered residues with deep multi-task learning. *Briefings in bioinformatics*, 23(1):bbab521, 2022.
 90. Jordy Homing Lam, Yu Li, Lizhe Zhu, Ramzan Umarov, Hanlun Jiang, Amélie Héliou, Fu Kit Sheong, Tianyun Liu, Yongkang Long, Yunfei Li, et al. A deep learning framework to predict binding preference of rna constituents on protein surface. *Nature communications*, 10(1):4941, 2019.

Reviewer #1 (Remarks to the Author):

I think the current version is good enough for acceptance.

Reviewer #2 (Remarks to the Author):

The authors have addressed my comments, and the additional data have strengthened the study's conclusions.

Reviewer #3 (Remarks to the Author):

The authors have addressed my comments.

Authors Point-by-Point Responses to the Reviewers' comments "Dynamic characterization and interpretation for protein-RNA interactions across diverse cellular conditions using HDRNet" by Zhu et al. (NCOMMS-23-08167A) submitted to *Nature Communications* on 9th-August-2023.

Reviewer #1:

I think the current version is good enough for acceptance.

Response: Thank you very much for your positive comments. They were highly appreciated and improved the manuscript tremendously. Thank you very much for your precious time and kind attention.

Reviewer #2:

The authors have addressed my comments, and the additional data have strengthened the study's conclusions.

Response: Thank you very much for the positive comments and thank you for the detailed suggestions. Thank you very much for your precious time and kind attention.

Reviewer #3:

The authors have addressed my comments.

Response: Thank you very much for all your comments, which were highly appreciated have made a significant improvement to the manuscript. Thank you very much for your precious time and kind attention.

DRAFT